# Extensive pedigrees reveal the social organization of a Neolithic community

**Maïté Rivollat**[1,2,3,11 ✉], **Adam Benjamin Rohrlach**[2,12], **Harald Ringbauer**[2], **Ainash Childebayeva**[2], **Fanny Mendisco**[1], **Rodrigo Barquera**[2], **András Szolek**[4,13], **Mélie Le Roy**[5], **Heidi Colleran**[6,14], **Jonathan Tuke**[7], **Franziska Aron**[8,15], **Marie-Hélène Pemonge**[1], **Ellen Späth**[8], **Philippe Télouk**[9], **Léonie Rey**[1], **Gwenaëlle Goude**[10], **Vincent Balter**[9], **Johannes Krause**[2], **Stéphane Rottier**[1 ✉], **Marie-France Deguilloux**[1] & **Wolfgang Haak**[2 ✉]

Social anthropology and ethnographic studies have described kinship systems and networks of contact and exchange in extant populations[1–4]. However, for prehistoric societies, these systems can be studied only indirectly from biological and cultural remains. Stable isotope data, sex and age at death can provide insights into the demographic structure of a burial community and identify local versus non-local childhood signatures, archaeogenetic data can reconstruct the biological relationships between individuals, which enables the reconstruction of pedigrees, and combined evidence informs on kinship practices and residence patterns in prehistoric societies. Here we report ancient DNA, strontium isotope and contextual data from more than 100 individuals from the site Gurgy 'les Noisats' (France), dated to the western European Neolithic around 4850–4500 BC. We find that this burial community was genetically connected by two main pedigrees, spanning seven generations, that were patrilocal and patrilineal, with evidence for female exogamy and exchange with genetically close neighbouring groups. The microdemographic structure of individuals linked and unlinked to the pedigrees reveals additional information about the social structure, living conditions and site occupation. The absence of half-siblings and the high number of adult full siblings suggest that there were stable health conditions and a supportive social network, facilitating high fertility and low mortality[5]. Age-structure differences and strontium isotope results by generation indicate that the site was used for just a few decades, providing new insights into shifting sedentary farming practices during the European Neolithic.

Kinship and biological relatedness are difficult to assess in prehistoric societies. With the optimization of ancient DNA (aDNA) methods, it is now feasible to obtain genome-wide data and reconstruct precise genetic relationships between individuals buried at the same site. Combined with evidence from archaeological, anthropological and isotopic records, information on the biological ties between individuals can provide a background against which basic elements of social relationships (kinship organization, residence or migration patterns) can be inferred or ruled out.

Studies on biological relatedness in the European Neolithic are still rare and to date have focussed only on groups from specific funerary contexts such as megaliths[6–8], which typically cover high-status groups or individuals, or mass graves[9], but have not included non-specific graveyards that may be more representative of the general population. The Paris Basin during the Middle Neolithic (around 4700–4300 BC) is well-known for the emergence of the monumental funerary architectural structures of the Passy phenomenon[10] dedicated to select individuals from Neolithic communities. Gurgy 'les Noisats', a burial site in the Cerny cultural horizon[11] without any monumental architecture, is located close to a dozen contemporaneous monumental sites (in a radius of 100 km; Supplementary Note 1 and Supplementary Fig. 1). With skeletal remains of 128 individuals, Gurgy is the biggest cemetery in the region, and is dated to the fifth millennium BC[12,13]. The burials feature different body positions and orientations, and architectural

[1]University of Bordeaux, CNRS, PACEA - UMR 5199, Allée Geoffroy Saint-Hilaire, Pessac, France. [2]Department of Archaeogenetics, Max Planck Institute for Evolutionary Anthropology, Leipzig, Germany. [3]Department of Archaeology, Durham University, Durham, UK. [4]Department of Immunology, Interfaculty Institute for Cell Biology, University of Tübingen, Tübingen, Germany. [5]Department of Archaeology & Anthropology, Bournemouth University, Bournemouth, UK. [6]Department of Human Behavior, Ecology and Culture, Max Planck Institute for Evolutionary Anthropology, Leipzig, Germany. [7]School of Computer and Mathematical Sciences, University of Adelaide, Adelaide, South Australia, Australia. [8]Leibniz Institute on Aging—Fritz Lipmann Institute (FLI), Jena, Germany. [9]Ecole Normale Supérieure de Lyon, CNRS, UCBL, LGL-TPE, Lyon, France. [10]CNRS, Aix Marseille University, Ministry of Culture, LAMPEA, Aix-en-Provence, France. [11]Present address: Department of Archaeology, Ghent University, Ghent, Belgium. [12]Present address: School of Computer and Mathematical Sciences, University of Adelaide, Adelaide, South Australia, Australia. [13]Present address: Applied Bioinformatics, Department of Computer Science, University of Tübingen, Tübingen, Germany. [14]Present address: BirthRites Lise Meitner Research Group, Max Planck Institute for Evolutionary Anthropology, Leipzig, Germany. [15]Present address: RNA Bioinformatics and High Throughput Analysis, Friedrich Schiller University, Jena, Germany. ✉e-mail: maite.rivollat@ugent.be; stephane.rottier@u-bordeaux.fr; wolfgang_haak@eva.mpg.de

variation from various cultural influences, but very few grave goods, limiting our ability to identify either a direct association to the Cerny culture, the organization of the site or the selection of specific individuals (Supplementary Note 1). Overall, an estimate of the duration of the use of the site was impossible to assess on the basis of the archaeological evidence.

To investigate the intrasite structure and the characteristics of Gurgy, we sampled the remains of 110 out of 128 individuals with suitable skeletal preservation (Methods) and retrieved genome-wide aDNA data for 94 individuals (Supplementary Table 3), data from 22 of whom were published previously[14]. We also generated immune gene data for 82 individuals, mitogenome data for 99 individuals and Y-chromosome data for 57 individuals (Supplementary Notes 6 and 7 and Supplementary Tables 4 and 5). To contextualize the new genomic data, we also generated $^{87}$Sr/$^{86}$Sr ratio data for 57 individuals (Supplementary Table 22) and report eight new radiocarbon dates (Supplementary Table 1).

## Large family trees
To estimate the biological relatedness between Gurgy individuals, we used two methods (READ[15] and lcMLkin[16]) that are suitable for low-coverage DNA data and are widely used in aDNA research, and can reliably detect relatedness up to the second degree and differentiate between first-degree parent–offspring and sibling relationships (Supplementary Note 2, Supplementary Tables 8 and 9 and Extended Data Fig. 1). We combined the results with haplogroup data from uniparentally inherited markers (mitochondrial DNA (mtDNA) and Y-chromosome lineages; Supplementary Tables 4–7 and Extended Data Figs. 2 and 3), as well as age-at-death and genetic sex to establish small initial pedigrees (Supplementary Note 2 and Supplementary Fig. 2), spanning two to three generations, that we then expanded incrementally, resulting ultimately in two large pedigrees. Pedigree A connects 64 individuals (20 female and 44 male) over 7 generations, and pedigree B connects 12 individuals (7 female and 5 male) over 5 generations (Fig. 1a). Among the remaining 18 individuals, 1 male adult has 2 second-degree relatives in pedigree A. We identified 3 additional pairs of first-degree relatives, and 11 remaining individuals who are not closely related to either of the pedigrees (Fig. 1a).

Throughout this text we use the terms mother/father, son/daughter and siblings, as well as the binary sex terms male and female, in the genetic sense. We acknowledge that these are western kinship terms, but they are not meant to imply kinship terminologies or identities here. We cannot know if they were understood in this way by the Gurgy community.

We also used a recently developed method to analyse shared identity-by-descent (IBD) blocks between individuals on the basis of imputed data, which enabled us to estimate degrees of relatedness reliably up to the fourth to fifth degree and to distinguish between lineal (direct generational) and non-lineal descent for intermediate-quality aDNA data[17] (>500,000 single-nucleotide polymorphisms (SNPs), $n = 72$; Methods, Supplementary Note 3, Supplementary Table 10 and Extended Data Fig. 1). The results of the IBD sharing analysis are fully consistent with the reconstructed pedigrees. Moreover, we detected more distant connections (Fig. 2, Supplementary Note 3 and Extended Data Fig. 4) that were also visible from an $f_3$-outgroup heat map comparing each pair of individuals (Supplementary Note 5, Supplementary Tables 10 and 11 and Extended Data Fig. 4). Notably, both pedigrees are linked through a third–fourth-degree relationship between GLN263 and GLN298, but we could not infer the exact relationship given the multiple alternative possibilities (Fig. 2 and Supplementary Note 3). A pair of siblings among the 18 remaining individuals, GLN211A and GLN211B, is also connected more distantly to the siblings of generation 3 of both larger pedigrees (more distantly than the third degree, also resulting in multiple alternative possibilities; Fig. 2 and Supplementary Note 3).

Finally, by using human leukocyte antigen (HLA) class I and II haplotypes, we reconstructed the transmission of biparental haplotypes in each generation, again confirming the larger pedigrees and revealing two recombination events in individuals GLN245B and GLN267 between both haplotypes of their respective parents (Supplementary Note 8, Supplementary Tables 12 and 13 and Extended Data Fig. 5).

## Social relationships and residence
Examining both pedigrees, we found that generations are linked almost exclusively through the male line, that is, all descendants but one are connected to the family tree through their father's line. The Y-chromosome haplogroup G2a2b2a1a2 (terminal SNP Z38302) is carried by 51 out of 57 male individuals and is the main male lineage of the group (Supplementary Table 7 and Extended Data Fig. 2). Pedigree A yields the sole exception, where a new lineage (C in Fig. 1a) is linked through a lineage woman (GLN325). Her reproductive partner and his brother, her sons and two other unrelated male individuals carry haplogroup H2m, the only other Y-chromosome lineage observed in the dataset (Fig. 1a and Extended Data Fig. 2). The main genealogical lineage of pedigree A is also visible in the funerary features, as one son, GLN237A, and one grandson, GLN221B, of the main ancestor GLN270B are buried in the largest pits of the necropolis (Fig. 1a, Supplementary Note 1 and Extended Data Fig. 6). GLN270B, the main ancestor of 52 individuals in pedigree A, represents the only secondary burial at the site, consisting of long bones that were buried together with the female individual GLN270A, for whom no genomic data could be obtained (Fig. 1d). The position at the apex of the pedigree suggests that his remains were transferred and buried during the early occupation of the site, representing, together with his brother GLN231A, the main male posthumous ancestors of pedigree A. The association between GLN270B and female individual GLN270A, and not with his brother, for example, suggests that this female individual was important—perhaps she was his partner, or someone genetically related representing the lineage.

When we investigated the spatial organization of the graves by measuring the physical distance between individuals, we observed significantly closer spatial proximity between each father–son pair than any other related pair (Fig. 1e, Supplementary Note 12 and Supplementary Table 20). Patterns of spatial organization beyond the specific father–son connections (Supplementary Note 12) seem to follow clusters of genetically closely related individuals (Fig. 1a,c). Indeed, after an initial phase of early burials that were grouped together in the eastern part of the funerary area, the siblings from the fourth generation were all buried near each other (Fig. 1a,c). For example, the four siblings of individual GLN317 were buried west of him, whereas the mother of his sons was buried east of him, and GLN223 and his son's daughter was buried on top of him. The other son of GLN317, GLN202, probably died later as he was buried in another part of the necropolis, together with other branches of pedigree A, and was possibly the most recently deceased of this family line. We co-analysed the spatial distribution of the few grave goods and adornments attributed to the dead, such as ornaments or ochre, and potential transmission along the paternal lineage using a Geographic Information System (Supplementary Note 12 and Extended Data Figs. 3 and 6). However, perhaps due to the general paucity of grave goods at the site (Supplementary Table 21), no association was detected. We also did not find any correlations between specific body positions (crouched/flexed or elongated), sidedness and orientation, type of grave and genetic lineages, genetic 'nuclear' families (that is, mother–father–child trios), and/or unlinked/unrelated individuals (Supplementary Note 12 and Extended Data Fig. 3). However, the spatial layout with minimal or no overlap between burials suggests that graves were visible or marked on the surface[18], and the pattern of expansion indicates that people knew who was buried where, and may have acknowledged lines of descent accordingly.

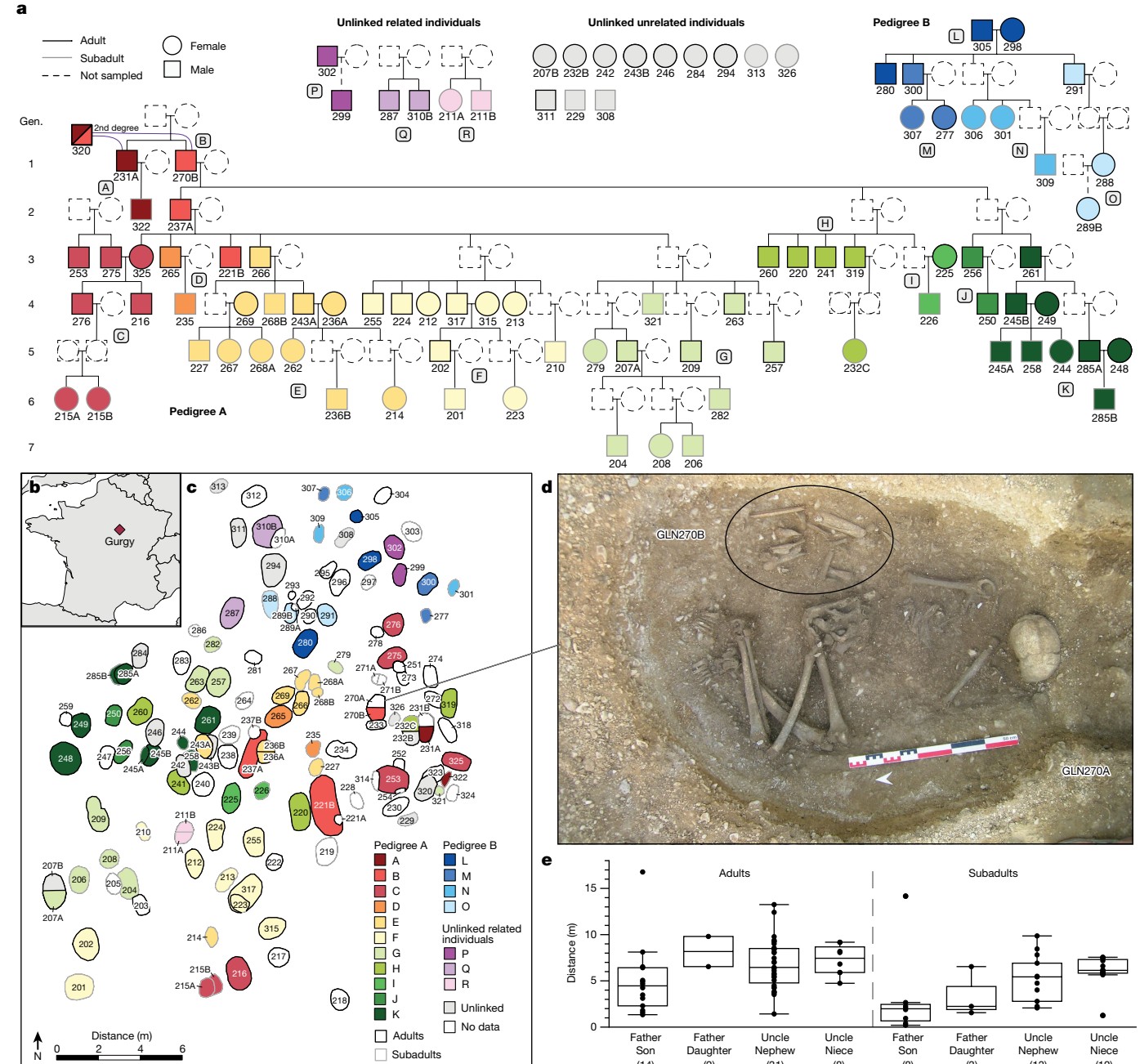

**Fig. 1 | Genetic relatedness at Gurgy in light of the spatial layout and generational succession. a**, Reconstructed pedigrees of the Gurgy group coloured according to family lineages (lineages A–R, according to the colour scale in **c**). Gen., generation. **b**, The geographical location of the Gurgy 'les Noisats' site in present-day France. The map was created using the R packages maps (v.3.3.0) and mapdata (v.2.3.0). **c**, The site layout, representing the spatial distribution of family lineages coloured as in **a. d**, Photograph of female individual GLN270A (no genetic results) with the reburied remains of the main male ancestor GLN270B of pedigree A. **e**, The spatial distances of father–offspring and uncle–nephew/niece pairs (the number of pairs is given in parentheses; Supplementary Table 20). Fathers and subadult sons are, on average, buried significantly closer to each other than any other pairs (Supplementary Note 12). The centre line shows the median, the box limits delineate the interquartile range and the whiskers extend to the maximum and minimum values, excluding the outliers.

Together, we observed a general trend at the biological and archaeological level, in which individuals are linked through the male lineage, potentially indicative of local understandings of genealogy or descent.

The pedigree structure of the burial community reveals further insights into the residential organization of the living. Apart from two individuals (GLN325 (see above) and GLN288), no adult mothers, present (n = 7) or absent, have parents/ancestors buried at the site. This suggests an exogenous origin of these females (Extended Data

Figs. 2 and 7). Moreover, only 6 out of the 20 female adults buried at the site are descendants of the main pedigree lines A and B. Another seven female adults buried at Gurgy had very few biological relationships with other individuals, and mainly not of the main pedigrees, as shown by IBD analyses (Fig. 2). One plausible hypothesis is that they were companions of male individuals from the main pedigrees: no joint children were buried on site, nor could they be linked through other individuals to the pedigrees. Indeed, 17 adult male individuals have no

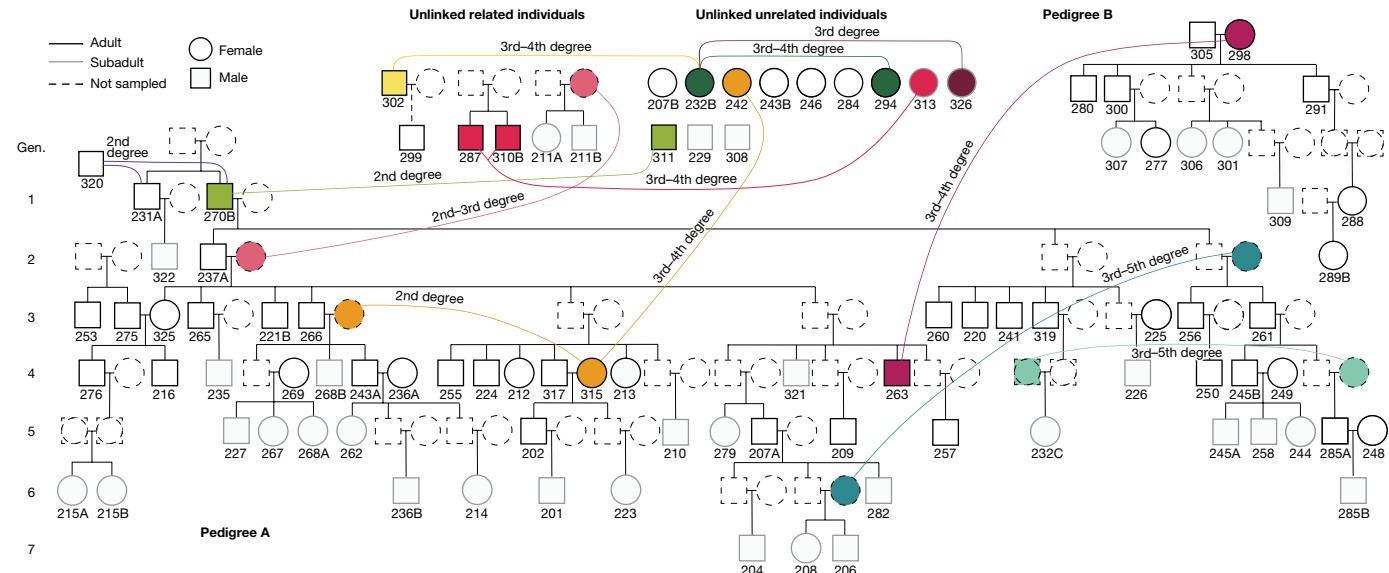

**Fig. 2 | IBD links.** Additional biological relatedness up to the fifth degree between individuals within and between the previously reconstructed pedigrees as revealed by the analysis of shared IBD blocks and $f_3$ statistics (Supplementary Tables 10 and 12 and Supplementary Notes 3 and 5).

children buried at the site, of which 13 are linked through their parents to either main pedigree. This general pattern points towards female exogamy and a virilocal residential system in which females in-migrated from their birthplace to their male reproductive partner's residence. Consistent with this pattern, additional links observed between the isolated female adults and the pedigrees could be due to (1) distantly related female individuals stemming from the same community; or (2) women who left the Gurgy community in previous generations with female descendants who subsequently returned to Gurgy. The latter scenario is indicative of reciprocal exchange typical in moiety systems[3]. Perhaps as a consequence, the sex ratio of adult offspring buried on site is unbalanced at 4.5:1 (confidence interval (CI) = 64.5–93%) in favour of male individuals ($n$ = 27 versus $n$ = 6, for male and female, respectively). By contrast, a sex ratio of 1.06:1 (CI = 34.4–68.1%) among subadult offspring ($n$ = 19 and $n$ = 18, for male and female, respectively) matches the natural expected ratio 1.05:1 at birth[19], ruling out sex-biased cultural practices affecting the subadult population (Extended Data Fig. 2). The vast majority of the subadults are younger than 15 years old ($n$ = 34), with most of these individuals being younger than 8 years old ($n$ = 27), and in equal proportions for both sexes. The differences between younger and older-age sex ratios suggest that older daughters, from around the age of 15, left to join new groups, again consistent with a female exogamic residential system. For four out of the six adult lineage daughters (GLN212, 213, 277 and 289B) who remained at Gurgy, no offspring could be identified at the site even though they had reached reproductive age. Female exogamy may not have been practiced strictly or, alternatively, these lineage daughters could be reproductive partners of unlinked adult males (with no offspring linking them to the pedigrees)—a scenario that further complicates the assumption of strict patrilocality and female exogamy. Alternative reasons for their stay in the community remain unclear.

In this context, we observed that women of genetically exogenous provenance tend to be spatially integrated into their reproductive partner's burial area, suggesting social integration into the host group (Fig. 1a). However, considering the 42 reproductive unions observed across all pedigrees that evidently had offspring buried at Gurgy, we noticed a shortage of mothers, with only 9 versus 20 fathers buried there. This imbalance is also observed in the total number of adult burials (38 male versus 20 female), and suggests that male adults were twice as likely to be buried than female adults. We therefore observed

a potential sex bias in burials independent of female exogamy. This could be explained by different funerary practices being reserved for these mothers, or by other social factors mitigating against a co-burial with their reproductive partner's group.

To gain independent information on individual mobility, we performed strontium isotope analyses ($^{87}$Sr/$^{86}$Sr) using laser ablation[20] on 57 individuals (Fig. 3, Supplementary Note 11, Supplementary Table 23 and Extended Data Figs. 3 and 8). Unrelated female adults and some of those with no parents at the site show lower $^{87}$Sr/$^{86}$Sr ratio values compared with male individuals from the same generation (Fig. 3 and Extended Data Fig. 3). Although the geological reference map does not enable us to infer a specific geographical origin, this finding provides further evidence that these female individuals grew up in different places before joining the Gurgy community (Supplementary Note 11 and Extended Data Fig. 8). Published stable isotope data (carbon, nitrogen, sulphur) measured on bones highlight a significant, sex-biased dietary division in adults[21]. On average, male individuals yielded higher $\delta^{13}$C and $\delta^{15}$N, and lower $\delta^{34}$S values, than female individuals, which could reflect a separation by sex, but could also signal female mobility (Supplementary Table 24). Genetic sex determination of subadults enabled us to confirm this difference also in childhood ($P$ = 0.01019), which could be explained by a sex-related differential treatment at certain ages, determined by social rules (Supplementary Note 14 and Extended Data Fig. 5). Notably, the funerary practices in Gurgy show a shift at around 7–8 years of age, when the children are buried with different types of grave goods compared with younger ages, and another shift at around 15–16 years of age when they are associated with the same grave goods as adults, which could reflect local age stages or other social thresholds. This pattern has previously been observed at other Neolithic sites in the northern half of France[22] (Supplementary Table 1).

The presumed patrilocal residential pattern in the Gurgy group also explains the mitochondrial diversity despite the deficit in female adults (35 different mitochondrial haplogroups are carried by 99 individuals; Supplementary Note 6, Supplementary Table 5 and Extended Data Fig. 3). Indeed, no mitochondrial haplogroup was transmitted further than one daughter/son generation, and the incoming mothers of each generation contributed new mtDNA lineages, except for the female descendant of the main lineage GLN325 who transmitted hers one generation further. By examining the affinities between all of the exogenous female adults, we demonstrate that they are not closely

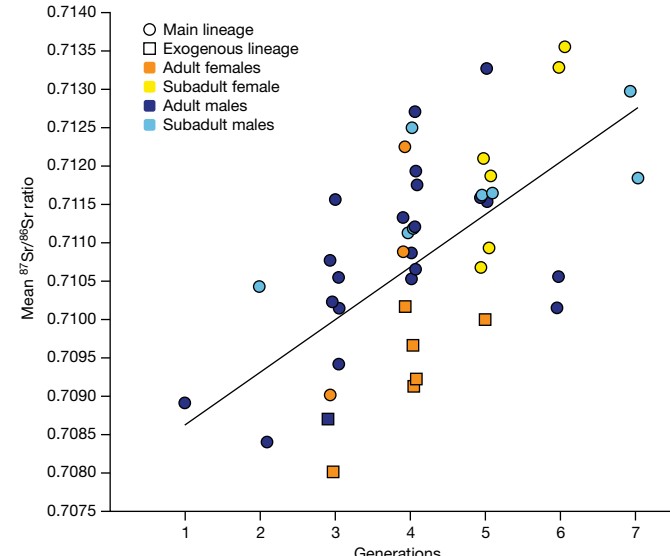

**Fig. 3 | Strontium data of pedigree A.** Mean $^{87}$Sr/$^{86}$Sr ratio per age and sex cohort across generations. A significant difference between sex was observed per generation (two-sided analysis of variance, $P = 0.01474$; Supplementary Note 11).

related, apart from two pairs of third- or fourth-degree relatives (Supplementary Note 5 and Extended Data Fig. 4). Moreover, this female diversity within the Gurgy group might also explain the overall phenotypic variation observed at the site (Supplementary Note 9 and Supplementary Tables 15 and 16). Taken together, these results suggest that the Gurgy community maintained a fairly clear pattern of female exogamy that may have been driven by a range of group features (for example, population size, resource access, network position) or identities (for example, linguistic or cultural affinities). The absence of long runs of homozygosity (ROH), typical for close-kin consanguinity, confirms the avoidance of reproduction between closely related individuals, except for a single individual (GLN282), with an amount of ROH consistent with a union between second- or third-degree relatives (Supplementary Note 4, Supplementary Table 11 and Extended Data Fig. 9). IBD sharing also revealed that groups of individuals in pedigree A are more related than expected from connections within the reconstructed pedigree alone (Fig. 2 and Supplementary Note 3). This can only be explained if there were additional relatives through maternal lines (which were not sampled). In our case, some mothers probably came from the same external group, only several generations apart, which would link different branches of the pedigree. For example, siblings GLN243A and GLN268B share the same mitochondrial haplotype H1 as GLN315 (Supplementary Table 5) and IBD-sharing typical of third-degree relationships, whereas GLN315 is an exogenous female individual. We interpret these as a second-degree relationship between GLN315 and the missing mother of GLN243A and 268B (Fig. 2). These additional connections through the female lines indicate a network of relationships with other groups, including occasional unions with (distantly) related women from the same source group. This pattern suggests preferential links or dependencies between some groups, albeit within a network of groups sufficiently large or diverse to sustain diversity of background relatedness and to avoid close-kin consanguinity.

Further insights about the social organization of the group can be gleaned from the notable lack of half-siblings in the entire sample, in contrast to recent findings from a later Neolithic long cairn in England[8] (Supplementary Notes 2 and 3). This indicates that polygamous reproductive unions were uncommon or perhaps socially proscribed, or that the burial of offspring from such unions was carried out elsewhere. Likewise, it also suggests that serial monogamy, including levirate and sororate unions in which a woman repartners with her deceased husband's brother or a man repartners with his wife's sister, was rare. We find this observation surprising given potential imbalances in the female/male sex-ratio, for example, an elevated risk of death from complications during childbirth (for female individual), potential conflicts or diseases in prehistoric societies. The pedigrees show no evidence in support of these assumptions. Indeed, if exogamous reproductive unions were routinely contracted with numerous groups for purposes of, for example, alliance or trade, then between-group networks of cooperation, rather than conflict, are implied.

Including unsampled, inferred adults, we observed two cases of up to six offspring from the same couple (Extended Data Fig. 6). Notably, all six full siblings had reached reproductive age, with several of those having four and five adult offspring on their own. Moreover, the majority of adult offspring being male individuals points towards additional unsampled female siblings (to statistically account for an equivalent number of females born[19]), as well as possibly a significant number of deceased infants expected at that period of time. These large family sizes suggest a high fertility rate and generally stable conditions of health and nutrition in this Neolithic community[5]—a fact that is also supported by stable isotope data[23]. Indeed, one could speculate that different elements, such as a potential emphasis on sublineage reproductive and/or productive units, spatial co-residence of numerous reproductive units and divisions of labour that may facilitate efficient reproduction, plausibly provide conditions for cooperative breeding that can generate high rates of population growth[24]. We estimated the effective population size of the communities contributing to the diversity observed at Gurgy to have been around 1,835 individuals (95% CI = 1,631–2,077)[25] (Supplementary Note 4 and Supplementary Table 26). The distribution of ROH in the group (Supplementary Table 11 and Extended Data Fig. 9) suggests that most pairs of parents were related to each other through co-ancestors within the preceding 5–30 generations (Supplementary Note 4).

## Occupation time of the site

The two large pedigrees are reflected in the spatial layout of the necropolis. Pedigree A occupies the main space, whereas pedigree B is located on the north-eastern side (Fig. 1c). Overlapping $^{14}$C date ranges and the third- to fourth-degree connection between GLN263 and GLN298 suggest relative contemporaneity of both pedigrees. In both pedigrees and with the secondary deposit of the main ancestor GLN270B in pedigree A, we observed a spread from the founder generations towards the south-west, by generations through time (Extended Data Fig. 6). Spatial distances are significantly correlated with genetic distances (Mantel test, $r = 0.2$, $P < 0.001$; Supplementary Note 12).

The pedigrees reveal an absence of subadults among the first four generations in pedigree A (5 out of 36 individuals; Extended Data Fig. 2), which is surprising given the expected mortality patterns in archaic populations[26,27]. Notably, this trend is reversed across the last three generations, with 20 out of 25 individuals being subadults. These observations are consistent with a scenario in which an entire group of several generations moved to this new burial site, leaving behind their deceased children at a previous funerary site, but transferring the 'ancestor/founder' GLN270B. Moreover, the fact that many parents are missing in the last generations suggests that the group moved on to settle elsewhere, leaving behind children who had passed away. The Sr isotope data provide a further line of evidence for these interpretations, as $^{87}$Sr/$^{86}$Sr values were low (around 0.709) for the earlier generations, overall similar to exogenous female individuals (Fig. 3, Supplementary Note 11, Supplementary Tables 23 and 28 and Extended Data Fig. 3), indicating a non-local origin of the founders. Strontium isotope ratios in male individuals then continuously increased with generations, resembling the local signal.

Despite reconstructing seven generations in pedigree A, the occupation time of the site was relatively short. Excluding the founding and migrating generations, the duration of the site use was probably only 3–4 generations or 84–112 years (1 generation is 28 years[28]). Bayesian modelling of all available radiocarbon dates (*n* = 33) enabled us to constrain the interval for pedigree A between the late forty-eighth and the late forty-seventh century BC (Supplementary Note 13 and Extended Data Fig. 10). We speculate that the use of the gravesite corresponds to the duration of dwellings. The typical duration of a long-house of the Neolithic Linear Pottery culture was estimated between 20–30 years[29] and up to 75–100 years when maintained[30], and experimental archaeology has suggested that the lake dwellings in the Jura of the late Neolithic period could last for a period of about 10 years without proper maintenance[31]. However, no settlement was found directly associated with the Gurgy gravesite, precluding the integration of contextual details. An alternative or complementary explanation for a limited occupation time could be the depletion of local soils and other natural resources, driven by non-sustainable agricultural practices that could have taken different forms, which is still a subject of intense debate[31,32]. In fact, strontium isotope values suggest a move of each generation to another geographical location, while maintaining a common burial ground, providing an additional argument for intergenerational mobility within a local territory (Supplementary Note 14.3).

The combined data from Gurgy indicate a group organized into potentially segmented pedigree groups of biological relatedness, who used the burial site for a limited number of generations. Our results demonstrate that biological relatedness mattered in the organization of the necropolis, and that whatever combination of social principles organized biological reproduction in this group left behind a strongly patrilineal pedigree structure. Some indicative elements that can be inferred from our data—female exogamy, monogamous reproductive partnering, emphasis on sublineage productive/reproductive units—are suggestive of specific kinship and union practices. Nonetheless, these elements do not preclude the existence of other social conventions that contributed to complex kinship organization, which we cannot access with genetic data. Moreover, we must keep in mind that our dataset represents a selection of individuals gathered in a funerary site, which might not be the reflection of the world of the living, if specific rules govern the access to this necropolis.

The contemporaneous monumental sites in the regional context were clearly built for selected individuals[10]. This has been demonstrated at Fleury-sur-Orne (Normandy), the only monumental site from the Cerny area genetically investigated to date, which shows a strong social selection of individuals of different patrilineal lineages, each buried in a separate monument[33]. By contrast, the absence of potential selection of individuals on the basis of sex, age, economic or social hierarchies given the available archaeological context, or consequences of specific funerary practices in Gurgy, leads us to speculate that the site represented the burial practices of the non-elite, which gives an unprecedented view of the microdemography of a non-elite community represented over several generations (Supplementary Note 14 and Supplementary Table 19). As the site was used by a single group composed of two distinguishable pedigrees, we would expect contemporaneous graveyards of similar sizes to represent lineage groups as well, but the number of such sites in the area is much smaller than we would expect for a representative cross-section of the population. Moreover, Gurgy does not show a clear cultural attribution to the Cerny horizon, although the site is contemporaneous with Cerny sites located nearby (Supplementary Fig. 1). Given the diverse cultural influences visible at Gurgy (Supplementary Note 1), the representativeness of the site and its social practices in the local context can be questioned. However, the archaeological hypothesis that Gurgy formed an isolated community stands in contrast to the genetic evidence of numerous links with a wider biological network over several generations (Supplementary Note 14).

This large pedigree reconstructed from ancient DNA represents a major and substantial step forwards in the understanding of the social organization of the human group from Gurgy 'les Noisats' and Middle Neolithic societies of Western Europe in general. What remains to be determined is whether our findings are a unique constellation among the Neolithic societies in which the variety of different funerary cultural settings is striking, or whether Gurgy represents a set of normative social structures and kinship organization during the fifth millennium BC. Thus, our research can be an anchor for further archaeogenetic studies to reach a general perspective on the potentially diverse social organization(s) of the Neolithic societies in Europe.

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

## Methods

### Sample selection

A total of 110 individuals were sampled and processed for this study. We aimed to apply the most exhaustive sampling of the 128 individuals excavated from the Gurgy 'les Noisats' site between 2004 and 2007. The skeletal remains of the 18 remaining individuals were macroscopically too poorly preserved. All human remains from the site are stored at the Centre Anthropologique de Pessac, Bordeaux University, France. Petrous bones were targeted whenever possible ($n = 94$), followed by teeth ($n = 7$) and other bones (tibia ($n = 1$), femur ($n = 1$), radius ($n = 1$), scapula ($n = 1$), phalanx ($n = 1$), unspecified bones ($n = 4$)). A list of all of the information is provided in Supplementary Table 1. Samples with a laboratory ID from GRG001 to GRG060 were entirely processed at the cleanroom facilities of the Max Planck Institute for the Science of Human History, Jena, Germany. Samples with a laboratory ID from GRG061 to GRG110 were prepared and processed in the cleanroom facilities of the Laboratory PACEA, Palaeogenetic Platform, Bordeaux University, France, up to the shotgun screening step, and were then were captured in Jena (Supplementary Table 1). The exact same protocols were applied in both laboratories, for the sake of consistency.

### Sample preparation

All human remains were treated with ultraviolet light from all sides for 15 min to reduce surface DNA contamination. Petrous bones were either cut in half and powder drilled from the denser regions around the cochlea[34] or drilled from the outside. All teeth were cleaned with a low-concentration bleach solution (3%), cut along the cementum/enamel junction and powder was collected by drilling into the pulp chamber or by being completely ground to fine powder. The surface layer of bone was mechanically removed from the other bones before powder drilling.

### Ancient DNA processing

A quantity of around 50 mg of powder was used for each extraction. The detailed extraction protocol has been published at protocols.io[35]. At least one library per individual was built following a partial uracil-DNA-glycosylase (UDG) double-stranded protocol with unique index pairs[36,37]. In the specific case of GLN270B, we built two extra non-UDG single-stranded libraries[38]. We first amplified[39] and screened all indexed libraries using shotgun sequencing. In total, 5 million reads were targeted for libraries processed in Jena on the Illumina HiSeq 4000 sequencer using either a single-end (1 × 75 base pair (bp) reads) or double-end (2 × 50 bp reads) kit. For libraries processed in Bordeaux, 1 million reads were targeted on the Illumina NextSeq 500 system based at the Institut de Recherche Biomédicale des Armées in Paris using a double end (2 × 75 bp reads) kit. We used EAGER[40] to process the raw data and to select libraries with >0.1% endogenous human DNA and those showing characteristic damage of ancient DNA patterns (4 to 18%) for downstream capture (the shotgun screening results are provided in Supplementary Table 2).

### Captures

After subsequent amplification using Herculase II Fusion polymerase (Agilent), selected libraries ($n = 105$) were hybridized in-solution to oligo-nucleotide probe sets synthesized by Agilent Technologies[41] to enrich for ~1.2 million informative nuclear SNP markers (1,240,000 SNP set)[42]. An in-house capture of the complete mitogenome according to a previous study[43], modified according to ref. 44 was applied to all of the samples. A capture targeting the entire mappable region of the Y chromosome was applied to all genetically determined male individuals ($n = 57$)[45]. Finally, a capture targeting 488 genes related to the immune response (including the highly polymorphic HLA) was applied to all of the libraries[46–48]. Enriched libraries were single- or paired-end sequenced on the HiSeq 4000 sequencer in Jena (Supplementary Tables 3, 4, 6 and 12) reaching an average coverage per site of 1.3× for the 1,240,000 SNP panel, 1.3× for the Y-chromosome capture, 153× for the mitogenome and 47.3× for the immune capture.

### Read processing

After demultiplexing, raw sequencing data were processed using EAGER. This included clipping adaptors with AdapterRemoval[49], mapping with BWA (Burrows-Wheeler Aligner, mapping quality ≥30; v.0.7.12)[50] against the human reference genome hs37d5, and removing duplicate reads with the same orientation and start (and end positions for paired-end sequencing reads). After using mapDamage (v.2.0.6)[51] to observe characteristic aDNA damage patterns, we used BamUtil (https://genome.sph.umich.edu/wiki/BamUtil) to clip two bases at the ends of each read for each sample to remove residual deamination. Different libraries of the same individual were processed separately until after quality control, after which the BAM files were treated accordingly and merged per individual. Duplicate removal was repeated on the merged libraries.

### Sex determination

According to a previous report[52], we determined the genetic sex by calculating the number of reads mapping to each of the sex chromosomes with respect to the autosomes. We set a threshold of $Y$ ratio <0.05 for a female and $Y$ ratio of >0.4 for a male (Supplementary Table 1). Samples yielding a $Y$-ratio outside of our established threshold were flagged with a question mark in Supplementary Table 1.

### Authentication criteria

Samples that were covered at less than 20,000 SNPs on the 1,240,000 SNP set were excluded from further analyses. We evaluated the authenticity of the samples by observing typical patterns of deamination towards read ends. We used the ANGSD (Analysis of Next Generation Sequencing Data) package to test for heterozygosity of polymorphic sites on the X chromosome in male individuals, applying a contamination threshold of 5% that none of our samples have reached[53] (maximum = 2.41%; Supplementary Table 1). For mito-captured samples, we estimated contamination levels using ContamMix (v.1.0.10)[54] by comparing the consensus mitogenome of the ancient sample to a panel of 311 worldwide mitogenomes as a potential contamination source (Supplementary Tables 1 and 5). We equally set our threshold at 5%. Three samples slightly exceeded this threshold, among which one was excluded from downstream analysis because of low-coverage (GLN264), and both others were male individuals (GLN253 and GLN275) and had very low X contamination estimates (1.05 and 0.47%, respectively), and we therefore decided to keep them in the analysis.

### Genotyping

For genome-wide analyses on the 1,240,000 SNP set, we genotyped our .bam files using pileupCaller (https://github.com/stschiff/sequence-Tools) by randomly calling one allele per position considering the human genome to be a pseudo-haploid genome. We called the SNP genotypes according to the Affymetrix Human Origin (HO) panel (around 600,000 SNPs)[55,56] and the 1,240,000 panel[42]. The numbers of SNPs covered at least once are provided in Supplementary Table 1.

### Imputation and screening for IBD sharing

The samples were imputed using the software GLIMPSE[57] according to the standard processing steps (https://odelaneau.github.io/GLIMPSE/docs/tutorials). Each sample was imputed separately to avoid batch effects. As a reference panel for imputation, we used the phased haplotypes from the 1000 Genomes dataset (http://ftp.1000genomes.ebi.ac.uk/vol1/ftp/release/20130502/). The imputed and phased data were then used as the input into the software ancIBD[17] (v.0.2a; https://pypi.org/project/ancIBD/). As recommended, we downsampled to 1,240,000 SNPs, for which ancIBD is optimized, and then screened

all pairs of all Gurgy individuals ($n$ = 72) with at least 500,000 SNPs covered for long IBD segments, using the recommended default settings of ancIBD. We recorded summary statistics for each pair with IBD detected (Supplementary Table 10 and Extended Data Fig. 1; https://doi.org/10.5281/zenodo.7224898).

## Mitochondrial and Y chromosome analysis

To process mtDNA data, we mapped reads from mito-capture data to the revised Cambridge reference sequence[58] using the circular mapper implemented in the EAGER pipeline[40]. We called consensus sequences using Geneious (R8.1.974)[59] and used HaploGrep 2 to determine mitochondrial haplotypes[60] (Supplementary Table 5). We assigned Y-chromosome haplogroups according to a method described previously[45] using the ISOGG SNP index v.14.07 (Supplementary Table 7).

## Kinship and reconstructed trees

To determine biological relatedness, we combined two established methods designed for aDNA data: (1) READ[15] to detect first- and second-degree relatives (Supplementary Table 8); and (2) lcMLkin (v.0.5.0)[16] to differentiate between parent–offspring and siblings among first-degree relationships (Supplementary Table 9 and Extended Data Fig. 1). We did not use the third-degree estimates and further given the implicit uncertainty of methods based on summary statistics of allele sharing. We then confirmed the links between related individuals by analysing the inferred IBD segments, which are highly informative about genealogical connections (Extended Data Fig. 1).

We finally combined these estimates with age-at-death, genetic sex and uniparentally inherited markers to reconstruct the pedigrees (Supplementary Notes 2 and 3). In the case of uncertain first-degree relationships being either siblings or parent–child, we investigated the spatial distribution of pairwise mismatch rate along the chromosome using a new tool under development, BREADR[61] (v.1.0.1; https://github.com/jonotuke/BREADR; Supplementary Note 2). In cases of pairs of individuals in which several relationships of different degrees were possible, we developed a method using the binomial distribution for the pairwise mismatch rate to assign posterior probabilities for relatedness classification (Supplementary Note 2; https://doi.org/10.5281/zenodo.7224898).

## HLA haplotypes

We obtained allele calls for the HLA class I (*HLA-A*, *HLA-B*, *HLA-C*) and class II (*HLA-DPA1*, *HLA-DPB1*, *HLA-DQA1*, *HLA-DQB1*, *HLA-DRB1*) genes by applying a development version of OptiType (v.1.3.2)[62] (Supplementary Note 8 and Supplementary Tables 13 and 14).

## Phenotypic traits

We investigated genotypes of 72 SNPs associated with phenotypes of interest in all individuals[42,63], including the HIris-Plex-S SNPs to predict the pigmentation of skin, eyes and hair[64–66]. Details are provided in Supplementary Tables 15 and 16.

## Population genetic analysis

We first inferred ROH using hapROH (v.0.60)[25] to examine consanguinity and estimate the effective population size. We screened all individuals with more than 300,000 SNPs on the 1,240,000 panel covered ($n$ = 86) (Supplementary Table 11 and Extended Data Fig. 9).

We merged our new data with published ancient data to the HO panel (around 600,000 SNPs)[55,56]. On this dataset, we performed principal component analysis using smartpca (v.10210; EIGENSOFT)[67] (Supplementary Note 10 and Extended Data Fig. 9). We computed principal components from 777 present-day west Eurasians onto which ancient individuals were then projected using the options lsqproject: YES and shrinkmode: YES. We excluded individuals with less than 10,000 covered SNPs. We then merged our data with published ancient data to the 1,240,000 SNP panel[42], including 300 present-day individuals from 142 populations sequenced to high coverage[68] and used this dataset restricted to the autosomes for subsequent genome-wide analyses. Outgroup $f_3$ statistics were calculated using qp3Pop from ADMIXTOOLS[56]. To investigate the group diversity, we performed outgroup $f_3$ statistics of the form $f_3$(individual, individual; outgroup) to create a similarity matrix, which was then used to generate the heat map using the heatmap.2 function of the R package gplots[69] (Supplementary Note 10 and Extended Data Fig. 4). We used qpAdm to estimate proportions of Anatolian Neolithic and Loschbour ancestries, as well as Goyet Q2 (ADMIXTOOLS)[44,70] (Supplementary Note 10, Supplementary Table 17 and Extended Data Fig. 9). Finally, we used the method DATES (v.753)[71] (https://github.com/priyamoorjani/DATES) to leverage patterns of ancestry covariance to estimate the date of admixture between Anatolia_Neolithic and Loschbour (Supplementary Note 10 and Supplementary Table 18).

## Geospatial analysis of burials

We performed a geospatial analysis approach using ArcGIS software to check for potential statistically significant spatial associations between burials considering combined funerary/osteological data and maternal/paternal haplotype, pedigree attribution and generations[72] (Supplementary Note 12 and Supplementary Table 21).

The maps (Fig. 1b and Extended Data Fig. 6) were created using the R packages maps (v3.3.0)[73] and mapdata (v2.3.0)[74]. The map of France in Supplementary Fig. 1 was created using the Free and Open Source QGIS (v3.30) under the Sharealike license (https://creativecommons.org/licenses/by-sa/3.0/).

## Radiocarbon dating and modelling

We used 33 radiocarbon dates, of which 25 were previously published[12] and 8 are newly reported, generated at the CEDAD - CEntro di DAtazione e Diagnostica, Salento University, Lecce, Italy, with the exception of the date of GLN275, which was generated at the Centre de Datation par le RadioCarbone (CDRC), Lyon 1 University, Lyon, France (Supplementary Table 1). We calibrated the radiocarbon dates with IntCal20.14c[75] and applied Bayesian chronological modelling based on the approach of the ChronoModel software[76] (Supplementary Note 13, Supplementary Table 25 and Extended Data Fig. 10).

## Strontium isotope analysis

We performed $^{87}Sr/^{86}Sr$ using the laser ablation technique[20] on first and second molars of 57 selected individuals (Fig. 3, Supplementary Note 11, Supplementary Tables 23, 27 and 28 and Extended Data Fig. 3; https://doi.org/10.5281/zenodo.7224898).

## Reporting summary

Further information on research design is available in the Nature Portfolio Reporting Summary linked to this article.

## Data availability

New genomic sequencing data (BAM format) are available at the European Nucleotide Archive (ENA; PRJEB61818). Previously published genomic sequencing data (BAM format) are available at the ENA (PRJEB36208 and PRJEB45741). The Genome Reference Consortium Human Build 37 (GRCh37) is available at the National Center for Biotechnology Information under accession number PRJNA31257. The revised Cambridge reference sequence for the mitochondrial genome is available at the National Center for Biotechnology Information under NCBI Reference Sequence NC012920.1. Previous published genotype data for ancient individuals was reported by the Reich laboratory in the Allen Ancient DNA Resource v.50.0 (https://reich.hms.harvard.edu/allen-ancient-dna-resource-aadr-downloadable-genotypes-present-day-and-ancient-dna-data). The soil strontium values used

for comparison come from the IRHUM database[77]. Additional data are available at Zenodo (https://doi.org/10.5281/zenodo.7224898).

## Code availability

The code for analysing and visualizing the haplotype sharing (IBD and ROH) has been deposited at GitHub (https://github.com/hringbauer/ibd_gurgy.git).

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

**Acknowledgements** This study benefitted from excavation grant support from the region of Bourgogne, France and the Service Régional de l'Archéologie de Bourgogne (site number 89 198 008 issued by SRA authority). This research was funded by a ministerial grant from the French Research Foundation as a program of prospects investments ANR-10-LABX-52 (to S.R., project DHP, LaScArBx-ANR, 2012-14); Fyssen Foundation Postdoctoral Stipend (to M.R., 2017-18); the Max Planck Society, by the French Research Foundation and German Research Foundation (to M.R., W.H. and M.-F.D., project INTERACT, ANR-17-FRAL-0010 and DFG-HA-5407/4-1, 2018-21; to A.S., project ImMiGeNe, DFG-433115696); and the European Research Council under the European Union's Horizon 2020 Research and Innovation Programme (PALEoRIDER, 771234, 2018-23, to W.H.). This research benefited from the scientific framework of the University of Bordeaux's IdEx 'Investments for the Future' program/GPR 'Human Past'. The isotopic analysis of this study was made possible by the INSU/CNRS MC-ICP-MS national facility at ENS-Lyon. We thank the staff at Elemental Scientific Lasers for providing a laser and technical support as part of the formal collaboration with ENS-Lyon. This project has received financial support from the CNRS through the MITI interdisciplinary programs. We thank G. Brandt, K. Nägele, R. Radzeviciute, R. Stahl and A. Wissgott for technical support in the DNA analyses; O. Gorgé at the Institut de Recherche Biomédicale des Armées for processing samples for next-generation sequencing; the staff at Pacha Cartographie for the rights to reproduce the map in Supplementary Fig. 1; the members of the teams at the Max Planck Institute for Evolutionary Anthropology and the laboratory PACEA (De la Préhistoire à l'Actuel, Culture, Environnement, Anthropologie), University of Bordeaux, for continued support and discussion, specifically, A. Arzelier, A. Ghalichi, S. Kacki, M. Le Luyer, D. López Onaindia, L. Maréchal, M. Pruvost, R. Risch, S. Schiffels, C. Schmid and V. Villalba-Mouco; and C. Fowler, K. Sirak and A. Williams for their comments.

**Author contributions** M.R., S.R., M.-F.D. and W.H. conceived the study. S.R. provided samples and archaeological contextualization. M.R., F.A., M.-H.P. and E.S. performed the laboratory work. M.R., A.B.R., H.R., A.C., F.M. and M.L.R. performed data analysis, with M.-F.D. and W.H. providing guidance and methodologies. A.B.R. and J.T. developed new analytical tools. R.B and A.S. performed HLA analysis. L.R., P.T., G.G. and A.S. performed radiogenic analysis. H.C. guided the social anthropological interpretation. J.K. provided access to resources and methodology. M.R. and W.H. wrote the paper with input from all of the authors.

**Funding** Open access funding provided by Max Planck Society.

**Competing interests** The authors declare no competing interests.

**Additional information**
**Correspondence and requests for materials** should be addressed to Maïté Rivollat, Stéphane Rottier or Wolfgang Haak.

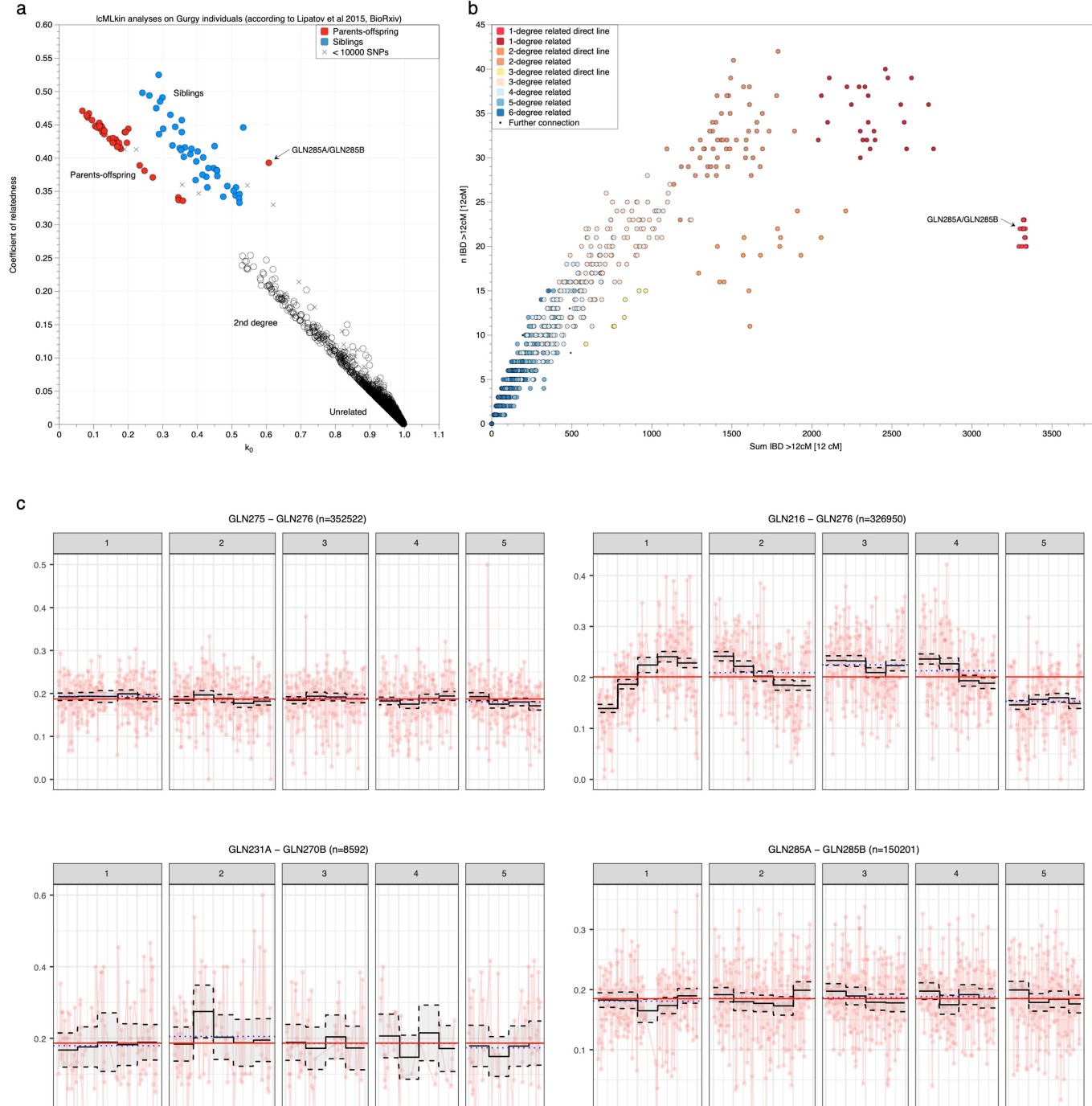

**Extended Data Fig. 1 | Biological relatedness analysis of the Gurgy individuals. a**, lcMLkin estimates of k0 and the coefficient of relatedness r. Clusters of different first-degree related individuals emerge when plotting these measures of relatedness against each other. Colours are given a posteriori according to the reconstructed trees. One inconsistency can be observed with the red point (parent-offspring relationship) plotting in the sibling cluster in blue, representing the pair GLN285A and GLN285B (Supplementary Note 2 and 3, Supplementary Table 9). **b**, Identity-by-descent (IBD) sharing analysis. Pairs of individuals plotted according to shared long blocks of IBD (>12cM), clustering by degrees of relatedness. Direct and indirect lineages form two different clines, indicating different numbers of

meiosis events. From the fourth degree of relatedness, these clusters overlap (Supplementary Note 3, Supplementary Table 10). The pair GLN285A and GLN285B falls in the parent-offspring cluster. **c**, PMR-window plots. Pairwise-mismatch Rates (PMR) plotted along the chromosomes 1 to 5 in windows of 1 Megabase width. The number of overlapping SNPs is given in brackets. For example, the windowed estimate of PMR (dark line) is stable around the average PMR (red line) for the parent-offspring relationship GLN275-GLN276, whereas it is more variable for the full sibling relationship GLN216-GLN276. The pairs GLN231A-GLN270B and GLN285A-GLN285B are discussed in Supplementary Note 2.

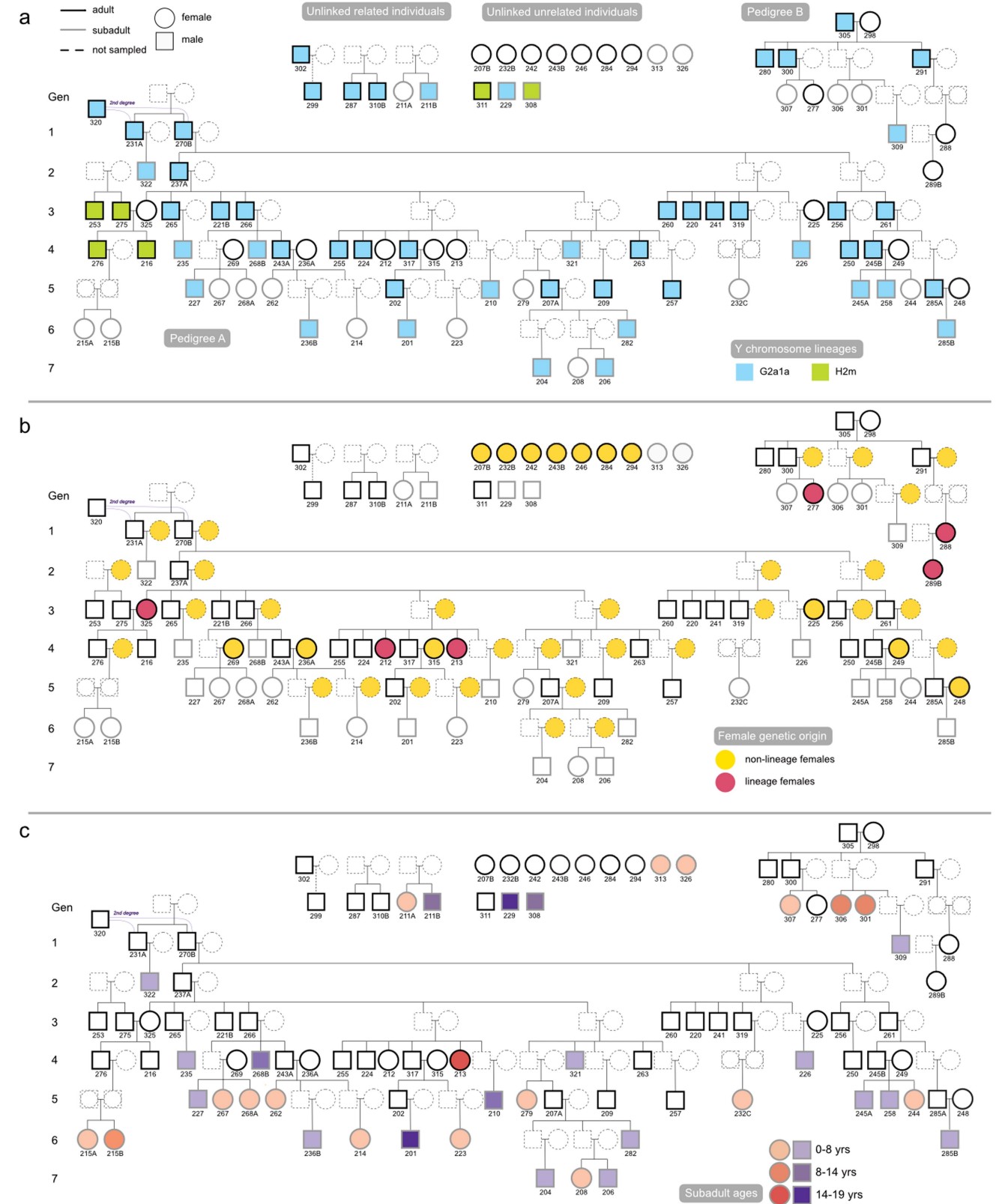

**Extended Data Fig. 2 | Y chromosome haplogroups, exogenous females, and age at death of subadults visualized on the Gurgy pedigrees.**
**a**, Y chromosome haplogroup distribution along the lineages. Male individuals of Gurgy carry only two Y haplogroups: G2a1a (Z38302) and H2m (P96), which illustrates the strong patrilineal pattern of the pedigrees (Supplementary Table 7). **b**, Exogenous females, i.e., all females with no parents buried at the site, including unsampled females, are highlighted in yellow. **c**, Distribution of age at death classes of subadult across the pedigrees (Supplementary Table 1).

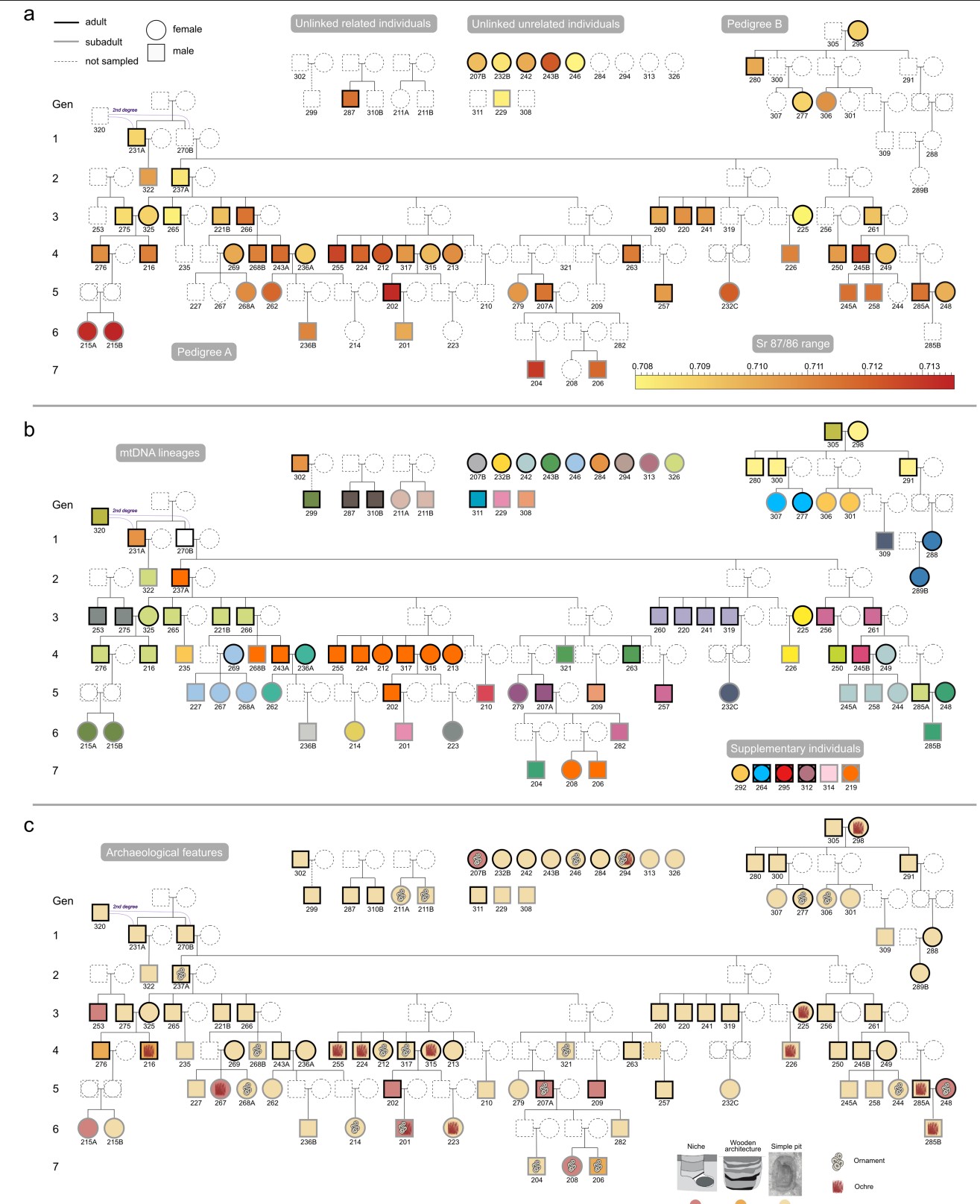

**Extended Data Fig. 3 | Strontium ratio, mitochondrial haplogroup diversity, and archaeological features visualized on the Gurgy pedigrees. a**, Strontium isotope ratio. The first generations and exogenous females carry lower values, which indicate non-local origins (Supplementary Note 11, Supplementary Table 23). **b**, Mitochondrial diversity. Each colour represents a different mitochondrial haplogroup. (Supplementary Note 3, Supplementary Table 5). **c**, Visualization of select archaeological features on the reconstructed pedigrees. The different types of pits, individuals buried with ornaments and ochre are represented. No clustering according to biological relatedness is visible (Supplementary Table 1).

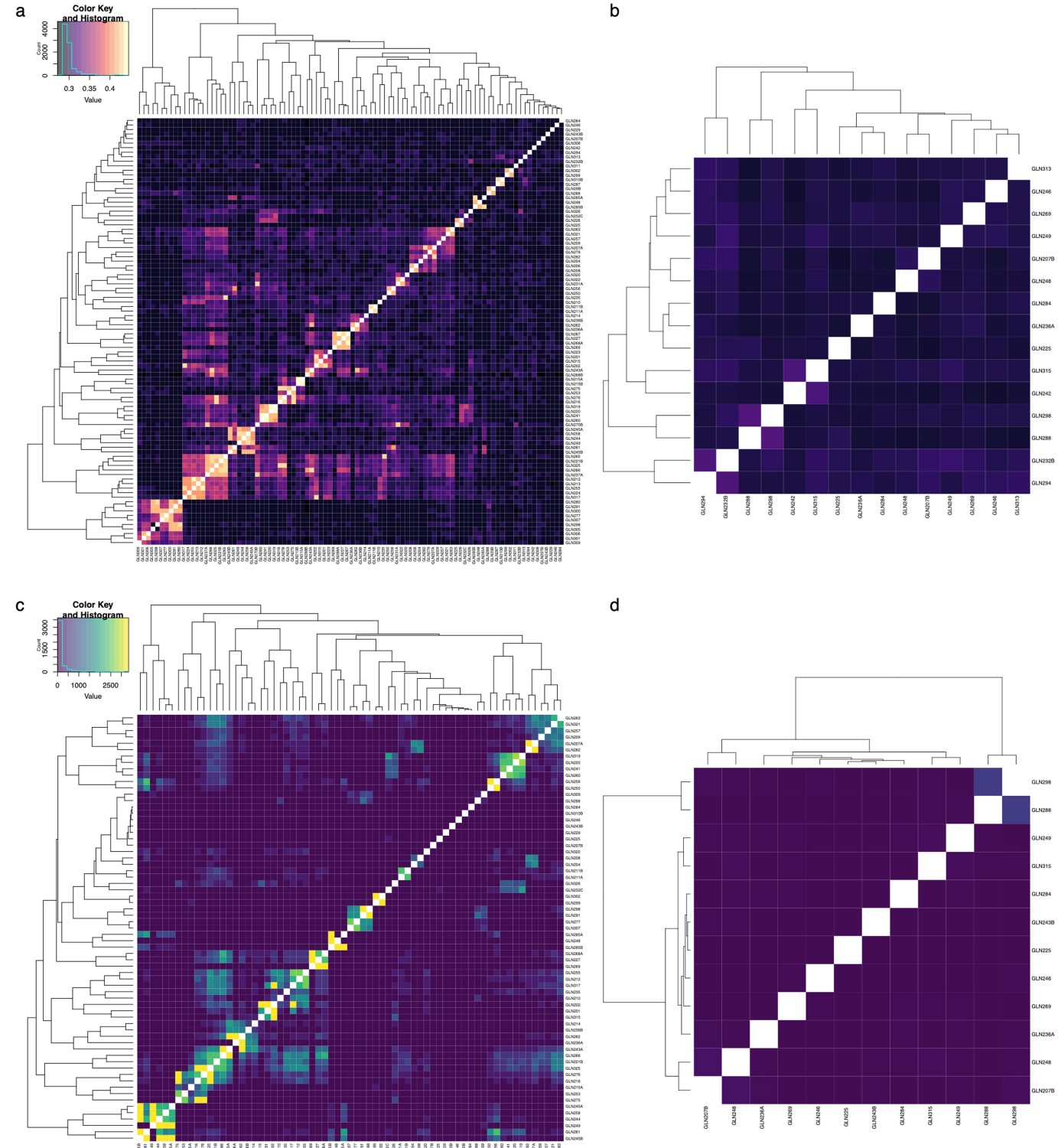

**Extended Data Fig. 4 | Heatmaps of biological relatedness. a**, Heatmap showing pairwise outgroup $f_3$ statistics between all individuals from Gurgy (n = 94). Lighter colours indicate higher shared genetic affinity between individuals. Each sub-lineage forms a cluster, and the two bigger pedigrees are also visible (Supplementary Note 5, Supplementary Table 12). **b**, Heatmap showing pairwise outgroup $f_3$ statistics between all exogenous female individuals (n = 16). Three pairs show distant relatedness (GLN232B-GLN294, GLN288-GLN298, GLN242-GLN315), but are otherwise unrelated (Supplementary Note 5, Supplementary Table 12). **c**, Heatmap showing pairwise IBD sharing between individuals with more than 500,000 SNPs (n = 72). Different clusters of lighter colour reveal the extra-links between the different branches and pedigrees (Supplementary Notes 3 and 5, Supplementary Table 10). **d**, Heatmap showing pairwise IBD sharing between all exogenous female individuals with more than 500,000 SNPs (n = 12). The absence of relatedness between the exogenous female individuals is confirmed by the deeper resolution provided by IBD sharing analysis, with a single pair of females related in the third degree, as detected by the $f_3$ statistics (Supplementary Notes 3 and 5, Supplementary Table 10).

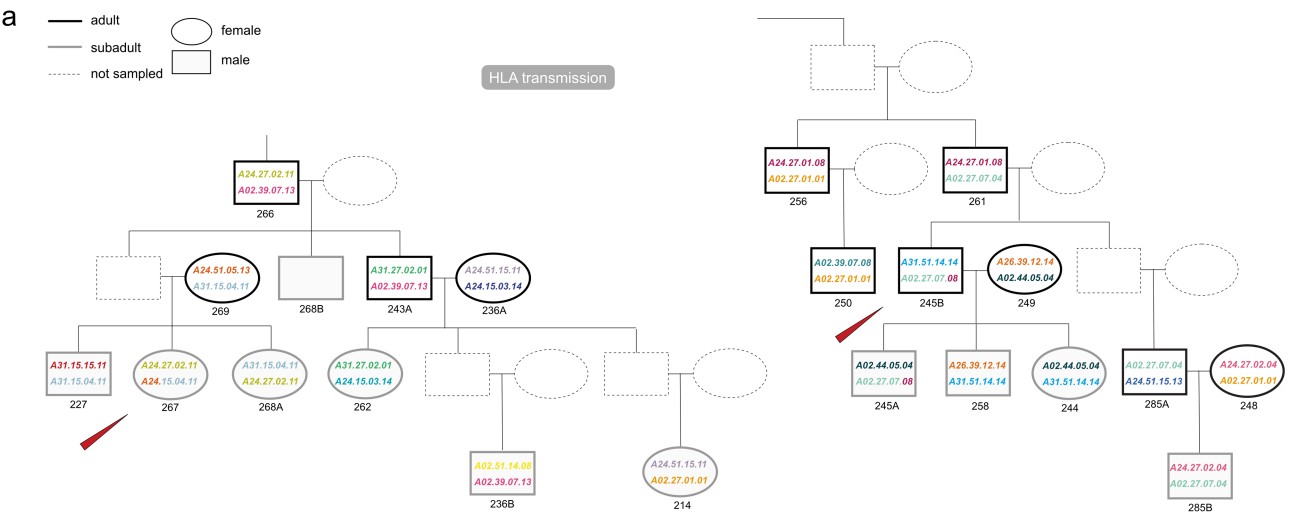

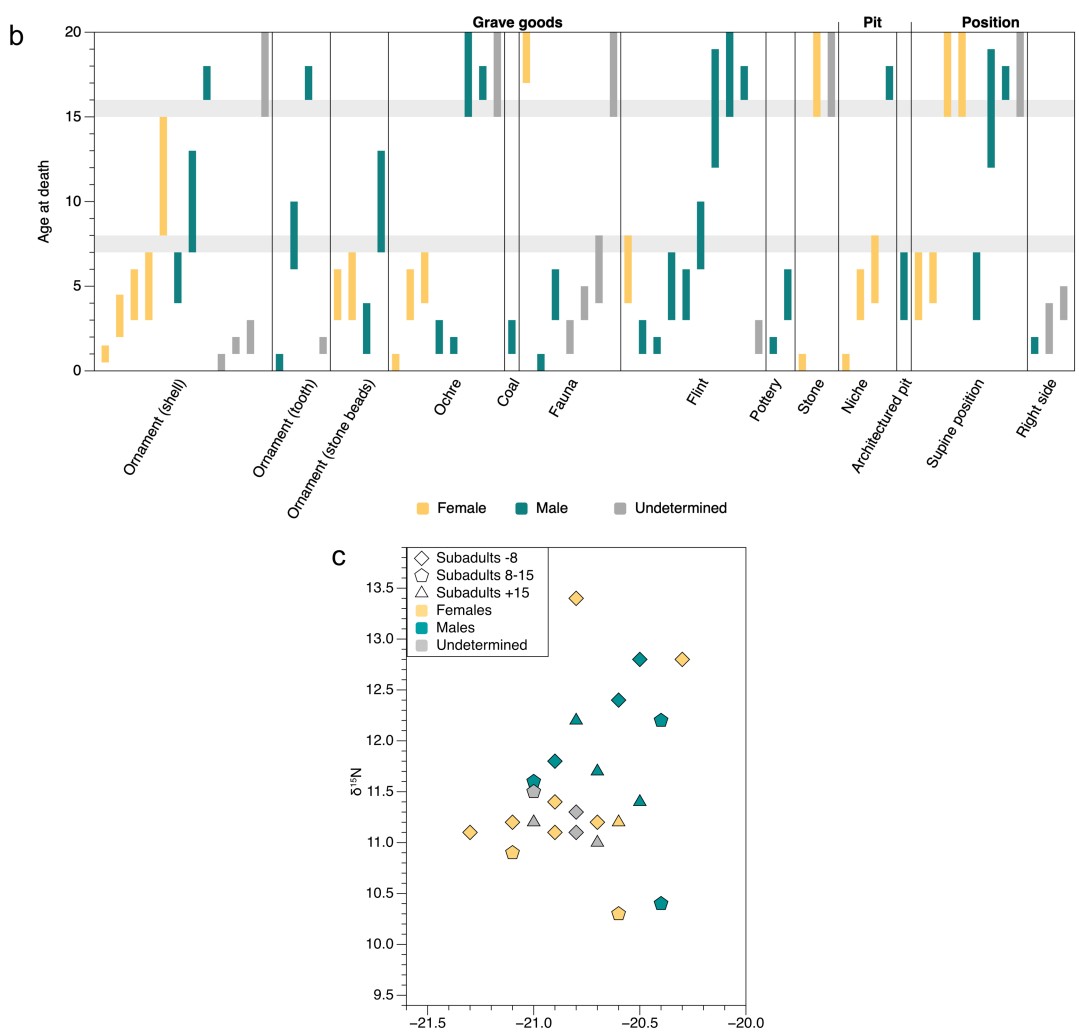

**Extended Data Fig. 5** | See next page for caption.

**Extended Data Fig. 5 | HLA recombination, and dietary isotopes of the Gurgy individuals. a**, Zoom in on two events of recombination of HLA haplotypes, one of class I in the individual GLN267, where both haplotypes carried by the mother recombined, and one in class II in the individual GLN245B where both haplotypes carried by the father recombined and transmitted to the son GLN245A (Supplementary Table 14). **b**, Distribution of specific archaeological features according to the age at death of all subadult individuals (irrespective of DNA preservation) and their sex. Each bar represents one individual for each feature, that is some individuals are represented several times. Grey lines represent the two observed thresholds before and after individuals tend to be buried with grave goods or with specific archaeological features. **c**, Stable isotope data (carbon and nitrogen) (Rey et al. 2019), plotted with the genetic sexing of subadult individuals provided by this study. Two significant clusters can be observed between male and female individuals (two-sided permutation test, p = 0.01019; Supplementary Note 14.1). The high values of $\delta^{15}$N of three outlier individuals (GLN232C, GLN326, GLN245A, all younger than 6 years of age) might be a signal for breastfeeding or weaning time.

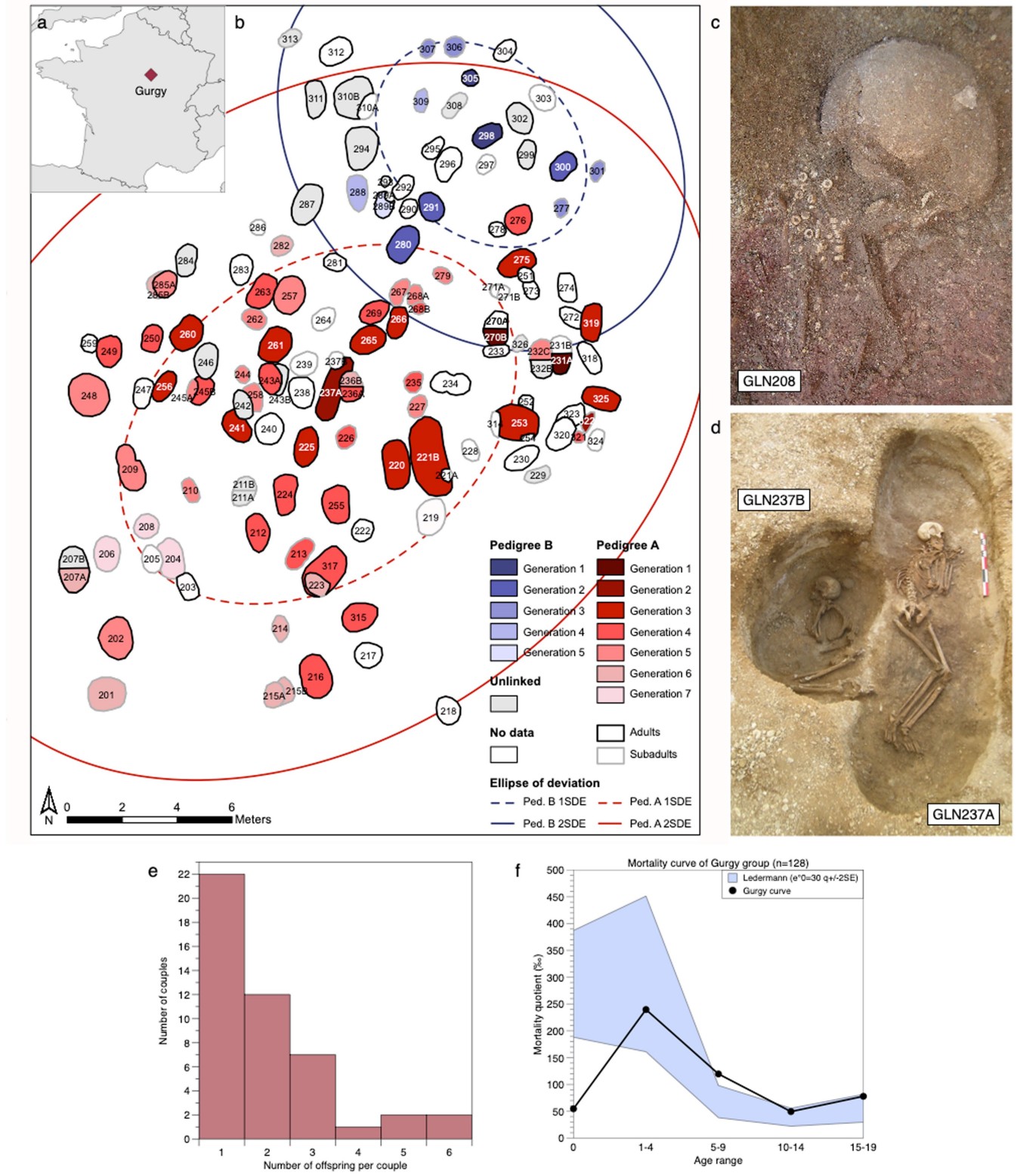

**Extended Data Fig. 6 | Archaeological features, spatial and demographic distribution of Gurgy individuals. a**, Geographical location of the Gurgy 'les Noisats' site in present-day France. Map created with R packages *maps* (v3.3.0; Becker et al. 2018) and *mapdata* (v2.3.0; Becker et al. 2018). **b**, Spatial layout of the site with burials visualized per generation. For both pedigrees, the expansion of the graveyard followed in the direction of North-East to South-West. Ellipses indicate one and two standard deviations (Supplementary Note 12). **c**, Picture of GLN208 buried with limestone beads. **d**, Picture of GLN237A and GLN237B. GLN237A has one of the two largest graves of the site. **e**, Histogram of the number of offspring per couple. Minimum estimation of number of offspring per couple necessary to explain the pedigrees. **f**, Comparison of the mortality curve of Gurgy individuals calculated on subadults (Supplementary Table 19) with an expected wide pattern of archaic mortality (Ledermann 1969). We notice a deficit of infants among the Gurgy cohort (Supplementary Note 14.2, Supplementary Table 19).

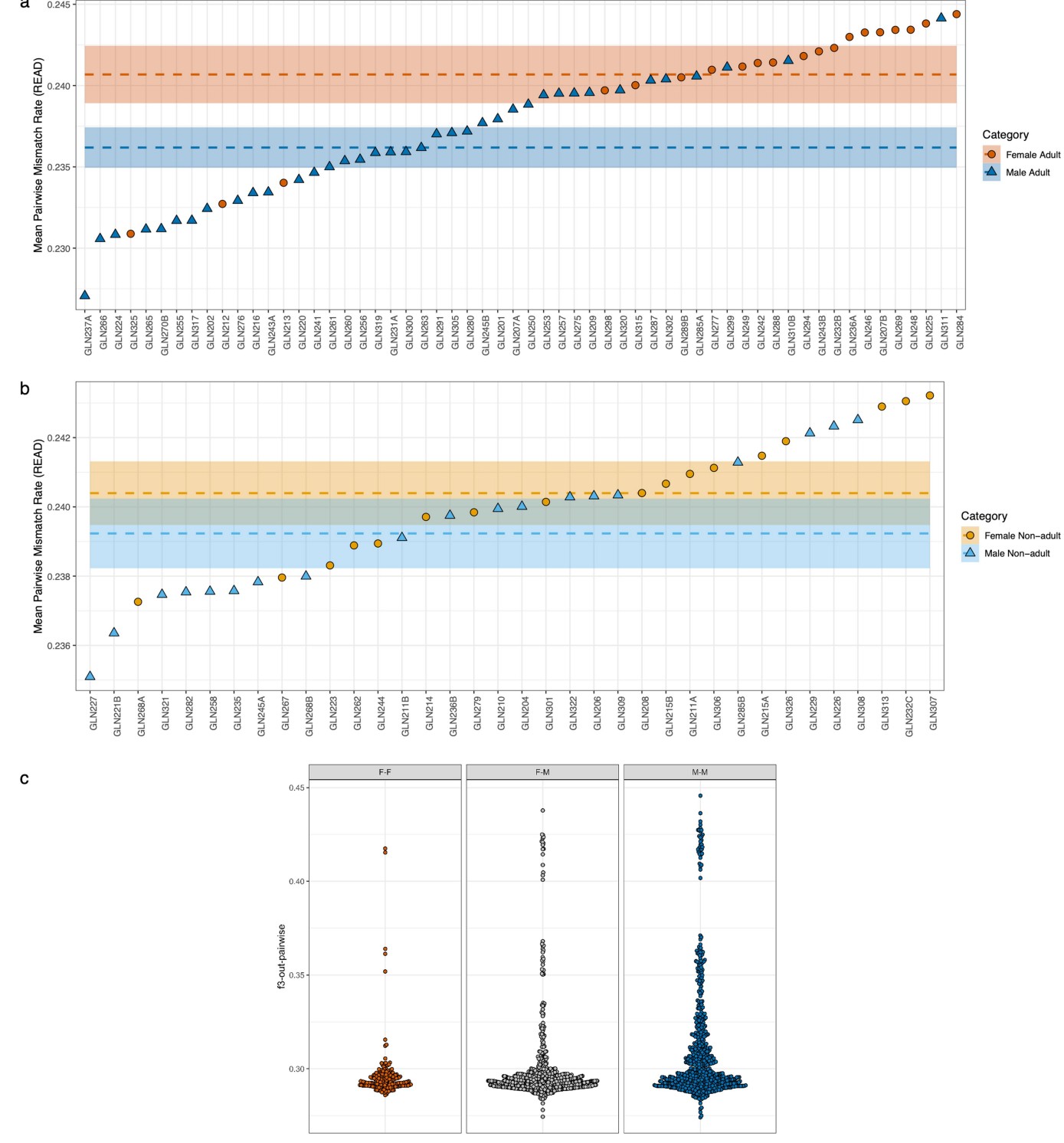

**Extended Data Fig. 7 | Trends of relatedness between female-male cohorts at Gurgy. a**, Average of the pairwise P0 values, obtained via READ, for all adult individuals. The lower the P0 value, the more related the individual is to the group, on average. Dashed lines show the average P0 values for each sex and 95% CIs are given as coloured bands. A two-sided Wilcoxon test shows that male individuals are significantly ($P = 1.375e-05$) more related to the group than female individuals (Supplementary Note 2, Supplementary Table 8). **b**, Average P0 for all subadult male and female individuals are not significantly differentially related to the group (two-sided Wilcoxon, p = 0.1067) (Supplementary Note 2, Supplementary Table 8). **c**, $f_3$-outgroup statistics of the form $f_3$(female, female; Mbuti), $f_3$(female, male; Mbuti), and $f_3$(male, male; Mbuti) for all adults. The deeper relatedness among pairs, whenever they involve at least one male individual, is, on average, higher than among female individuals (first- and second-degree related pairs are excluded from the calculation; Supplementary Table 12).

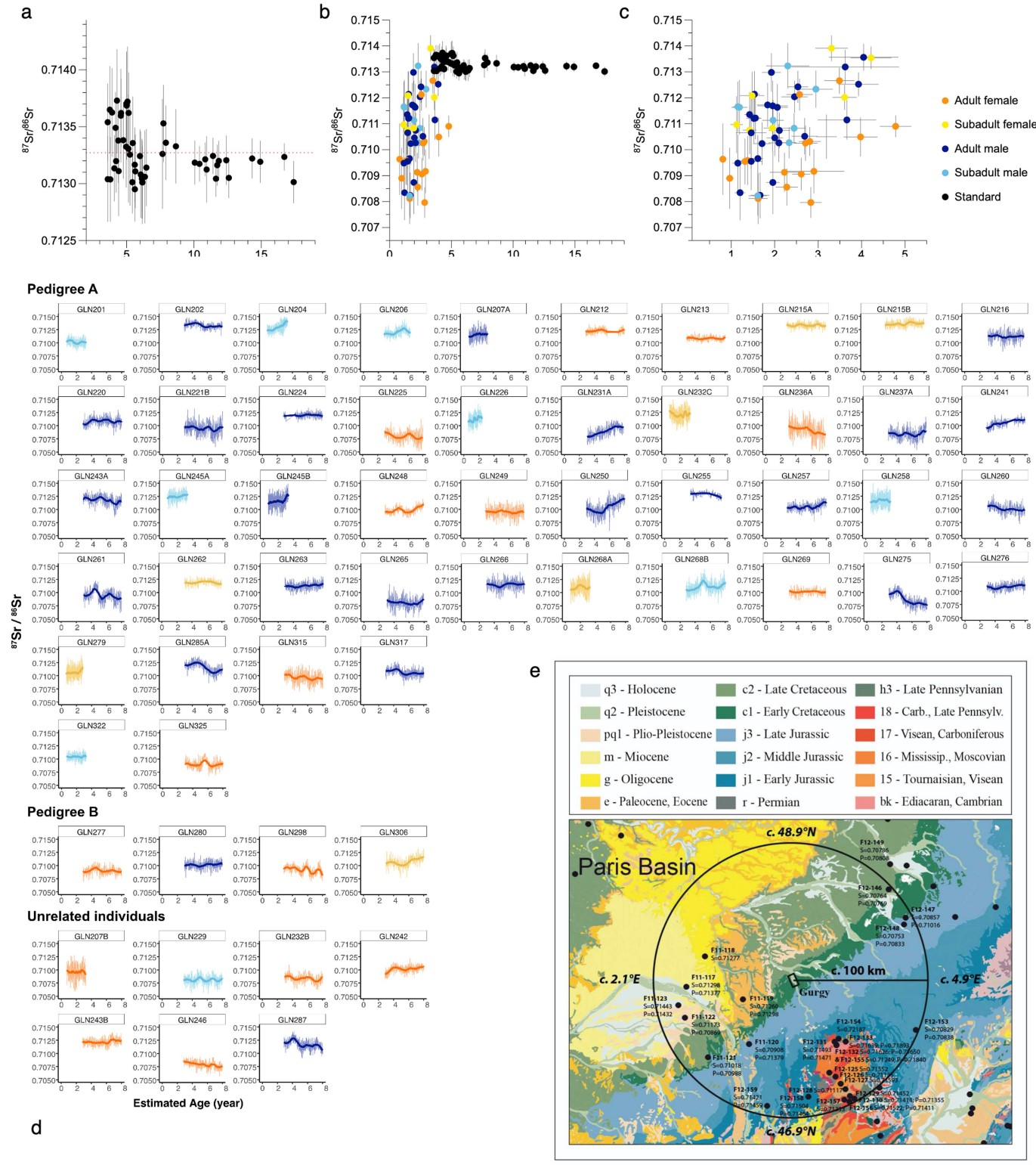

**Extended Data Fig. 8 | Results of radiogenic Sr isotope analyses.**
**a**, Distribution of the standards (n = 52) average $^{88}$Sr and $^{87}$Sr/$^{86}$Sr values and SD used to bracket the individuals $^{87}$Sr/$^{86}$Sr values. The accepted $^{87}$Sr/$^{86}$Sr value of 0.71310 for SRM-1400 is indicated by the dashed red line (Supplementary Table 22). **b**, Distribution of the standards (n = 52) and samples (n = 57) average $^{88}$Sr and $^{87}$Sr/$^{86}$Sr values and SD (Supplementary Table 22). **c**, Distribution of the samples (n = 57) average $^{88}$Sr and $^{87}$Sr/$^{86}$Sr values and SD (Supplementary

Table 22). **d**, $^{87}$Sr/$^{86}$Sr profiles of all individuals analysed in this study (n = 57). Each curve is plotted as a function of the reconstructed dental age, according to the sampled tooth (M1 or M2; https://doi.org/10.5281/zenodo.7224898). **e**, Geological map of the area surrounding Gurgy, as reconstructed by BRGM (https://infoterre.brgm.fr), with environmental $^{87}$Sr/$^{86}$Sr ratios from the IRHUM database (Willmes et al. 2013). 'P' stands for 'plants' and 'S' for 'soil'.

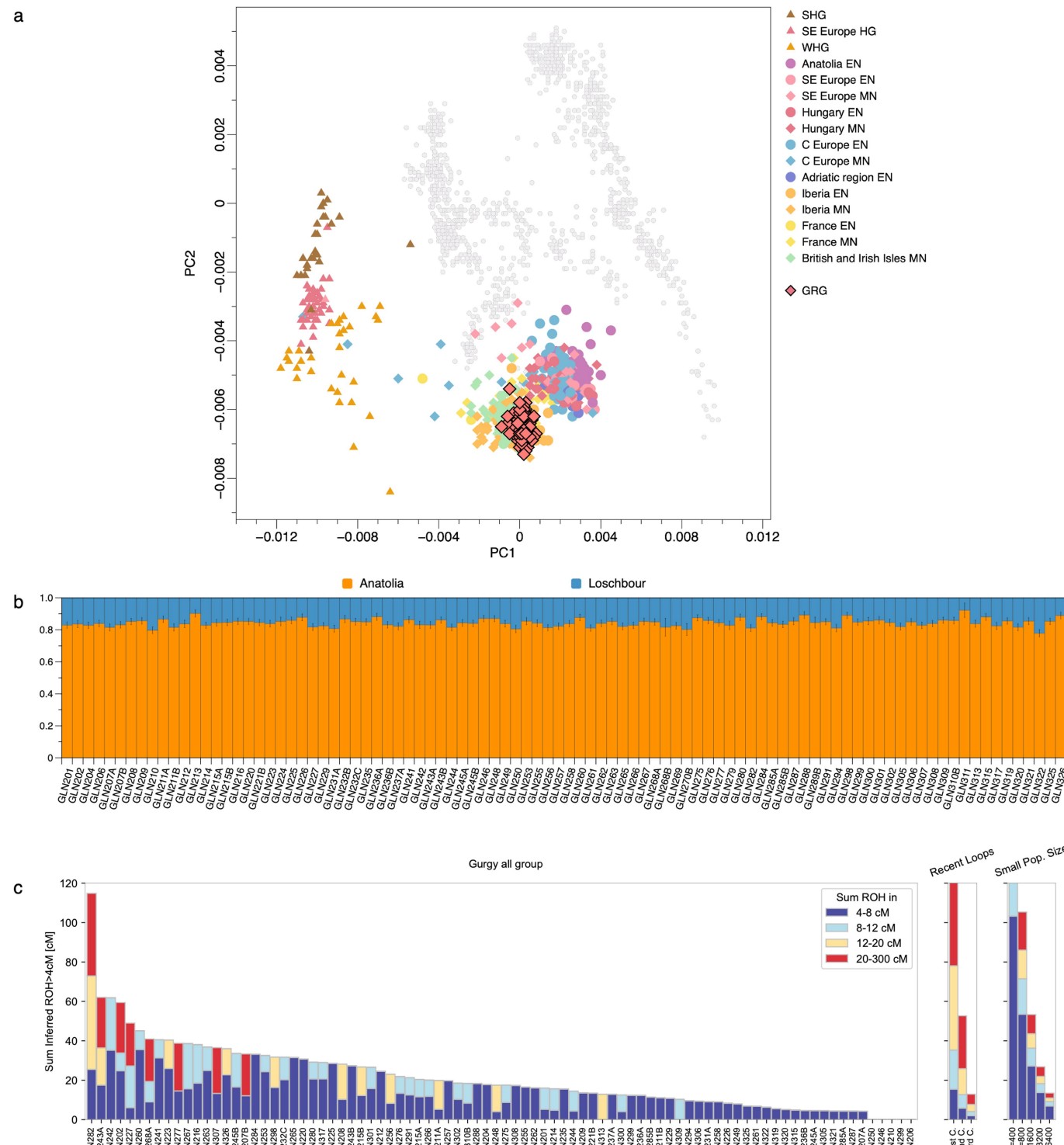

**Extended Data Fig. 9 | Population genetic analyses of Gurgy individuals.**
**a**, Principal component analysis. Published ancient (symbols with no outline)
and Gurgy (black outlined symbols) individuals projected onto 777 present-day
west Eurasians (grey circles; Supplementary Note 10). **b**, Genetic ancestry
proportions. Results of qpAdm (MODEL 1) ancestry modelling of Loschbour
hunter-gatherer and Anatolia_Neolithic ancestry for all Gurgy individuals
(Supplementary Note 10). **c**, Runs of homozygosity. Selected individuals with

more than 300,000 SNPs on the 1.2 million SNP panel (n = 86) and simulated
data expected for inbred individuals from parents related at the first- to
third-degree, and for individuals from small populations with different sizes.
Individual GLN282 shows an inbreeding signal similar to a first-cousin union,
but both carried ROH are 20–22 cM long, therefore this individual is more
plausibly the offspring of second or third cousins (Supplementary Note 4,
Supplementary Table 1).

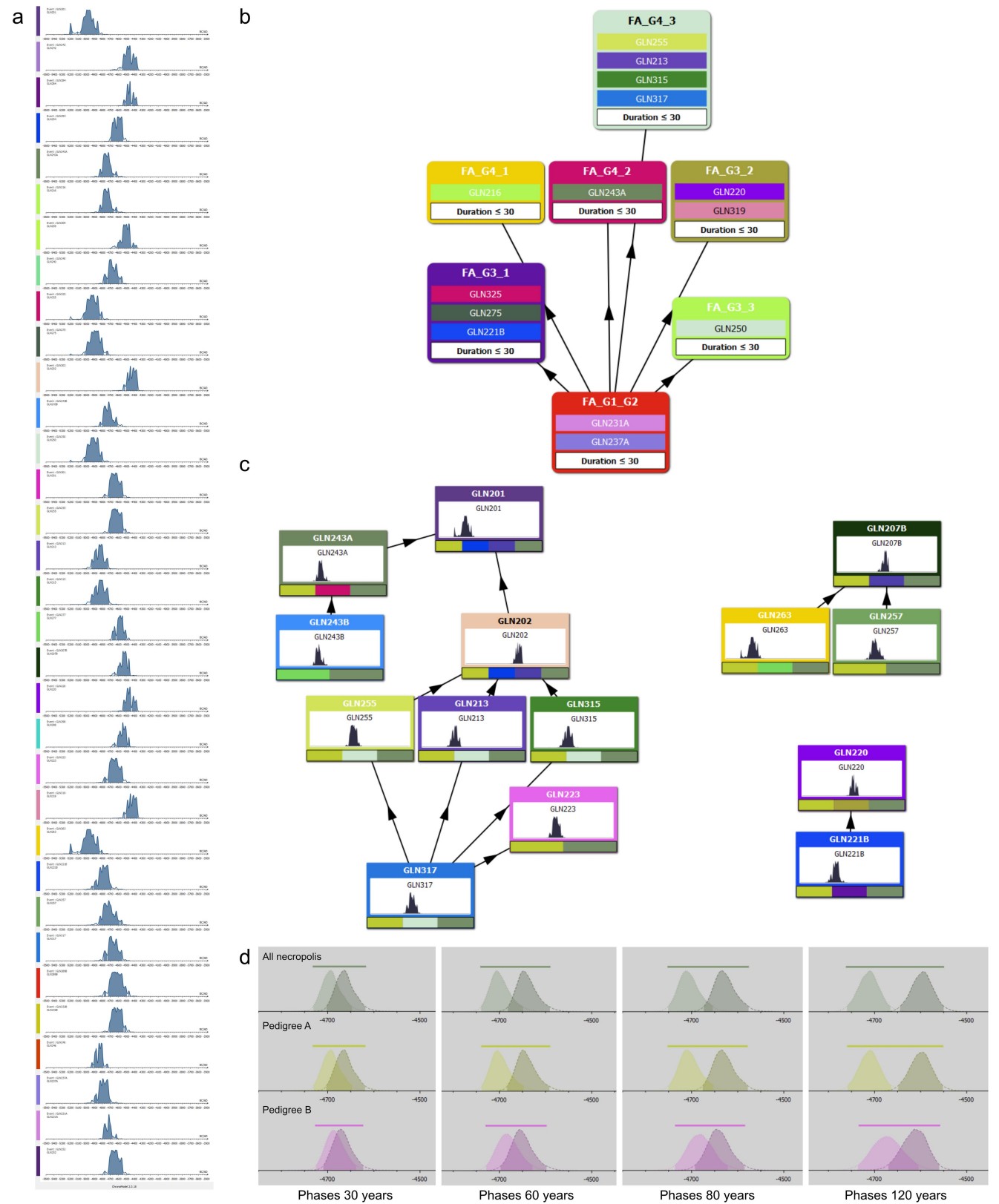

**Extended Data Fig. 10 | Bayesian modelling of radiocarbon dates.**
**a**, Radiocarbon dates available for Gurgy, calibrated with Intcal20.14c.
**b**, Bayesian modelling with ChronoModel 2.0.18. Model with constrained relationships inferred from the pedigrees (Supplementary Note 13, Supplementary Table 25). F = Pedigree and G = Generation. **c**, Model with constrained relationships inferred from archaeology (Supplementary Note 13, Supplementary Table 25). **d**, Modelled intervals for each run set at 30, 60, 80 and 120 years for Pedigrees A and B, and for the whole group (111,000 iterations for 3 independent chains per run, Supplementary Note 13, Supplementary Table 25).

Stephane Rottier
Wolfgang Haak

# Reporting Summary

## Statistics

For all statistical analyses, confirm that the following items are present in the figure legend, table legend, main text, or Methods section.

| n/a | Confirmed | |
|---|---|---|
| ☐ | ☒ | The exact sample size (*n*) for each experimental group/condition, given as a discrete number and unit of measurement |
| ☐ | ☒ | A statement on whether measurements were taken from distinct samples or whether the same sample was measured repeatedly |
| ☐ | ☒ | The statistical test(s) used AND whether they are one- or two-sided *Only common tests should be described solely by name; describe more complex techniques in the Methods section.* |
| ☒ | ☐ | A description of all covariates tested |
| ☒ | ☐ | A description of any assumptions or corrections, such as tests of normality and adjustment for multiple comparisons |
| ☐ | ☒ | A full description of the statistical parameters including central tendency (e.g. means) or other basic estimates (e.g. regression coefficient) AND variation (e.g. standard deviation) or associated estimates of uncertainty (e.g. confidence intervals) |
| ☐ | ☒ | For null hypothesis testing, the test statistic (e.g. *F*, *t*, *r*) with confidence intervals, effect sizes, degrees of freedom and *P* value noted *Give P values as exact values whenever suitable.* |
| ☐ | ☒ | For Bayesian analysis, information on the choice of priors and Markov chain Monte Carlo settings |
| ☒ | ☐ | For hierarchical and complex designs, identification of the appropriate level for tests and full reporting of outcomes |
| ☒ | ☐ | Estimates of effect sizes (e.g. Cohen's *d*, Pearson's *r*), indicating how they were calculated |

*Our web collection on statistics for biologists contains articles on many of the points above.*

## Software and code

Policy information about availability of computer code

| Data collection | No specific software was used for data collection. The processing steps from raw Illumina sequence data to genotype data involved the following software: fastqc (0.11.4), EAGER v1 (1.92.56), AdapterRemoval (v2.3.0), BWA (v 0.7.12), DeDup (v 0.12.1), MapDamage (v 2.0.6), samtools (v 1.3.1), pileupCaller (v 1.4.0.2), and bamUtils (v 1.0.13). |
|---|---|
| Data analysis | ANGSD (0.910) ContamMix (v1.0.10) ADMIXTOOLS (5.1) (qp3Pop, qpDstats, qpAdm) Haplogrep 2 (v2.4.0) Picard tools (v2.27.3) GLIMPSE (v1.1.1) (https://odelaneau.github.io/GLIMPSE/index.html) AncIBD (v0.2a) (https://pypi.org/project/ancIBD/) hapROH (v0.60) Geneious (R8.1.974) READ (v.3) (https://bitbucket.org/tguenther/read) lcMLkin (v0.5.0) (https://github.com/COMBINE-lab/maximum-likelihood-relatedness-estimation) BREADR (v.1.0.1) (https://github.com/jonotuke/BREADR) smartpca (v10210; EIGENSOFT v6.0.1) qpAdm (v.810) DATES (v 753) ISOGG SNP index (v.14.07) |

For manuscripts utilizing custom algorithms or software that are central to the research but not yet described in published literature, software must be made available to editors and reviewers. We strongly encourage code deposition in a community repository (e.g. GitHub). See the Nature Portfolio guidelines for submitting code & software for further information.

## Data

Policy information about availability of data

All manuscripts must include a data availability statement. This statement should provide the following information, where applicable:

- Accession codes, unique identifiers, or web links for publicly available datasets
- A description of any restrictions on data availability
- For clinical datasets or third party data, please ensure that the statement adheres to our policy

New genomic sequence data (BAM format) is available at the European Nucleotide Archive (ENA) under project accession number PRJEB61818. Previously published genomic sequence data (BAM format) is available at the ENA under the numbers PRJEB36208 and PRJEB45741.
The Genome Reference Consortium Human Build 37 (GRCh37) is available via the National Center for Biotechnology Information under accession number PRJNA31257. The revised Cambridge reference sequence for the mitochondrial genome is available via the National Center for Biotechnology Information under NCBI Reference Sequence NC 012920.1. Previous published genotype data for ancient individuals was reported by the Reich Lab in the Allen Ancient DNA Resource v50.0 (https://reich.hms.harvard.edu/allen-ancient-dna-resource-aadr-downloadable-genotypes-present-day-and-ancient-dna-data).

## Human research participants

Policy information about studies involving human research participants and Sex and Gender in Research.

| Reporting on sex and gender | Use the terms sex (biological attribute) and gender (shaped by social and cultural circumstances) carefully in order to avoid confusing both terms. Indicate if findings apply to only one sex or gender; describe whether sex and gender were considered in study design whether sex and/or gender was determined based on self-reporting or assigned and methods used. Provide in the source data disaggregated sex and gender data where this information has been collected, and consent has been obtained for sharing of individual-level data; provide overall numbers in this Reporting Summary.  Please state if this information has not been collected. Report sex- and gender-based analyses where performed, justify reasons for lack of sex- and gender-based analysis. |
|---|---|
| Population characteristics | Describe the covariate-relevant population characteristics of the human research participants (e.g. age, genotypic information, past and current diagnosis and treatment categories). If you filled out the behavioural & social sciences study design questions and have nothing to add here, write "See above." |
| Recruitment | Describe how participants were recruited. Outline any potential self-selection bias or other biases that may be present and how these are likely to impact results. |
| Ethics oversight | Identify the organization(s) that approved the study protocol. |

Note that full information on the approval of the study protocol must also be provided in the manuscript.

# Field-specific reporting

Please select the one below that is the best fit for your research. If you are not sure, read the appropriate sections before making your selection.

☒ Life sciences ☐ Behavioural & social sciences ☐ Ecological, evolutionary & environmental sciences

For a reference copy of the document with all sections, see nature.com/documents/nr-reporting-summary-flat.pdf

# Life sciences study design

All studies must disclose on these points even when the disclosure is negative.

| Sample size | No statistical methods were used to determine ancient DNA sample size a priori. Sample sizes for ancient populations depended solely on the availability of the archaeological human remains and on ancient DNA preservation. |
|---|---|

| | |
|---|---|
| Data exclusions | Data from specimens that showed insufficient levels of ancient DNA content or high levels of DNA contamination were excluded from further analyses. |
| Replication | We studied unique individuals form past populations and did not perform different experiments, so replication is not applicable. |
| Randomization | We studied unique individuals form past populations and did not perform different experiments, so randomization is not applicable. |
| Blinding | We studied unique individuals form past populations and did not perform different experiments, so blinding is not applicable. |

# Reporting for specific materials, systems and methods

We require information from authors about some types of materials, experimental systems and methods used in many studies. Here, indicate whether each material, system or method listed is relevant to your study. If you are not sure if a list item applies to your research, read the appropriate section before selecting a response.

## Materials & experimental systems

| n/a | Involved in the study |
|---|---|
| ☒ ☐ | Antibodies |
| ☒ ☐ | Eukaryotic cell lines |
| ☐ ☒ | Palaeontology and archaeology |
| ☒ ☐ | Animals and other organisms |
| ☒ ☐ | Clinical data |
| ☒ ☐ | Dual use research of concern |

## Methods

| n/a | Involved in the study |
|---|---|
| ☒ ☐ | ChIP-seq |
| ☒ ☐ | Flow cytometry |
| ☒ ☐ | MRI-based neuroimaging |

# Palaeontology and Archaeology

| | |
|---|---|
| Specimen provenance | Specimens come from an excavation supported by the region of Bourgogne, France, and the Service Régional de l'Archéologie de Bourgogne, and were sampled with permission from the Service Régional de l'Archéologie de Bourgogne. |
| Specimen deposition | The skeletal remains are currently stored at the ostéothèque of PACEA, Pessac, France. Further sampling will require permission from the region of Bourgogne, France, and the Service Régional de l'Archéologie de Bourgogne. |
| Dating methods | New AMS 14C dates were obtained from ultra-filtrated collagen. Collagen extraction and 14C measurements were carried out at the CEDAD - CEntro di DAtazione e Diagnostica, Salento University, Lecce, Italy, and at the CDRC - Centre de Datation par le RadioCarbone, Lyon 1 University, Lyon, France. |

☒ Tick this box to confirm that the raw and calibrated dates are available in the paper or in Supplementary Information.

| | |
|---|---|
| Ethics oversight | No ethics oversight was required strictly, however we confirm that we followed established ethical guidelines for archaeogenetic research. |

Note that full information on the approval of the study protocol must also be provided in the manuscript.

