## [Peer Review File · Nature]

Manuscript Title: Extensive pedigrees reveal the social organisation of a Neolithic community

Reviewer Comments & Author Rebuttals

Reviewer Reports on the Initial Version:

Referee expertise:

Referee #1: archaeology, kinship practices

Referee #2: geochemistry

Referee #3: relatedness inference

Referee #4: aDNA

Referees' comments:

Referee #1 (Remarks to the Author):

Review of Rivollat et al. Nature research article ms 'Extensive pedigrees reveal social organisation of Neolithic communities'

This is a very impressive study, which breaks new ground in exploring aspects of kinship in Neolithic France in great detail – and indeed provides an exemplar for further such studies of prehistoric burial grounds. An exceptionally rich combination of archaeological contextual data, osteological analysis of age-at-death, stable isotope data, and genetic data on a high number of individuals yields a fine-detail analysis that supports a series of important inferences about kinship and social relations. Key results include: spatial patterning took place within the cemetery on kinship grounds, including by lineage and sub-lineage; equal numbers of subadult lineage children from both sexes were present, indicating there was no sex-based selection when burying those who died in childhood; women reproducing with lineage males had grown up on different geologies to those lineage males with whom they reproduced (which is consistent with female exogamy and virilocality); the first three generations in lineage A had lived elsewhere in childhood and either their remains were relocated to the burial ground after they had died (individual 270B) or they moved to a location near the burial ground in later life, and; changing age profiles for the earliest and latest generations indicate the inclusion of an already-deceased lineage founder (270B) along with adults who had formed the site in generations two and three, while the burial ground was abandoned by some 5th and 6th-generation adults who would end up being buried in a different place to their 6th or 7th-generation children who had died in childhood and were the last burials at Gurgy. The study makes use of some data and findings from earlier studies of the same burial ground, but very significantly expands upon these findings and adds an extensive new dataset including Strontium isotope analyses and new genetic results on c. 100 individuals. The study is therefore original, and its

significance lies in presenting these novel findings for a large number of biologically-related individuals in an early burial ground. As an archaeologist specialising in prehistoric northern Europe, my comments are limited to the archaeological and anthropological components of the project, including how these incorporate the results of the genetic analyses, and its archaeological value. The genetic analyses themselves are beyond my field of expertise.

All of the results I am qualified to review appear valid and robust, with no obvious flaws. The methods and data are outlined in comprehensive detail in the supplementary materials. The images are extremely effective in supporting the text and elucidating the analyses, and the conclusions are well-supported by the data and argument, though I query two specific social conclusions: the inference of nuclear families from the results and the inference that Gurgy was used by low-status sections of society while Passy type cemeteries include the remains of an elite. The argument for these could be strengthened or qualified by mentioning alternative interpretations. The flow of argument is clear. The authors explain appropriately that they are aware that human reproductive relationships are only one aspect of kinship (L 345-7), and that they are aware of the problems with using western kinship terms in this context (L108-111). However, the piece lacks a statement on sex identification, and one of supplementary table 1 should identify the results of the osteological sex analysis alongside the genetic sex identification of each individual. Appropriate references to key relevant literature are provided. Below I set out some specific queries or identify areas where this already very strong article could be further strengthened with a little editing, but overall this is an excellent submission which merits publication in this venue. I expect it to become a leading point of reference in studies of prehistoric European societies, and archaeological explorations of kinship more generally.

Abstract

L45: 'local vs. non-local signatures': Given that the Sr data informs on locality in childhood, would it be more accurate to say something like 'those who lived locally in childhood and those who did not', or 'local vs. non-local childhood signatures'?

L57: 'stable conditions' – Can it be made clearer that this means supportive conditions for good health and survival into adulthood (as per page 13)?

L59: Isn't it the combination of age-structure differences by generation and the Sr isotope results that suggest 'shifting sedentary farming'?

Main text

Line 74: 'graveyards, which are expected a priori to represent the non-elite people.' Why start with this a priori expectation given the wide range of mortuary treatments practiced in Neolithic Europe?

L78: 'in close vicinity to' - how far is close?

L86-91: It is not clear here if there is at least one kind of genetic result for each of the 128 individuals buried at the site or not. Could this be reworded to make that clear?

L118: Fig 1 caption and L159: The term 'posthumous ancestors' is ambiguous – what is implied by adding 'posthumous' to 'ancestors'? If the point here is that the remains of 231A and 270B both featured in extended or secondary mortuary treatments so their remains were deposited long after

they died, please make that clearer. It appears that only 270B was treated in this way but editing Supplementary Table 1 to include data on burial contexts and bodily treatment would help here. The text does not mention that 231A is buried with 231B and 231B does not appear in the pedigree visualisation or in Supplementary Table 1 although the site plan shows 231B, so it is not clear to the reader what this grave contained.

L143: 'Genetic relationships inform on social relations and residence'. This heading is a little misleading as while the coming section draws solid inferences about the co-location of biological relatives (L163-187), and endogenous compared with exogenous lineage members, the inferences drawn about in-life residence depend on a combination of genetic data, spatial relationships between burials, and stable isotopes.

L174-187: The authors here demonstrate that biological relatedness was important in deciding who each person should be buried near to, and the article overall demonstrates patrilineal descent was a key factor in where a person was buried. However, the term 'inheritance' is used at the end of this paragraph without explanation as there do not seem to be any specific aspects of funerary practice that were inherited by certain sub-lineages or any kinds of artefacts consistently associated with the graves of those sub-lineages (nor runs of Sr isotope results that might relate to land transfer). It seems to me that while lineage was important, differences in the social standing of the sub-lineages were either minimal (e.g. denoted by grave size in a few cases, similar numbers of grave goods for parents and a child in a couple of cases in supplementary table 21), or were not referred to when it came to depositing the remains of the dead (i.e. there were no systematic differences in mortuary structures, features or markers, or grave goods – suppl. Note 12). The exceptional treatment of 270A was seemingly carried out sometime after initial deposition and may relate to their acquiring status as the ancestors of a successful lineage. Would the authors say that there was little evidence of enduring social status differences between sub-lineages?

L215: It would be more accurate to say 'reproductive unions' instead of 'heterosexual unions' given that we have no way of knowing the sexual orientations of the individuals involved (also applies in supplementary text).

L219: 'We thus observe a preferential funerary bias, which needs to be considered independently of female exogamy.' This is interesting, and a similar conclusion was reached for the tomb at Hazleton North (Fowler & Olalde et al 2022). In that case those exogenous women who we know reproduced with lineage men were all from the first two generations, forming founding ancestors for the sub-lineages, but that does not seem to be the case at Gurgy. Do the authors have a view on why some exogenous mothers and not others were included across several generations?

L229-31: Is it possible to phrase this in a way that gives the reader a more explicit description of what the differences were?

L232-236: Can a schematic image be produced to illustrate the specific changes in grave goods by age at the site (e.g. similar to that used by Sofaer Derevenski 2000, figure 1), or Hofmann (2009, figure 3)? This could be included in supplementary note 14, and could also include reference to the changes in diet mentioned in the text. I am aware similar schematics are published in Le Roy et al.

(2018) and Rey et al. (2019): those could be referred to instead, but they are not specific to Gurgy and do not compare sex categories.

L268-295: It does seem extraordinary that there are no cases of half-siblings among the burial population. The interpretation presented seems plausible, but is it also worth considering the possibility that there were taboos against, or significant sanctions against, engaging in sexual intercourse with multiple partners?

L333: 'strontium isotope values suggest a move of each generation to another geographical location': The supplementary note 11 is very clear on this topic, and the novel inference that this suggests shifting agricultural plots from which food was sourced is convincing, but there are questions about some of the interpretation of social units that seem to rest on this evidence. P49 of the supplementary document suggests 'it is possible that individuals from a given family branch and close generations share similar diets, resources, and environments, demonstrating resource management strategies at the sub-family level'. This makes sense, but I am concerned that the references to 'nuclear families' in the article gives a misleading picture of what can be said about 'sub-family' (or, I would suggest, 'sub-lineage') groups. Are the Sr results also consistent with an interpretation that the whole community sourced their food from shared locales and when they shifted sources, they did so together? Is it even possible then that similar results within a generation (or two) tell us something about which individuals grew up contemporaneously when the group was living in one specific locale and drawing its food from the same range of locations? For instance, could there be an age gap between (a) siblings 255, 224, 212 and (b) siblings 213, 317, with an episode of relocation between the periods in which the sampled teeth of these two groups of siblings were forming? And could the similarity in Sr results suggest that 317 was contemporary and co-resident with 250 (and the same for 276 and 245B with 255, 224 and 212), growing up in the same locale (or at least eating food from the same locale)? The spatial clustering of these burials does not suggest a close relationship between 317 and 250, compared with 317 and his siblings, so if we do infer that proximity in the cemetery relates to social proximity there is no suggestion that 317 and 250 were socially particularly close (same for 276, 245B and 255) – the question is rather whether these distant biological relatives with similar Sr readings from the same generation resided in the same locale at the same time. Overall, could the authors clarify here whether they are suggesting (a) one large residence group relocating repeatedly, rather than (b) a series of dispersed sub-lineages residing separately from one another and each moving periodically?

L344 (and use of the term in L852, extended data fig 5 caption): 'emphasis on "nuclear" family units': As above, I am less convinced about this interpretation than the rest of the article because the authors have not presented a clear case for who resided with whom at a household level. This might just be a matter of terminology: it is not clear to me how the study can distinguish a nuclear family from an extended and/or cross-generational family, but I think a clear case has been made that the community traced descent at the sub-lineage level so 'sub-lineage' would be a more appropriate term. The correlation between grave proximity and biological relatedness (including reproductive partners) is sound, but if the implication is that this indicates separate nuclear family residences in life then I would point out that overall the graves are very close together in the same nucleated burial ground: can it be demonstrated that the living contemporary kin did not also reside as one or two extended co-residential groups?

L350-358 and supplementary note 14.1, p46: The conclusion that those buried at Passy tombs were the elites while those in cemeteries such as Gurgy were non-elites is not the only possible interpretation, and the comparison across these sites deserves further discussion given the complexity of the regional situation and the various cultural affinities attested at Gurgy, as outlined in Supplementary note 1. The recent aDNA study of Fleury-sur-Orne demonstrated that only one or two representatives from each lineage were selected for deposition there (one of which was female) (Rivollat et al. 2022) – but we do not know where their relatives' remains were deposited (if they were at all), so we do not know if their kin should be characterised as an elite community. By comparison, the Gurgy results illustrate wider social representation. It seems possible that under certain circumstances a representative of such a community could potentially be selected for special treatment at a different site to the family burial ground, such as a Passy type burial ground. This need not divide society into distinct sets of elites and commoners with status inherited by all members of a lineage across multiple generations. We also know lineage ancestor 270B was given some unusual treatment at Gurgy, illustrating differential selective treatment could occur. Are the burial patterns and aDNA results from Fleury-sur-Orne, part of a cluster more than 200 km north of Gurgy, thought to be typical for the cluster of STP sites local to Gurgy? Does the radiocarbon modelling shed light on whether Gurgy and local Passy burial grounds were contemporary? How much variation was there in terms of the numbers of grave goods or how exotic they were between the individuals at Passy sites, compared to the degree of variation between burials at Gurgy? Several of the graves at Gurgy include scaphopod shells, and there are examples of beaver incisors, deer bones and pig bones – are these not equivalent to the grave goods made from wild species from Passy burial grounds? I note that 270A was buried with three arrowheads – were these the same kinds of arrowheads sometimes found in Passy graves? Either way, could 270A and 270B have held roughly equivalent standing in the Gurgy community to people buried elsewhere in Passy tombs – heads of lineages, perhaps? Could the Gurgy community (characterised here as healthy, co-operative and connected with other communities) have rejected (or predated) the more hierarchical relations or funerary practices developed by others in the region? Are the following all possible, and if not what is the evidence that supports the first one above the others: (a) an elite/non-elite distinction encompassing Passy sites and Gurgy, (b) a society generally using local burial grounds but which might infrequently deposit selected people at shared Passy necropoli and (c) two kinds of burial grounds used by neighbouring communities with different attitudes towards social ranking and inclusiveness of burial. The final section of supplementary note 14.1 might be the place to address some of these questions, supported by a little edit of these lines in the main text.

L358-62: This needs editing for clarity, I can't follow the point (though supplementary note 14 explains it clearly).

Images:

Figure 1c: Can the colour coding for grave 270 be altered to show that 270A is 'no data'; same for 231B?

Supplementary data:

The extended data figures are extremely useful, but there are several different tables and images that an archaeologist needs to navigate to piece together the basic contextual information for each

individual, and some of that data is not present. Can Table 1 be modified to include columns providing data on body positioning and orientation, and information about the positioning of the grave goods in relation to the body? Combining the isotopic results into this table (including the previously-published Carbon, Nitrogen and Sulphur as well as the new Strontium data) and identifying which lineage and generation each individual is assigned to would make the table of the most use to archaeologists. Table 1 currently does not include unsampled individuals buried at the cemetery, but it would be useful to include the archaeological and osteological information for them too, especially 231B and 207A given their importance to the analysis. Osteological data, including on pathologies, would be useful given the statements about conditions supporting a good rate of survival into adulthood (and a few lines discussing this would also improve the supplementary notes). If the Sr data is to be kept in a separate table then it would be useful if a column could be added identifying the sex of each individual and their place in the lineages (e.g. lineage A or B, which generation, and whether they were endogenous or exogenous to the lineage).

Supp note 1:

P4: 'double but mainly simple' – I think simple should be single.

It would be very useful to include a map here marking up the distribution of STP mounds and the other funerary sites mentioned in this section in relation to Gurgy.

Suppl note 11:

What criteria were used for selecting individuals to be sampled for Sr – i.e. were all individuals with the relevant surviving dentition sampled?

Suppl note 14:

P42: Perhaps caution could be voiced here about absent people (especially sons) from generations 2 and 3 given that some graves central to the burial ground did not provide aDNA results. Given the spatial analysis suggests that children were buried near parents, graves near to 270 and 237 could have held the sons of those individuals that the text implies were missing from the burial ground.

P45: I suggest removing the term 'patriarch' here. While he may have been recognised in life as a lineage founder, we do not really know what social standing or authority that individual had in life: his importance may have grown only through the achievements of his descendants, for instance, some of whom seem to have relocated his remains.

Should 288-289B be added to the list of exceptions to the patrilineal scheme discussed in suppl.

Note 14?

References:

Hofmann, D. 2009. Cemetery and settlement burials in the Lower Bavarian LBK. In D. Hofmann and P. Bickle (eds), *Creating communities. New advances in central European Neolithic research*, 220-34. Oxford: Oxbow

Sofaer Derevenski, J. 2000. Rings of life: the role of early metalwork in mediating the gendered life course. *World Archaeology* 31: 389-406.

Reviewed by Chris Fowler

Referee #2 (Remarks to the Author):

The manuscript of Rivollat and co-authors "Extensive pedigrees reveal social organisation of Neolithic communities" reports on genome-wide, strontium isotope and contextual data from >100 Neolithic individuals from the site Gurgy 'les Noisats' (Paris Basin, France), to reconstruct kinships and social organization in this Neolithic community. One of the most interesting aspects is the study of "non-elite people" from a common graveyard of the European Neolithic. Indeed, as the authors highlighted, studies from this period are commonly focused on specific funerary contexts, implying select groups or individuals of specific status.

The questions are interesting and can be of great interest to the general readers in several disciplines. In my humble opinion, this paper is a very good candidate for Nature. Abstract, introduction and conclusions appropriate and fully meet Nature criteria.

The dataset is impressive, including more than 100 genetic data, 8 new radiocarbon dates, and high spatial resolution strontium isotopes data from 57 individuals. Procedures have been well described, and appropriately administered and interpreted. The reporting of data and methodology is detailed and transparent to enable reproducing the results of the research.

The conclusions are valid and reliable. Every aspect of different possible interpretations has been considered and the conclusions are strongly supported by the data interpretation.

Here following some minor comments.

In Figure 1d, is reported a photograph of the grave 270 with two individuals, A and B. While in the caption 270B is described as main male ancestor of pedigree A, it took me a while to understand that ID 270A didn't get enough coverage to be included in the pedigree, and this is mentioned only later in the main text. I would suggest specifying this in the caption.

Line 193: "Notably, four of these six females (GLN212, 213, 277 and 289B) had no known offspring, although having reached reproductive age".

In my opinion, this sentence can be misleading. No offspring of these female adult individuals have been found in the burials, and the reasons (they had no children, the children have been grown and moved elsewhere, the remains are poorly preserved in terms of odontoskeletal preservation and DNA coverage) cannot be known in any way.

In Supplementary Note 13, paragraph "87Sr/86Sr intra-profile variation and individual life histories", reference #84 is clearly wrongly assessed. Being aware of the difficulty in managing such amount of Supplementary materials from different authors, I suggest the authors to check the reference in the Supplementary notes.

Lines 279-280: "Including unsampled inferred adults, we observe two cases of up to six children from the same couple (Extended Data Fig. 6)".

Is the reference to Extended data fig. 6 correct?

Referee #3 (Remarks to the Author):

Rivollat et al. present ancient DNA sequences from individuals buried in Gurgy, France, including genome-wide data for 94 individuals (72 new), mitochondrial DNA (mtDNA) for 100 individuals (78 new), and Y chromosome data from male subjects. Remarkably, using relatedness detection software designed for ancient DNA, the authors were able to reconstruct two large pedigrees—the larger spanning 7 generations—that include most of the 94 subjects. From these pedigrees, they find strong evidence for patrilocality and female exogamy; for example, they found only two Y haplogroups in the study subjects versus 36 distinct mtDNA haplogroups. Furthermore, adult female descendants in the pedigrees are strikingly few, suggesting that there may have been networks of groups that also practiced female exogamy to which these women migrated. Distant relatedness detected between some married-in females lends support to this theory. Another striking feature in the pedigree is a complete absence of half-sibling relationships, implying this community practiced monogamy, with no evidence even for serial monogamy from the data. Finally, the authors argue that this burial ground, which lacks the monuments present in nearby contemporary burial sites, may give a window into the practices of non-elite individuals at this time period (4300-4700 BCE, the Middle Neolithic).

One concern relates to the PMR method for evaluating relatedness. It makes sense methodologically but hasn't been validated to the same degree as the other relatedness methods. For example, might it be biased towards calling pairs as more closely related than they really are? Given that several somewhat ambiguous relatives were judged as truly second degree on the basis of the PMR approach, running some analyses of this using simulated or perhaps real data seems important. Are its probabilities well-calibrated? Use of the PMR statistics seems only to occur in questionable cases; is it worth checking all classifications with the PMR approach? Another simpler question is whether the median P0 value used with READ is downwardly biased because the data are so enriched for relatives. Do any of the relatedness signals shift if you calculate the median P0 value with male pairs removed?

Besides the pedigree, I wonder if the results from strontium isotope analyses may be a bit overstated. Figure 1e plots these isotope values, with a fitted line that increases with generation, but are the differences e.g., between adult females and adult males statistically significant? I didn't see any statistical tests for the claims here. The same concerns apply to the carbon, nitrogen, and sulfur isotope analyses which seem not to include p-values.

The main text mentions that the female diversity in this group is visible from the inferred phenotypes, but the cited Supplementary Note (9) does not go into detail on female versus male phenotypic diversity.

Minor:

Caption to Extended Data Fig 9c: "Individual GRG095 shows an inbreeding signal similar to a first-cousin union." - Should this say GLN282? Notably the main text and other places suggest this person is the offspring of a second or third-cousin union whereas this caption seems to say first-cousin. Also

pgs. 44 and 45 of the supplement uses first-cousin again. The Ext Data Fig 9c plot seems most similar to the first cousin union and quite different from the second and third cousin plots. Perhaps it's worth reconsidering this classification?

Detailed:

Line 87: typo - "wide-genome"

Paragraph starting on line 122: I know the IBD method isn't published, but it seems worth citing something; even the webpage helps.

Line 202-3: "could explain some missing links between the isolated adult females and the rest of the pedigrees" - I didn't quite follow this sentence.

Line 248: "structured by a range of group features" - it might be nice to reword this given the term structure in population genetics has a specific meaning that is perhaps best to avoid confusion with here.

Line 259: this might read more clearly if worded as "siblings GLN243A and GLN268B share the same mitochondrial haplogroup H1 as GLN315".

Line 275: what does "shortfalls" mean?

Line 292: since the ROH analysis uses varied lengths of segments, should the word "long" be deleted here?

Line 515: what is YMCA?

Line 555: are citations 52 and 53 appropriate here?

Line 572-4: I may be misunderstanding, but this seems to say that the reads are mapped with both samtools and EAGER?

Line 581: typo - "and"

Caption to Extended Data Fig 1b: please note the inset here.

Caption to Extended Data Fig 7a: what are the bands around the dashed lines?

Supplementary table 19: I'm not sure what each of the columns here are; these may be standard, but adding definitions would make this more easier to understand.

Minor in Supplementary Notes:

Pg. 15 "Comparing IBD sharing with inferred pedigrees": it might be worth noting what the analysis shows for the outlier parent-child pair that looks like a full sibling pair.

Detailed:

Pg. 6 defines $r = k_1/4 + k_2/2$, but this is half what is normally called the coefficient of relatedness and is instead the kinship coefficient. It appears that the the values in the paper are indeed coefficients of relatedness, with $r = k_1/2 + k_2$. Indeed, a few lines above this formula, the text says, "r equals 1/2 in the case of parent-offspring or siblings" but just after the formula, it says "While $r=1/4$ for both of the 1st degree relationships ..." So there is one discrepancy here.

Pg. 6 says "the probabilities k_0 , k_1 and k_2 differ ... depending on rate of recombination." Note that recombination only affects the variance in sharing among siblings (not parent-child pairs) and not the expected values given in the text.

Pg. 7: "we thinned the data such that all sites were at least 200K bases apart" - how did you choose the sites to retain?

Pg. 8 says GLN226 is a daughter, but the figure shows this subject as male?

The Tree Reconstruction subsection of Note 2 is nicely organized. It would be nice (but may be more

work than the authors feel like making) to depict the procedure of reconstructing the pedigree that one can look at while reading. I found myself having to visually hunt in Figure 1 to find the subjects (and searching didn't work as text is not embedded in this - perhaps this could be addressed?). One option would be to outline each of the (say) first degree connections in one figure and add labels then reference these from the text. A second figure could similarly show the second degree links.

Pg. 10: "with the offspring of GLN237A, as expected as he is their" should the second "as" be "if"?

Pg. 11 final bullet: although I see that GLN210 cannot be an uncle, the text doesn't make this fully explicit. (Several of the subsequent bullets on pg. 12 also do not fully explain why uncle/aunt relationships are ruled out.)

Pg. 12: "2nd-degree related to the siblings GLN237A, GLN325, GLN265, GLN221B and GLN266, which is not the case" - in fact GLN237A is the parent of the rest.

Pg. 13, line 2: why can't the person be a grandmother? Age?

Pg. 13: typo - "un"

Pg. 13, first bullet under Pedigree B: should this say that the connection is through the father?

Pg. 15: citations 40 and 41 seem incorrect.

Pg. 15: Paragraph just under "Comparing IBD sharing with inferred pedigrees" - "we obtain the degree of relatedness". Nitpicky, but how does this handle full vs half relatives?

Pg. 15: typo - "IDB".

Pg. 18: "unrelated subadult males" - are these 229 and 308?

Pg. 20: typo - "ROH>4cM24".

Pg. 21: typo - "All these females are not or very distantly related".

Pg. 29: "WHG" - I don't think this abbreviation is defined elsewhere.

Pg. 42: "Here, the burials of fathers and their subadult male offspring are located significantly closer to each other than any other pairs of individuals (Supplementary Note 12, Extended Data Fig. 6c)" - is there a p-value for this?

Pg. 42: typo - "long a time"

Pg. 45: "legitimized through the 2nd degree relationship to GLN270B." - is this through GLN311? Could spell this out a bit more.

Pg. 47: "and k a correction coefficient worth 10% of t." - not sure I understand this.

Referee #4 (Remarks to the Author):

General comments:

The authors generated genome-wide DNA data for 94 individuals, 8 new 14C dates, and Sr ratio data for 57 individuals and reconstruct pedigrees to study the social organization of a Neolithic community (please see my comment below about the manuscript title which I think must be edited).

The manuscript is technically extremely sound, and I have no analytical concerns (at least regarding the generation and analysis of DNA). It's fantastic that the entire graveyard was studied, which is one particular strength of this work. The figures associated with this manuscript are also excellent and serve as a great tool for the reader.

I recommend some moderate revisions are made to this manuscript prior to publication.

The primary critiques that I have surround the style of writing and the novelty of this work. First, the body of this manuscript reads mostly as a series of observations instead of a cohesive "story" that is well-supported by the data. I applaud the authors for making so many astute observations, but would encourage them to ensure that the data are being used to clearly support central interpretations -- it feels at present like there are long stretches where they are presenting observations one after the other, which can be exhausting and confusing to the reader.

There is also room for improvement in tying in observations with their implications - e.g., "In addition, only six of the 20 adult females buried at the site are descendants of the main pedigree lines A and B. Notably, four of these six females (GLN212, 213, 277 and 289B) had no known offspring, although having reached reproductive age." now it would be very helpful to hypothesize about what this could mean and tie it in to the conclusions you are drawing. Has this pattern been seen before? If so, where? If not, why not? Why would this be special/important/unique/worth writing about? I tried to point out places throughout the manuscript where I noticed this in particular, although I should also note that there are places where this is done well. I especially thought the conclusion to this manuscript was well-written.

While this paper is technically sound and interesting, I'm not fully convinced of its novelty. Other than being the largest pedigree constructed to date with ancient DNA and focusing on a cemetery instead of a structure like a megalith, what makes this paper exceptionally unique? Maybe this needs to be pointed out more clearly. I think that we are at a point in the field of ancient DNA research where being the largest study (especially in an area like Europe) cannot itself be considered particularly novel.

A small comment is that there are a moderate number of grammatical errors throughout the manuscript especially in terms of comma usage that should be fixed prior to publication. Some run-on sentence with multiple clauses should also be split for ease of interpretation and I indicated several of them in my detailed comments.

Finally, not having data for GLN270A weakens some of the conclusions that could be drawn about a potentially very important individual (and beyond this individual as well) - is it really impossible to generate sufficient data for this individual? I imagine if it was possible the authors would have done so -- but I would encourage an attempt to improve the data for this individual if one has not yet been made.

General recommended edit:

Title - This analysis doesn't really tell us about the organization of multiple communities, but instead only a single one. I would strongly suggest a change to "Extensive pedigrees reveal social organization of Neolithic community at Gurgy 'les Noisats'"

Specific recommended edits:

Line 43 - Arguably not only burial grounds, but also sites of habitation, work sites, etc, where

evidence of exchange and interaction may be even more evident than in burials. I would change this to "...these systems can only be studied indirectly through biological and material culture remains" (or something like this)

Line 45- Define "burial community" so that there is no confusion as to what you mean (this may mean different things to different readers and there is a chance of confusion)

Line 48- tone this down a bit: "...and can aid in the interpretation of kinship practices in prehistoric societies."

Line 49- this wording is strange. I recommend "genome-wide DNA data as well as strontium isotope and contextual data"

Line 54- for more clarity, change 'of the linked and unlinked individuals to the pedigrees'  'of individuals both linked and unlinked to the pedigrees'

Line 57- how are stable conditions and high fertility consistent with social ties? these things can certainly exist independently. why should we believe they are connected here?

Line 58 - change to 'first and last generation of the pedigree'

Lines 64-70 - this explanation is excellent!

Lines 80-84 - review this sentence, it seems incomplete

Line 84 - "duration of use of the site" would be better

Line 86- much more precise would be "we took samples from the remains of every individual in the graveyard"

Lines 86-89 - this sentence has too many clauses, break up into two (preferably starting a new sentence after "22 of which were published previously")

Line 103- add "connects" after Pedigree B

Lines 105-106 - better wording would be "one adult male has two second-degree relatives in Pedigree A"

Line 107 - it would be helpful to tell at this point how many individuals out of the original 94 have no detectable relatives at all.

Line 109 - "for ease of use" is unnecessary

Lines 109-111 - this sentence needs to be re-written because it does not work well grammatically.

Lines 122-126 - add a citation for this "recently developed method" and a pointer to the supplementary note where it is described

Line 127- change further  additional

Line 135 - "beyond a 3rd degree" sounds strange

Line 155- remove "itself"

Line 156- not having data from GLN270A is a big bummer. Is there really no hope of generating data? If there is any way that it is possible to do so, this should be done. The conclusions made in the rest of the paragraph are substantially weakened because of the lack of data from this individual.

Line 190 - "all adult mothers" -- how many, exactly?

Lines 190-191 - does the Sr ratio data corroborate this?

Lines 191-192 - is this a significant deviation from expectation? This number actually is higher than I might expect.

Lines 193-194 - what does this imply?

Lines 204-208 - this is really nice!

Lines 219-220 - can you elaborate on this potential bias? has this been seen before? how might this be explained?

Line 230 - change to "a sex-based dietary division"

Line 232 - give more details on this purported sex-related different treatment? what is it and why? what other evidence supports this possibility?

Line 236 - is this seen elsewhere?

Lines 238-239 - is this diversity significantly higher than expected? Why do you believe this is notably high?

Line 251- "confirms the will to avoid inbreeding" -- this is strange wording, can it be changed to something like "confirms the avoidance of mating among close relatives"?

Lines 306-314 - these are really interesting findings!

Line 644- why isn't IntCal20 being used? It should be.

Author Rebuttals to Initial Comments:

Title: "Extensive pedigrees reveal the social organisation of a Neolithic community

Tracking #: 2022-10-16582

Authors: Rivollat et al.

Referee expertise:

Referee #1: archaeology, kinship practices

Referee #2: geochemistry

Referee #3: relatedness inference

Referee #4: aDNA

Referees' comments:

Referee #1 (Remarks to the Author):

Review of Rivollat et al. Nature research article ms 'Extensive pedigrees reveal social organisation of Neolithic communities'

This is a very impressive study, which breaks new ground in exploring aspects of kinship in Neolithic France in great detail – and indeed provides an exemplar for further such studies of prehistoric burial grounds. An exceptionally rich combination of

archaeological contextual data, osteological analysis of age-at-death, stable isotope data, and genetic data on a high number of individuals yields a fine-detail analysis that supports a series of important inferences about kinship and social relations. Key results include: spatial patterning took place within the cemetery on kinship grounds, including by lineage and sub-lineage; equal numbers of subadult lineage children from both sexes were present, indicating there was no sex-based selection when burying those who died in childhood; women reproducing with lineage males had grown up on different geologies to those lineage males with whom they reproduced (which is consistent with female exogamy and virilocality); the first three generations in lineage A had lived elsewhere in childhood and either their remains were relocated to the burial ground after they had died (individual 270B) or they moved to a location near the burial ground in later life, and; changing age profiles for the earliest and latest generations indicate the inclusion of an already-deceased lineage founder (270B) along with adults who had formed the site in generations two and three, while the burial ground was abandoned by some 5th and 6th-generation adults who would end up being buried in a different place to their 6th or 7th-generation children who had died in childhood and were the last burials at Gurgy. The study makes use of some data and findings from earlier studies of the same burial ground, but very significantly expands upon these findings and adds an extensive new dataset including Strontium isotope analyses and new genetic results on c. 100 individuals. The study is therefore original, and its significance lies in presenting these novel findings for a large number of biologically-related individuals in an early burial ground. As an archaeologist specialising in prehistoric northern Europe, my comments are limited to the archaeological and anthropological components of the project, including how these incorporate the results of the genetic analyses, and its archaeological value. The genetic analyses themselves are beyond my field of expertise.

We thank reviewer 1 for his very constructive feedback.

All of the results I am qualified to review appear valid and robust, with no obvious flaws. The methods and data are outlined in comprehensive detail in the supplementary materials. The images are extremely effective in supporting the text and elucidating the analyses, and the conclusions are well-supported by the data and argument, though I query two specific social conclusions: the inference of nuclear families from the results and the inference that Gurgy was used by low-status sections of society while Passy type cemeteries include the remains of an elite. The argument for these could be strengthened or qualified by mentioning alternative interpretations.

1) Thanks. We address these two comments in points 18 and 19 below.

The flow of argument is clear. The authors explain appropriately that they are aware that human reproductive relationships are only one aspect of kinship (L 345-7), and that they are aware of the problems with using western kinship terms in this context (L108-111). However, the piece lacks a statement on sex identification, and one of supplementary table 1 should identify the results of the osteological sex analysis alongside the genetic sex identification of each individual.

2) We have completed supplementary table 1 with extra information related to osteological and archaeological information. See a more detailed answer in point 22 below. The first paragraph in Supplementary Notes 1, section "Gurgy individuals", explains in detail which methods have been used to estimate the osteological sex and how many individuals were assigned.

Appropriate references to key relevant literature are provided. Below I set out some specific queries or identify areas where this already very strong article could be further strengthened with a little editing, but overall this is an excellent submission which merits publication in this venue. I expect it to become a leading point of reference in studies of prehistoric European societies, and archaeological explorations of kinship more generally.

Abstract

L45: 'local vs. non-local signatures': Given that the Sr data informs on locality in childhood, would it be more accurate to say something like 'those who lived locally in childhood and those who did not', or 'local vs. non-local childhood signatures'?

3) We modified the sentence according to the second proposition.

L57: 'stable conditions' – Can it be made clearer that this means supportive conditions for good health and survival into adulthood (as per page 13)?

4) We agree and have extended to 'stable health conditions' but cannot be more detailed in the abstract given the length restrictions.

L59: Isn't it the combination of age-structure differences by generation and the Sr isotope results that suggest 'shifting sedentary farming'?

5) Yes, this point was missing in the abstract. We revised the last sentence of the abstract accordingly, now reading: "Age-structure differences and strontium isotope results by generation indicate that the site was used for just a few decades, providing new lines of evidence for shifting sedentary farming practices during the European Neolithic."

Main text

Line 74: 'graveyards, which are expected a priori to represent the non-elite people.'
Why start with this a priori expectation given the wide range of mortuary treatments practiced in Neolithic Europe?

6) Thanks, this is a fair point. We agree and have rephrased this sentence to remove any preconceived ideas about the Gurgy graveyard at this point of the text, which now reads: "Studies on biological relatedness in the European Neolithic are still rare and so far have only focussed on groups from specific funerary contexts like megaliths⁶⁻⁸ or mass graves⁹, which typically cover high-status groups or individuals of specific status, but have not included non-specific graveyards that may be more representative of the general population."

L78: 'in close vicinity to'- how far is close?

7) Thanks. Yes, one attribute is sufficient. We added distances to the next sites in brackets (in a radius of 100 km, see Supplementary Note 1)

L86-91: It is not clear here if there is at least one kind of genetic result for each of the 128 individuals buried at the site or not. Could this be reworded to make that clear?

8) This paragraph describes the archaeological and anthropological background. The rationale, sampling and success rate are described in the follow up paragraph in lines 89-91 where we describe the genetic results. We noticed that a reference to the Material and Methods section was missing, where the sampling strategy is explained in more detail. We also revised the sentence in the main text, now reading: "To investigate the intra-site structure and the characteristics of Gurgy, we sampled the remains of 110 out of 128 individuals with suitable skeletal preservation (Materials and Methods) and retrieved wide-genome aDNA data for 94 individuals (Supplementary table 3), ..."

L118: Fig 1 caption and L159: The term ‘posthumous ancestors’ is ambiguous – what is implied by adding ‘posthumous’ to ‘ancestors’? If the point here is that the remains of 231A and 270B both featured in extended or secondary mortuary treatments so their remains were deposited long after they died, please make that clearer. It appears that only 270B was treated in this way but editing Supplementary Table 1 to include data on burial contexts and bodily treatment would help here. The text does not mention that 231A is buried with 231B and 231B does not appear in the pedigree visualisation or in Supplementary Table 1 although the site plan shows 231B, so it is not clear to the reader what this grave contained.

9) We removed the term “posthumous” in Fig. 1 caption as it was unclear and replaced it with “reburied remains” for individual GLN270B. Of note, this term never referred to GLN231A who has a single primary burial.

We added archaeological information about graves, body treatment, position and orientation in a new Supplementary table 21. We also clarified which individuals were buried together and the type of grave in which they were buried (see also answer to point 22) in Supplementary table 21, as well as in the section titled “The Gurgy site” of Supplementary Note 1, which now reads: “In Supplementary table 21, we include information about single and double burials, in which individuals were buried simultaneously in the same pit (for instance GLN207A and GLN207B) and which individuals were buried one after the other in the same pit (for instance GLN221A and GLN221B)”.

We also modified the layout of the site map to show both individuals when they are in double burials, which was indeed not shown clearly.

L143: ‘Genetic relationships inform on social relations and residence’. This heading is a little misleading as while the coming section draws solid inferences about the co-location of biological relatives (L163-187), and endogenous compared with exogenous lineage members, the inferences drawn about in-life residence depend on a combination of genetic data, spatial relationships between burials, and stable isotopes.

10) Thanks. We fully agree and have changed the header to read “Combined evidence informs on social relations and residence”

L174-187: The authors here demonstrate that biological relatedness was important in deciding who each person should be buried near to, and the article overall demonstrates patrilineal descent was a key factor in where a person was buried.

However, the term 'inheritance' is used at the end of this paragraph without explanation as there do not seem to be any specific aspects of funerary practice that were inherited by certain sub-lineages or any kinds of artefacts consistently associated with the graves of those sub-lineages (nor runs of Sr isotope results that might relate to land transfer). It seems to me that while lineage was important, differences in the social standing of the sub-lineages were either minimal (e.g. denoted by grave size in a few cases, similar numbers of grave goods for parents and a child in a couple of cases in supplementary table 21), or were not referred to when it came to depositing the remains of the dead (i.e. there were no systematic differences in mortuary structures, features or markers, or grave goods – suppl. Note 12). The exceptional treatment of 270A was seemingly carried out sometime after initial deposition and may relate to their acquiring status as the ancestors of a successful lineage. Would the authors say that there was little evidence of enduring social status differences between sub-lineages?

11) We had used the word "inheritance" in reference to the fact that some features were found to be shared between fathers and sons, e.g. the status from 270B as main ancestor which is visible in the bigger graves of his son and his grandson, as well as the significant associations of adult males with their sons rather than with their daughters, nephews or nieces. However, we acknowledge that the funerary practices at Gurgy are not reflecting a systematic inheritance pattern, neither in the grave goods, the body position, the pit types, nor the orientation of the graves. We therefore removed the word "inheritance" at the end of that paragraph.

As a consequence, it is also very difficult to assess potential differences in social status between the sub-lineages at Gurgy, as there is almost no evidence on which to rely. The only visible feature is the marking of the main lineage over the first two generations following the main ancestor of the group, which we interpret as inheritance of status from GLN270B to his son and grandson. However, this marking is not maintained in later generations and no comparable observation is made between generations of other lineages.

L215: It would be more accurate to say 'reproductive unions' instead of 'heterosexual unions' given that we have no way of knowing the sexual orientations of the individuals involved (also applies in supplementary text).

12) We agree and have modified the main text and the supplementary text accordingly.

L219: 'We thus observe a preferential funerary bias, which needs to be considered independently of female exogamy.' This is interesting, and a similar conclusion was

reached for the tomb at Hazleton North (Fowler & Olalde et al 2022). In that case those exogenous women who we know reproduced with lineage men were all from the first two generations, forming founding ancestors for the sub-lineages, but that does not seem to be the case at Gurgy. Do the authors have a view on why some exogenous mothers and not others were included across several generations?

13) We think that the pattern in Hazleton North is not that clear either: one mother is missing in the first generation, and only one is buried at the site in the second generation, and then none after that, although individuals from only one sub-lineage are still buried there in generations 3 and 4.

At Gurgy, exogenous females are buried at the site in generations 3 to 5, but not systematically either. One possibility could be that exogenous females return (or are returned) to their group of origin according to certain rules, which could explain why also some adult daughters are buried at the site, who might have (been) returned to the community to be buried. Alternatively, it is also possible that some of these women were granted a different funerary practice that is not readily visible in the archaeological record. At this point, however, despite the observed systematic bias, and potential explanations, the actual reasons for the clear deficit remain elusive. We nevertheless modified the last sentence of this paragraph, which now reads: "We thus observe a potential sex-bias in burials independent of female exogamy. This could be explained by different funerary practices being reserved for these mothers, or by other social factors mitigating against a co-burial with their reproductive partner's group."

L229-31: Is it possible to phrase this in a way that gives the reader a more explicit description of what the differences were?

14) We rephrased and developed this section, which now reads: "Published stable isotope data (carbon, nitrogen, sulphur) measured on bones highlight a significant, sex-based dietary division in adults²¹. On average, males yielded higher $\delta^{13}\text{C}$ and $\delta^{15}\text{N}$, and lower $\delta^{34}\text{S}$ values than females, which could reflect a separation by sex, but could also be a signal of female mobility (Supplementary table 24)."

L232-236: Can a schematic image be produced to illustrate the specific changes in grave goods by age at the site (e.g. similar to that used by Sofaer Derevenski 2000, figure 1), or Hofmann (2009, figure 3)? This could be included in supplementary note 14, and could also include reference to the changes in diet mentioned in the text. I am aware similar schematics are published in Le Roy et al. (2018) and Rey et al. (2019): those could be referred to instead, but they are not specific to Gurgy and do not compare sex categories.

15) We created a new figure panel (Extended data Fig. 4c) to explicitly show the changes in grave goods, pit types and body positions for the subadult individuals, based on the new Supplementary table 21. We added information for all burials, not only those which yielded aDNA data, therefore the genetic sex is not available for all individuals. No sex-bias could be observed according to grave goods, pit types or body positions.

We also produced a new plot (Extended data Fig. 4d) which shows stable isotope results for female and male subadults. Here, we find that subadult females and males form two different clusters and discuss these new results in Supplementary Note 14. We also modified the main text which now reads: "Published stable isotope data (carbon, nitrogen, sulphur) measured on bones highlight a significant, sex-based dietary division in adults²¹. On average, males yielded higher $\delta^{13}\text{C}$ and $\delta^{15}\text{N}$, and lower $\delta^{34}\text{S}$ values than females, which could reflect a separation by sex, but could also be a signal of female mobility (Supplementary table 24). Genetic sex determination of subadults allowed us to confirm this difference also in childhood ($p=0.01019$), which could be explained by a sex-related differential treatment at certain ages, determined by social rules (Supplementary Note 14, Extended data Fig. 4)."

L268-295: It does seem extraordinary that there are no cases of half-siblings among the burial population. The interpretation presented seems plausible, but is it also worth considering the possibility that there were taboos against, or significant sanctions against, engaging in sexual intercourse with multiple partners?

16) We agree that other explanations are possible. Of course, social taboos or sanctions might have existed against relationships outside of the primary partner, but these sanctions could also be extended to offspring from such unions. We modified the beginning of the paragraph accordingly, which now reads: "This indicates that polygynous reproductive unions were uncommon or perhaps socially proscribed, or that the burial of offspring from such unions was carried out elsewhere. It also suggests that serial monogamy including levirate and sororate unions (in which a woman re-partners with her deceased husband's brother or a man re-partners with his wife's sister) were rare, in contrast to recent findings from a later Neolithic long cairn in England⁸."

L333: 'strontium isotope values suggest a move of each generation to another geographical location': The supplementary note 11 is very clear on this topic, and the novel inference that this suggests shifting agricultural plots from which food was sourced is convincing, but there are questions about some of the interpretation of social units that seem to rest on this evidence. P49 of the supplementary document suggests 'it is possible that individuals from a given family branch and close generations

share similar diets, resources, and environments, demonstrating resource management strategies at the sub-family level'. This makes sense, but I am concerned that the references to 'nuclear families' in the article gives a misleading picture of what can be said about 'sub-family' (or, I would suggest, 'sub-lineage') groups. Are the Sr results also consistent with an interpretation that the whole community sourced their food from shared locales and when they shifted sources, they did so together? Is it even possible then that similar results within a generation (or two) tell us something about which individuals grew up contemporaneously when the group was living in one specific locale and drawing its food from the same range of locations? For instance, could there be an age gap between (a) siblings 255, 224, 212 and (b) siblings 213, 317, with an episode of relocation between the periods in which the sampled teeth of these two groups of siblings were forming? And could the similarity in Sr results suggest that 317 was contemporary and co-resident with 250 (and the same for 276 and 245B with 255, 224 and 212), growing up in the same locale (or at least eating food from the same locale)? The spatial clustering of these burials does not suggest a close relationship between 317 and 250, compared with 317 and his siblings, so if we do infer that proximity in the cemetery relates to social proximity there is no suggestion that 317 and 250 were socially particularly close (same for 276, 245B and 255) – the question is rather whether these distant biological relatives with similar Sr readings from the same generation resided in the same locale at the same time. Overall, could the authors clarify here whether they are suggesting (a) one large residence group relocating repeatedly, rather than (b) a series of dispersed sub-lineages residing separately from one another and each moving periodically?

17) We agree with the statement concerning the use of the terms "nuclear family" and "sub-family" and have changed it to "sub-lineage", which is more precise and technically correct.

As far as the strontium results and interpretations are concerned, there is an important aspect to take into consideration. The Sr data have been obtained with the laser ablation MC-ICPMS technique/method, which means that the $^{87}\text{Sr}/^{86}\text{Sr}$ ratios were measured along the profile of the tooth, obtaining several thousands of values for each individual. Details of the method are described in Supplementary Note 11 and all the obtained profiles are provided in Extended data Fig. 8. The data shown in Extended data Fig. 3b with the colour gradient in the pedigrees represent the average value of all values for each individual. Therefore, the variation of average values between individuals that we can observe in the Extended data Fig. 3b must be considered carefully and referred to for every individual profile. What we can observe in Extended data Fig. 8 is a high diversity of profiles, and no common pattern that could show similar individual trajectories. For instance, if GLN212, 224 and 255 (mentioned by the reviewer) do show a similar profile that could be interpreted as contemporaneity and co-residence for these three individuals, other individuals showing similar average values such as GLN213 and 317, GLN220, 241 and 260, GLN243A and 268B, do not show similar patterns in their profiles. Another important parameter is the narrow time

window given by the values, corresponding to the time of tooth formation (2nd molar (~2.5-8.5 years old) when possible, 1st molar (~0.5-3.5 years old) otherwise). To compare the profile patterns, these would need to be in perfect synchronicity. At present, the chronological resolution is too crude and there is no isotopic 'calibration curve' that would permit the correct temporal alignment of individual profiles. Moreover, it has been shown that dental growth is highly variable, which increases the complexity of matching potential similar curves to estimated ages (Scharlotta et al, 2018, "Mind the gap" - Assessing methods for aligning age determination and growth rate in multi-molar sequences of dietary isotopic data, AJHB, e23163). Therefore, it is not possible to detect contemporary groups and to define the kind of mobility that was practised by the individuals buried at Gurgy. We refer to these limitations in the two last sections of Supplementary Note 11.

L344 (and use of the term in L852, extended data fig 5 caption): 'emphasis on "nuclear" family units': As above, I am less convinced about this interpretation than the rest of the article because the authors have not presented a clear case for who resided with whom at a household level. This might just be a matter of terminology: it is not clear to me how the study can distinguish a nuclear family from an extended and/or cross-generational family, but I think a clear case has been made that the community traced descent at the sub-lineage level so 'sub-lineage' would be a more appropriate term. The correlation between grave proximity and biological relatedness (including reproductive partners) is sound, but if the implication is that this indicates separate nuclear family residences in life then I would point out that overall the graves are very close together in the same nucleated burial ground: can it be demonstrated that the living contemporary kin did not also reside as one or two extended co-residential groups?

18) We agree that the term "nuclear" is overstated in the frame of our data and we replaced it in the relevant sections of the main text and in Extended data Fig. 5 caption. We do not know whether the Gurgy community in life was organised as households per sub-lineage or extended and/or cross-generational family differentiated per sub-lineage. On the basis of the estimated sizes of the pedigrees, the number of offspring reaching adulthood per reproductive union, and good overall health status of the group, we hypothesise that this was likely promoted by a spatial co-residence of numerous reproductive unions, as supported by similar strontium values yielded by individuals from different sub-lineages at similar generation levels (see point 17).

L350-358 and supplementary note 14.1, p46: The conclusion that those buried at Passy tombs were the elites while those in cemeteries such as Gurgy were non-elites is not the only possible interpretation, and the comparison across these sites deserves further discussion given the complexity of the regional situation and the various

cultural affinities attested at Gurgy, as outlined in Supplementary note 1. The recent aDNA study of Fleury-sur-Orne demonstrated that only one or two representatives from each lineage were selected for deposition there (one of which was female) (Rivollat et al. 2022) – but we do not know where their relatives' remains were deposited (if they were at all), so we do not know if their kin should be characterised as an elite community. By comparison, the Gurgy results illustrate wider social representation. It seems possible that under certain circumstances a representative of such a community could potentially be selected for special treatment at a different site to the family burial ground, such as a Passy type burial ground. This need not divide society into distinct sets of elites and commoners with status inherited by all members of a lineage across multiple generations. We also know lineage ancestor 270B was given some unusual treatment at Gurgy, illustrating differential selective treatment could occur. Are the burial patterns and aDNA results from Fleury-sur-Orne, part of a cluster more than 200 km north of Gurgy, thought to be typical for the cluster of STP sites local to Gurgy? Does the radiocarbon modelling shed light on whether Gurgy and local Passy burial grounds were contemporary? How much variation was there in terms of the numbers of grave goods or how exotic they were between the individuals at Passy sites, compared to the degree of variation between burials at Gurgy? Several of the graves at Gurgy include scaphopod shells, and there are examples of beaver incisors, deer bones and pig bones – are these not equivalent to the grave goods made from wild species from Passy burial grounds? I note that 270A was buried with three arrowheads – were these the same kinds of arrowheads sometimes found in Passy graves? Either way, could 270A and 270B have held roughly equivalent standing in the Gurgy community to people buried elsewhere in Passy tombs – heads of lineages, perhaps? Could the Gurgy community (characterised here as healthy, co-operative and connected with other communities) have rejected (or predated) the more hierarchical relations or funerary practices developed by others in the region? Are the following all possible, and if not what is the evidence that supports the first one above the others: (a) an elite/non-elite distinction encompassing Passy sites and Gurgy, (b) a society generally using local burial grounds but which might infrequently deposit selected people at shared Passy necropoli and (c) two kinds of burial grounds used by neighbouring communities with different attitudes towards social ranking and inclusiveness of burial. The final section of supplementary note 14.1 might be the place to address some of these questions, supported by a little edit of these lines in the main text.

19) Thanks for this very constructive comment.

Although the site Fleury-sur-Orne belongs to the Cerny cultural horizon, the STP (structure de type Passy) in Normandy are not exactly the same as those in the Paris Basin. First, out of the four sites (Blainville-sur-Orne, Les Rots, Cuverville, Fleury-sur-Orne) detected in Normandy thanks to aerial survey, Fleury-sur-Orne is the only one which has been excavated extensively. Therefore we do not know how representative the site was with regards to regional specificities. The Fleury monuments are earthen

long barrows, some measuring up to 300 m in length, which makes them the longest in Europe. They were built for a single individual, sometimes also two, highlighting the very specific status of the individuals buried there. In our recent study on Fleury (Rivollat et al. 2022) we could show that a) almost all the individuals were males (13/14), b) the pairs of individuals buried in the same monuments/graves were father and son, and c) that there was no close biological relationship between individuals from the different monuments. We argue that the combined data and documented father–son line of descent suggest a male-mediated transmission of sociopolitical authority.

Overall, the situation at Fleury is different from what is described from the Paris Basin (Chambon and Thomas 2010). Here, while monuments also contain a single axial grave, up to 8 or 9 additional graves can be found in the same monument in some sites (e.g. Balloy). The high status of the individuals buried in the STP is also visible, mediated through the size and architecture of the monuments, and the associated grave goods, notably the arrowheads, which are exclusively associated with males, and 'Eiffel Tower' spatula, made from a cervid scapula, that are found only with the male central figure of these monuments. The crucial difference to the STP sites in Normandy is the demographic profile: the majority of burials in Fleury are males, while the Paris Basin STP sites contain burials of adult males and females, as well as subadults, thus showing a wider representation of the groups. The Paris Basin STP sites are also more variable in the funerary organisation and the funerary scenography can be very specific. Overall, the STP sites from the Paris Basin show both a hierarchy in the funerary context and a transmission of the status.

The relative contemporaneity of Gurgy (Bayesian modelling of dates spans 4700–4600 BCE) with STP (~4700–4300 BCE) and other sites without monuments known in the region described in the Supplementary Note 1 (~4800–4000 BCE) are consistent with the parallel existence or sufficient temporal overlap of that all these sites. In this regional context, several arguments lead us to consider Gurgy as likely representing the commoners of such a society. First, Gurgy has a very similar demographic profile as the groups buried in the STP, including adult and subadult female/males. The lack of genetic data from the individuals buried in STP contexts does not yet allow a comparison of the kinship structure and social organisation of the STP group. Second, none of the elements present in STP sites which emphasises a hierarchical structure (monuments, gender-related scenography, grave goods) are visible in Gurgy, except perhaps for the larger sizes of the graves of GLN270B's son (GLN237A) and grandson (GLN221B). However, this is far from the ostentatious demonstration visible in the STP. It is true that GLN270B has a very specific funerary treatment compared to all other individuals buried in Gurgy, but no other sign of power/wealth in the material culture was found. We suggest that it was important for his community/family to (re)bury him in the very place where his descendant will be buried for a few generations, but not at the level of supra-regional significance outside the community, as no externally visible signs of representation were found.

Elements of Gurgy grave goods are similar to those found in the Cerny, but also in the Grossgartach, Rössen, Chasséan and in the Chamblande cultures, which demonstrates their ubiquitous character. This does not allow for a strict cultural attribution of the Gurgy site to the Cerny culture and therefore for us to make parallels with the grave goods found in the STP, but it does show that the region of the Yonne has received diverse contemporary influences.

[Redacted text]

On the basis of the current evidence, a tentative interpretation would thus lean towards the scenarios (a) and (c) as suggested by the reviewer. Scenario (b) is still possible, but difficult to assess without genetic data from STP-associated individuals (this is a work in progress as far as we are aware). Scenario (c) is possible but difficult given the uncertainties in assigning Gurgy to the Cerny culture (paucity of diagnostic grave goods). However, the strict distinction according to material culture does not fit well with the genetic evidence of a wider connected network, as we alluded to in the conclusion and in Supplementary Note 14.

We acknowledge that our text lacked some of these elements and discussions. We modified the main text slightly, but due to limited space refer to the revised Supplementary Note 1 which now contains an extended description of the STP regional context, as well as Supplementary Note 14.1 in the section “Inferences beyond the site” for a deeper discussion of these aspects. We also added a map in the Supplementary Note 1 to help the reader to follow our description.

L358-62: This needs editing for clarity, I can't follow the point (though supplementary note 14 explains it clearly).

20) We agree the sentence was not clear and rephrased it as follow: “As the site was used by a single group composed of two distinguishable pedigrees, we would expect contemporaneous graveyards of similar sizes to represent lineage groups as well, but the number of such sites in the area is much smaller than we would expect for a representative cross-section of the population.”

Images:

Figure 1c: Can the colour coding for grave 270 be altered to show that 270A is ‘no data’; same for 231B?

21) We modified the colour coding for all the double burials in the layout of Figure 1c and Extended data Fig. 6 to visualise the contextual information for both individuals.

Supplementary data:

The extended data figures are extremely useful, but there are several different tables and images that an archaeologist needs to navigate to piece together the basic contextual information for each individual, and some of that data is not present. Can Table 1 be modified to include columns providing data on body positioning and orientation, and information about the positioning of the grave goods in relation to the body? Combining the isotopic results into this table (including the previously-published Carbon, Nitrogen and Sulphur as well as the new Strontium data) and identifying which lineage and generation each individual is assigned to would make the table of the most use to archaeologists. Table 1 currently does not include unsampled individuals buried at the cemetery, but it would be useful to include the archaeological and osteological information for them too, especially 231B and 207A given their importance to the analysis. Osteological data, including on pathologies, would be useful given the statements about conditions supporting a good rate of survival into adulthood (and a few lines discussing this would also improve the supplementary notes). If the Sr data is to be kept in a separate table then it would be useful if a column could be added identifying the sex of each individual and their place in the lineages (e.g. lineage A or B, which generation, and whether they were endogenous or exogenous to the lineage).

22) We modified Supplementary table 1 by adding more information. Specifically, we now also included all non-sampled individuals, added the pedigree/sub-lineage/generation information, the osteological sex, and the “exogenous” status as inferred from the genetic pedigrees. In addition, we added a new Supplementary table 21, where we provide details about archaeological features such as the type of burial, structure of the pits, body orientations and positions, and grave goods. The position of the grave goods within the grave is difficult to encode in a table format, given the number of different types of goods for some individuals. The full catalogue of grave descriptions will be part of a planned monograph in the near future. Unfortunately, it was not possible to assess the pathologies on the bones in a reliable and systematic way given the rather poor macroscopic preservation.

We decided to keep all of the Sr isotope-related information in table 23, as there are numerous values that are difficult to summarise in Supplementary table 1. Here, for the purpose of cross-referencing, we also added the individual information such as genetic sex, age-at-death, pedigree, sub-lineage and generation. We provide a complementary table (Supplementary table 24) including carbon, nitrogen and sulphur isotope data

(published in Rey et al 2019), which was incorporated in Extended data Fig. 4d (see point 14 for more details).

Supp note 1:

P4: 'double but mainly simple' – I think simple should be single.

23) Thanks. This is fixed.

It would be very useful to include a map here marking up the distribution of STP mounds and the other funerary sites mentioned in this section in relation to Gurgy.

24) We acknowledge that the funerary context in the Paris Basin and Normandy during the Cerny period is complicated. Therefore, we now include a map as suggested to provide a general overview alongside the description in Supplementary Note 1.

Suppl note 11:

What criteria were used for selecting individuals to be sampled for Sr – i.e. were all individuals with the relevant surviving dentition sampled?

25) The Sr analysis was initiated in the framework of other isotopic analyses of the Neolithic in the Paris Basin led by Léonie Rey (see e.g., Rey et al. 2016, 2019, 2021). The teeth were first selected according to reliable results obtained for carbon and nitrogen analysis (evidence of good collagen preservation). The corpus of data has then been completed according to the genomic results and the reconstructed pedigrees, and we analysed additional samples to fill the gaps in the pedigree A, when possible. The 2nd molars were systematically sampled when available, and the 1st molars as a second choice, mainly for young subadults. To clarify this approach, we added a sentence in the first paragraph of Supplementary Note 11, which reads: "The selection was made on the basis of the availability of the teeth for this type of analysis (macroscopic preservation, age at death), previous evidence of collagen preservation^{32,36}, and the position of the individuals in the reconstructed pedigrees."

Suppl note 14:

P42: Perhaps caution could be voiced here about absent people (especially sons) from generations 2 and 3 given that some graves central to the burial ground did not provide aDNA results. Given the spatial analysis suggests that children were buried near parents, graves near to 270 and 237 could have held the sons of those individuals that the text implies were missing from the burial ground.

26) Indeed, we have rephrased this section, and it now reads: "In fact, the importance of the main paternal lineage can also be traced in the subsequent generation directly following GLN270B. The two largest graves at the site were built for his son GLN237A, the only individual of his offspring who yielded DNA data, even though he must have had at least two more sons as inferred from the pedigrees who might have been buried at the site, but for whom no DNA data were obtained, as well as his grandson GLN221B, i.e., the son of GLN237A (Fig. 1, Extended Data Fig. 6b)."

P45: I suggest removing the term 'patriarch' here. While he may have been recognised in life as a lineage founder, we do not really know what social standing or authority that individual had in life: his importance may have grown only through the achievements of his descendants, for instance, some of whom seem to have relocated his remains.

Should 288-289B be added to the list of exceptions to the patrilineal scheme discussed in suppl. Note 14?

27) We agree and have removed the term "patriarch".

We added a discussion about GLN288 and GLN289B in this section, that we developed more. It reads now "Another exception to the observed predominant form of social organisation, inferred from the Gurgy funerary community, can be found in Pedigree B, and concerns female individual GLN288 and her daughter GLN289B. As the parents of GLN288 are missing, we cannot determine whether GLN288 is linked to the pedigree through her mother or her father. If she was linked through her mother to the main lineage of Pedigree B, we cannot exclude that the mother is missing because she had left the group to join another group as hypothesized for most of the other females from Gurgy, while her daughter GLN288 later returned to the group, then considered as an 'exogenous' partner. If she was linked through her father to Pedigree B, the main lineage is continued by one more generation. Irrespective of the two possibilities, GLN288 and GLN325 represent adult daughters, who had adult offspring buried at the site, which gives both of these individuals a special status. By contrast, the daughters GLN289B, GLN212 and GLN213 from Pedigree A represent the only three adult daughters without offspring. If we were to apply a strictly patrilocal and female exogamic system, these three should also have left to another group, therefore their presence at the site is also part of the exceptions observed in this cemetery."

References:

Hofmann, D. 2009. Cemetery and settlement burials in the Lower Bavarian LBK. In D. Hofmann and P. Bickle (eds), *Creating communities. New advances in central European Neolithic research*, 220-34. Oxford: Oxbow

Sofaer Derevenski, J. 2000. Rings of life: the role of early metalwork in mediating the gendered life course. *World Archaeology* 31: 389-406.

Reviewed by Chris Fowler

Referee #2 (Remarks to the Author):

The manuscript of Rivollat and co-authors "Extensive pedigrees reveal social organisation of Neolithic communities" reports on genome-wide, strontium isotope and contextual data from >100 Neolithic individuals from the site Gurgy 'les Noisats' (Paris Basin, France), to reconstruct kinships and social organization in this Neolithic community. One of the most interesting aspects is the study of "non-elite people" from a common graveyard of the European Neolithic. Indeed, as the authors highlighted, studies from this period are commonly focused on specific funerary contexts, implying select groups or individuals of specific status.

The questions are interesting and can be of great interest to the general readers in several disciplines. In my humble opinion, this paper is a very good candidate for Nature. Abstract, introduction and conclusions appropriate and fully meet Nature criteria.

The dataset is impressive, including more than 100 genetic data, 8 new radiocarbon dates, and high spatial resolution strontium isotopes data from 57 individuals. Procedures have been well described, and appropriately administered and interpreted. The reporting of data and methodology is detailed and transparent to enable reproducing the results of the research.

The conclusions are valid and reliable. Every aspect of different possible interpretations has been considered and the conclusions are strongly supported by the data interpretation.

We thank reviewer 2 for their valuable feedback.

Here following some minor comments.

In Figure 1d, is reported a photograph of the grave 270 with two individuals, A and B. While in the caption 270B is described as main male ancestor of pedigree A, it took me a while to understand that ID 270A didn't get enough coverage to be included in the pedigree, and this is mentioned only later in the main text. I would suggest specifying this in the caption.

28) Thanks. We added this in the caption and also updated the colouring of grave pit 270 to reflect the fact that genetic results could only be retrieved for one individual (see also our response to point 21).

Line 193: "Notably, four of these six females (GLN212, 213, 277 and 289B) had no known offspring, although having reached reproductive age".

In my opinion, this sentence can be misleading. No offspring of these female adult individuals have been found in the burials, and the reasons (they had no children, the children have been grown and moved elsewhere, the remains are poorly preserved in terms of odontoskeletal preservation and DNA coverage) cannot be known in any way.

29) We agree. This sentence was moved further down in the text and was updated accordingly. It now reads "For four out of the six adult lineage daughters (GLN212, 213, 277 and 289B), who remained at Gurgy, no offspring could be identified at the site even though they had reached reproductive age. Female exogamy may not have been practiced strictly, or alternatively, these lineage daughters could be reproductive partners of unlinked adult males (with no offspring linking them to the pedigrees): a scenario which further complicates the assumption of strict patrilocality and female exogamy. Alternative reasons for their stay in the community remain elusive."

In Supplementary Note 13, paragraph “87Sr/86Sr intra-profile variation and individual life histories”, reference #84 is clearly wrongly assessed. Being aware of the difficulty in managing such amount of Supplementary materials from different authors, I suggest the authors to check the reference in the Supplementary notes.

30) Thanks for noticing. Indeed, references 84 and 85 were assigned wrongly. We have now fixed this and have double-checked all other references.

Lines 279-280: "Including unsampled inferred adults, we observe two cases of up to six children from the same couple (Extended Data Fig. 6)".

Is the reference to Extended data fig. 6 correct?

31) Yes, panel d in Extended Data Fig. 6 shows the number of (adult) children per couple. We modified the sentence to make it clearer and it now reads “Including unsampled, inferred adults, we observe two cases of up to six offspring from the same couple (Extended Data Fig. 6)”

We also modified it in the Extended Data Fig. 6 and the associated caption.

Referee #3 (Remarks to the Author):

Rivollat et al. present ancient DNA sequences from individuals buried in Gurgy, France, including genome-wide data for 94 individuals (72 new), mitochondrial DNA (mtDNA) for 100 individuals (78 new), and Y chromosome data from male subjects. Remarkably, using relatedness detection software designed for ancient DNA, the authors were able to reconstruct two large pedigrees—the larger spanning 7 generations—that include most of the 94 subjects. From these pedigrees, they find strong evidence for patrilocality and female exogamy; for example, they found only two Y haplogroups in the study subjects versus 36 distinct mtDNA haplogroups. Furthermore, adult female descendants in the pedigrees are strikingly few, suggesting that there may have been networks of groups that also practiced female exogamy to which these women migrated. Distant relatedness detected between some married-in females lends support to this theory. Another striking feature in the pedigree is a complete absence of half-sibling relationships, implying this community practiced monogamy, with no evidence even for serial monogamy from the data. Finally, the authors argue that this burial ground, which lacks the monuments present in nearby contemporary burial

sites, may give a window into the practices of non-elite individuals at this time period (4300-4700 BCE, the Middle Neolithic).

We thank Reviewer 3 for the constructive comments and the very detailed review of the text and the supplementary Notes, especially on Supplementary Notes 2 and 3. This has been very useful to spot the residual errors.

One concern relates to the PMR method for evaluating relatedness. It makes sense methodologically but hasn't been validated to the same degree as the other relatedness methods. For example, might it be biased towards calling pairs as more closely related than they really are? Given that several somewhat ambiguous relatives were judged as truly second degree on the basis of the PMR approach, running some analyses of this using simulated or perhaps real data seems important. Are its probabilities well-calibrated? Use of the PMR statistics seems only to occur in questionable cases; is it worth checking all classifications with the PMR approach?

We now provide the citation for/refer to the preprint by Rohrlach et al. titled "BREAD: Inference of Biological Relatedness from Ancient DNA" (<https://github.com/jonotuke/BREAD>), which describes the method and evaluation in detail. In brief, we model the number of overlapping sites which mismatch (i.e., the PMR) using a binomial distribution. For each pair of individuals, we calculate the probability that they share the observed number of mismatching sites, given that they are either twins/same individuals, first-degree related, second-degree related, or "unrelated" (i.e., third-degree or more distantly related). Using these probabilities, we find the "most likely" degree of relatedness, as well as calculate the posterior probability of the other degrees of relatedness. We compare the performance of our method to READ, and find that for low-coverage (0.04X) simulated data, we correctly identify the degree of relatedness in 129/136 of pairs, and only misclassify the degree of relatedness when either (a) (2/7) READ also misclassified the data (hence statistically abnormal pairs), or (b) (5/7) the pair was third-degree related, but BREAD classified them as second-degree, however READ also returned the relationship as "not significantly different to second-degree relatedness". We then tested our method on empirical data from the Early Bronze Age site of La Almoloya, in Spain (Villalba-Mouco et al. (2021)), and compared our results to those found using READ from the manuscript (which also employed IcMLkin, IBD and contextual data). BREAD agreed with the findings of the study in 2276/2278 (99.91%) pairs. The remaining two pairs were assigned as unrelated by READ, but where we found a second-degree relationship. In one case, additional information yielded a third-degree relationship was true, and the second case had no additional information, and remains uncertain.

We systematically used the method for every single pair of individuals in the Gurgy dataset and all plots are available on the online repository Zenodo following this link: <https://doi.org/10.5281/zenodo.7224898>. We acknowledge the fact that the access is restricted to the authors and reviewers and that keeping the anonymity of the reviewers might be preventing them from getting access. We hope access might be possible through the editor. The link is provided in the text (Method section and Supplementary Note 2).

Another simpler question is whether the median P_0 value used with READ is downwardly biased because the data are so enriched for relatives. Do any of the relatedness signals shift if you calculate the median P_0 value with male pairs removed?

First, we use the median P_0 as it is, by definition, resistant to outliers and hence downward bias. It would be required that the overwhelming majority of pairs of individuals were first- or second-degree related for us to observe a qualitative change in our results when removing these pairs. Nevertheless, we calculated the median with all pairs (0.24530753), and then when removing all first-degree related pairs (0.24538174) and both first and second-degree related pairs (0.24549929). In these cases, there was only a 0.03% and a 0.078% reduction in the observed median P_0 when removing the first- and second-degree relatives, respectively.

Besides the pedigree, I wonder if the results from strontium isotope analyses may be a bit overstated. Figure 1e plots these isotope values, with a fitted line that increases with generation, but are the differences e.g., between adult females and adult males statistically significant? I didn't see any statistical tests for the claims here. The same concerns apply to the carbon, nitrogen, and sulfur isotope analyses which seem not to include p-values.

32) Indeed, we did not provide any statistical test for the strontium analysis. Given that we are interested in differences between males and females in each generation, we ran a multivariate linear regression, including a full interaction model including the three variables: generation, sex and age category. We used ANOVA (p-value cutoff of 0.05), and further confirmed using stepwise AIC, to compare the residuals of the nested models, and we found that the model which fits best is a model with generation and sex, as well as an interaction term between sex and generation, as predictor variables. However, the age group variable could be completely removed from the model. The diagnostic plots for the models were all inspected to ensure that the assumptions of linear regression were satisfied.

The coefficient of the interaction term "male:generation" is negative, meaning that per generation, the strontium ratio for females was increasing significantly faster than it was for males (p=0.01474). We added this last p-value in the caption of Figure 1, which now reads "A significant difference between sex can be observed per generation (p=0.01474)."

Stable isotope data for adult individuals have been analysed and reported in the study by Rey et al. 2019 (reference 21), in which they show significant differences between adult males and females (Wilcoxon test, p < 0.05). We added the word "significant" in the sentence in the main text to point this out alongside the citation, which now reads: "Published stable isotope data (carbon, nitrogen, sulphur) measured on bones highlight a significant, sex-based dietary division in adults²¹."

For the carbon and nitrogen analysis, we obtained new information compared to the previous publication by Rey et al. 2019 as we could now include the genetic sex determination of the subadult individuals. As explained in response 15) above, we also provide a new plot (Extended data Fig. 4d) showing these results, and ran a cluster comparison using the Calinski-Harabasz Index, to calculate the ratio of the sum of between-cluster dispersion and of inter-cluster dispersion for all clusters. The empirical p-value, calculated using a permutation test with all $\frac{17!}{9!8!} = 24,310$ permutations of the possible cluster labels, is significant (p=0.01069), and remains significant when including the "undetermined" individuals. However, we excluded three outlier individuals GLN232C, GLN245A and GLN326 from this analysis due to their young age, who carry very high $\delta^{13}C$ values, which likely represents a breastfeeding or weaning signal. We provide this information in the caption of the new Extended data Fig. 4d, which reads: "Stable isotope data (carbon and nitrogen) published in Rey et al 2019, plotted with the genetic sex of the subadult individuals provided by this study (less than 20 years old). Two significant clusters can be observed between males and females (p=0.01019). Two females (GLN232C, 2-6 yo, and GLN326, 2-5 yo) and one male (GLN245A, 2-6 yo) are outliers in the upper part of the plot. They are the youngest individuals of the analysis and the high values of $\delta^{15}N$ might be a signal for breastfeeding or weaning time."

The main text mentions that the female diversity in this group is visible from the inferred phenotypes, but the cited Supplementary Note (9) does not go into detail on female versus male phenotypic diversity.

33) Thanks. We agree that this sentence was unclear. We did not mean a difference in diversity between males and females, but a considerable phenotypic diversity in the group overall, which we explain on the basis of the unrelated female newcomers in every generation. We rephrased this in the main text, which now reads: "This female diversity within the Gurgy group also explains the overall phenotypic diversity of the group (Supplementary Note 9, Supplementary tables 15 and 16)." We also added a

more detailed explanation of the phenotype results at the end of the Supplementary Note 9: "Overall, the individuals from Gurgy represent the full spectrum of variation in skin, hair and eye color pigmentation, ranging from 'Blond/Dark-Blond' to 'Black' for hair colour, including ten individuals with red hair, and from 'Very pale' to 'Dark-Black' for the skin pigmentation. The eye colors were also variable, including blue (N=23) and brown (N=42)."

Minor:

Caption to Extended Data Fig 9c: "Individual GRG095 shows an inbreeding signal similar to a first-cousin union." - Should this say GLN282? Notably the main text and other places suggest this person is the offspring of a second or third-cousin union whereas this caption seems to say first-cousin. Also pgs. 44 and 45 of the supplement uses first-cousin again. The Ext Data Fig 9c plot seems most similar to the first cousin union and quite different from the second and third cousin plots. Perhaps it's worth reconsidering this classification?

34) Yes, it is indeed GLN282. We fixed the individual name in the caption and double-checked for the discrepancy in the text. According to the Extended data Fig. 9, this individual shows indeed a ROH signal consistent with him being the offspring of a first cousin union. However, GLN282 carries two ROH of ~20-22 cM, therefore this individual is more plausibly the offspring of 2nd or 3rd cousins. We added this explanation in the caption to Extended data Fig. 9c, and updated the main text and Supplementary Notes, accordingly.

Detailed:

Line 87: typo - "wide-genome"

35) Thanks for spotting the typo. Fixed.

Paragraph starting on line 122: I know the IBD method isn't published, but it seems worth citing something; even the webpage helps.

36) During the revision phase, the pre-print describing the ancIBD method has been released online. We now cite the pre-print (Ringbauer et al., 2023, bioRxiv), as well as the webpage in the Material and Methods under the section titled "Imputation and screening for IBD sharing" and in Supplementary Note 3. We think it is not necessary to

add the link in the main text, but we added both of the references to Material and Methods and Supplementary Note 3 in the text.

Line 202-3: “could explain some missing links between the isolated adult females and the rest of the pedigrees” - I didn’t quite follow this sentence.

37) We rephrased this sentence, which now reads: “Consistent with this pattern, additional links observed between the isolated adult females and the pedigrees could be due to: (a) distantly related females stemming from the same community; or (b) women who left the Gurgy community in previous generations having female descendants that subsequently returned to Gurgy. The latter scenario is indicative of reciprocal exchange typical in moiety systems³.”

Line 248: “structured by a range of group features” - it might be nice to reword this given the term structure in population genetics has a specific meaning that is perhaps best to avoid confusion with here.

38) Thanks. We replaced the word “structured” by “driven”, and the sentence now reads: “Taken together these results suggest that the Gurgy community maintained a fairly clear pattern of female exogamy that may have been driven by a range of group features (e.g., population size, resource access, network position) or identities (e.g., linguistic or cultural affinities).”

Line 259: this might read more clearly if worded as “siblings GLN243A and GLN268B share the same mitochondrial haplogroup H1 as GLN315”.

39) Thanks. We agree and have changed the sentence accordingly.

Line 275: what does “shortfalls” mean?

40) We mean shortfall in the sense of deficit, i.e. imbalance of the female/male ratio. We revised the sentence to make this clearer, now reading: “We find this observation surprising given potential imbalances in the female/male ratio, e.g., an elevated risk of

death from complications during childbirth (for females), potential conflicts or diseases in prehistoric societies.”

Line 292: since the ROH analysis uses varied lengths of segments, should the word “long” be deleted here?

41) We agree. Indeed, as it is the overall distribution of the various lengths of ROH that characterises the background relatedness, we removed the word “long”.

Line 515: what is YMCA?

42) We meant the Y chromosome capture described in reference 43. This is made clearer now.

Line 555: are citations 52 and 53 appropriate here?

43) Yes, they are. Patterson et al. 2012 (Citation 53) report Human Origin Affymetrix data of modern-day individuals around the globe. The study by Lazaridis et al. 2016 (Citation 52) generated and published 238 new individuals, which were added to this dataset. We used individuals from both publications in our principal component analysis (PCA).

Line 572-4: I may be misunderstanding, but this seems to say that the reads are mapped with both samtools and EAGER?

44) Indeed, this is a mistake, thanks for pointing this out. We rephrased the sentence as follows: “To process mitochondrial DNA data, we mapped reads from mito-capture data to the revised Cambridge reference sequence⁵⁷ using the circular mapper implemented in the EAGER pipeline⁴⁰”.

Line 581: typo - “and”

45) Thanks. We rephrased the entire sentence, now reading: "To determine biological relatedness we combined two established methods designed for aDNA data: i) READ¹⁴ to detect first and second degrees (Supplementary table 8), and ii) lcMLkin¹⁵ to differentiate between parent-offspring and siblings among first degree relationships (Supplementary table 9, Extended Data Fig. 1)."

Caption to Extended Data Fig 1b: please note the inset here.

46) Thanks. We decided to remove the inset showing the zoom-in on the higher-degree relationships, as it did not reveal any additional, critical information.

Caption to Extended Data Fig 7a: what are the bands around the dashed lines?

47) The bands indicate the 95% CIs around the average P_0 values for adult/subadult females/males. We added this information to the figure caption.

Supplementary table 19: I'm not sure what each of the columns here are; these may be standard, but adding definitions would make this more easier to understand.

48) Thanks for pointing this out. We now provide the definitions of all column headers, as well as the calculations within each cell, to make them clearer.

Minor in Supplementary Notes:

Pg. 15 "Comparing IBD sharing with inferred pedigrees": it might be worth noting what the analysis shows for the outlier parent-child pair that looks like a full sibling pair.

49) We agree and have modified Extended data Fig. 1 to show where the pair GLN285A and GLN285B is plotting in the results from lcMLkin and ancIBD.

We modified the respective paragraph in Supplementary note 3, which now reads: "Considering the first-, second- and 3rd-degree related pairs, the IBD sharing analysis matches perfectly with previous reconstructed biological relatedness found via the output of READ and lcMLkin (Supplementary Note 2), confirming both the reliability of IBD sharing method as well as the robustness of our reconstructed pedigrees. The IBD

analysis also confirms the parent-offspring relationship between GLN285A and GLN285B, which IcMLkin determined as siblings, but which was clearly inconsistent with both the tree reconstruction and the PMR-window analysis (Supplementary Note 2, Extended data Fig. 1a and 1b).

We note that beyond the 3rd degree, overlaps between IBD clusters start to appear, and it is no longer possible to assign a single degree relative IBD cluster anymore (Extended Data Fig. 1b). This is expected given the biological variation of IBD sharing (and consequently all genetic relatedness). However, we can still detect definite recent genealogical links as multiple long IBD are distinctly shared for most individuals up to six degrees apart.”

Detailed:

Pg. 6 defines $r = k_1/4 + k_2/2$, but this is half what is normally called the coefficient of relatedness and is instead the kinship coefficient. It appears that the values in the paper are indeed coefficients of relatedness, with $r = k_1/2 + k_2$. Indeed, a few lines above this formula, the text says, “r equals 1/2 in the case of parent-offspring or siblings” but just after the formula, it says “While $r=1/4$ for both of the 1st degree relationships ...” So there is one discrepancy here.

50) We thank the reviewer for pointing out this discrepancy. In reality, we used the measure of relatedness as defined by the software, but had given the wrong name and value. IcMLkin, reports the coefficient of relatedness (as stated by the reviewer), and we have changed the text to reflect this, and the method in which it is calculated (i.e., 2 times the coancestry coefficient).

Pg. 6 says “the probabilities k_0 , k_1 and k_2 differ ... depending on rate of recombination.” Note that recombination only affects the variance in sharing among siblings (not parent-child pairs) and not the expected values given in the text.

51) Thanks, we concede that this is a wrong statement, and we removed this portion of sentence, now reading “ ..., the probabilities k_0 , k_1 and k_2 differ, and are (1/4,1/2,1/4) for full siblings, and (0,1,0) for parent/child”.

Pg. 7: “we thinned the data such that all sites were at least 200K bases apart” - how did you choose the sites to retain?

52) For each pair: we took all of the overlapping sites, and their relative positions. We started with chromosome 1, and then we took the first overlapping site, and retained it (let this be position n_1^1). We then took the next site that was a **minimum** of 200K sites along the chromosome (at least $n_1^1 + 2 \times 10^5$), and retained this site (denoted n_2^1), and then found the next overlapping site that was a minimum of 200K sites from n_2^1 (at least $n_2^1 + 2 \times 10^5$), and retained this site. We repeat this until we retain the j^{th} until we had no overlapping sites left, i.e., when there is no overlapping site after $n_j^1 + 2 \times 10^5$. We then repeat this “per chromosome” process for chromosomes 2 through to 22. In this way we found a (likely) optimally large set of thinned, overlapping SNPs.

Pg. 8 says GLN226 is a daughter, but the figure shows this subject as male?

53) Indeed, there is a typo pg 8. GLN226 is a male and we rephrased the sentence accordingly, which reads now “GLN225 and GLN226 are two individuals who are 1st-degree related as parent-offspring. Given the young age of GLN226 (1-3 years old) and their shared mitochondrial haplogroup, they must be mother and son”.

The Tree Reconstruction subsection of Note 2 is nicely organized. It would be nice (but may be more work than the authors feel like making) to depict the procedure of reconstructing the pedigree that one can look at while reading. I found myself having to visually hunt in Figure 1 to find the subjects (and searching didn't work as text is not embedded in this - perhaps this could be addressed?). One option would be to outline each of the (say) first degree connections in one figure and add labels then reference these from the text. A second figure could similarly show the second degree links.

54) Thanks. We understand that it can be difficult to track the large number of related individuals. Following the reviewer's suggestion, we generated a new figure, Figure Suppl. Note 2 added directly in the text, which shows the pedigrees resulting from first-degree related pairs only. We used Roman numerals to number each pedigree and used this numbering scheme in the Supplementary Note 2. We hope this makes it easier to follow the description of each smaller pedigree unit. Of note, the actual final pedigree (as shown in Fig. 1 and all other Extended Figures) is the result of the integration of all second-degree related individuals.

Pg. 10: “with the offspring of GLN237A, as expected as he is their” should the second “as” be “if”?

55) Indeed, we modified it accordingly.

Pg. 11 final bullet: although I see that GLN210 cannot be an uncle, the text doesn't make this fully explicit. (Several of the subsequent bullets on pg. 12 also do not fully explain why uncle/aunt relationships are ruled out.)

56) Thanks. We indeed missed a part of the argument, which is now provided for GLN210: "If he was an uncle via the paternal line, he would be expected to be 1st-degree related to the missing father's siblings GLN221B, GLN265, GLN266 and GLN325, which is not the case. If he was an uncle via the maternal line, he would be expected to share the same mitochondrial haplotype with the missing mother's offspring GLN212, GLN213, GLN224, GLN255, GLN317, which is also not the case. Therefore, he can only be a nephew via the missing father's side, because of non-matching mitochondrial haplotypes with the uncles GLN224, GLN255, and GLN317, and the aunts GLN212 and GLN213 (Fig. 1a)."

We made similar modifications to the text for the paragraphs describing GLN207A and GLN279, and subsequently for GLN209 and GLN257. The rationale has also been completed for GLN226.

Pg. 12: "2nd-degree related to the siblings GLN237A, GLN325, GLN265, GLN221B and GLN266, which is not the case" - in fact GLN237A is the parent of the rest.

57) This is correct, we removed "the siblings" as the rationale for the second-degree relationship is still true for all these individuals, i.e, one being the father (GLN237A), and the others the siblings.

Pg. 13, line 2: why can't the person be a grandmother? Age?

58) She cannot be the grandmother because she was indeed only 2-6 years of age and because she does not share the expected 2nd-degree relationship with the full siblings of GLN219. This is made clearer now.

Pg. 13: typo - "un"

59) Thanks, this is fixed.

Pg. 13, first bullet under Pedigree B: should this say that the connection is through the father?

60) Indeed, this has been added.

Pg. 15: citations 40 and 41 seem incorrect.

61) Indeed, citation 40 (Haak et al. 2015), which reports the initial version of the SNP capture targeting ~390k SNPs, was incorrect and thus removed. However, citation 41 (Mathieson et al. 2015) is correct, as it is the first study to report the 1240k SNP capture method.

Pg. 15: Paragraph just under “Comparing IBD sharing with inferred pedigrees” - “we obtain the degree of relatedness”. Nitpicky, but how does this handle full vs half relatives?

62) Thank you for pointing out the case of half-siblings. Relationships via those were indeed not included in our original pedigree crawler program. We had assumed all pedigree connections occur only via full siblings because there were no half-sibling connections in any of the reconstructed Gurgy pedigrees.

To increase our program’s general application range, we have now updated it so it checks for possible half-siblings too (and adjusts the degree of relatedness where necessary), see updated text in Supplementary Note 3 in the section “Comparing IBD sharing with inferred pedigrees”: “Moreover, we store half-sibling relationships: the algorithm checks when the first common ancestor is found whether also this ancestor’s partner is a common ancestor of the same depth.”

We note, however, that since there were no half-siblings in the Gurgy pedigree, none of our results changed.

Pg. 15: typo - “IDB”.

63) *Thanks, this is fixed.*

Pg. 18: “unrelated subadult males” - are these 229 and 308?

64) *Yes, we have added the labels in the text.*

Pg. 20: typo - “ROH>4cM24”.

65) *Thanks, we fixed this typo.*

Pg. 21: typo - “All these females are not or very distantly related”.

66) *We rephrased the sentence: “All of the adult females are either unrelated, or only very distantly related, to each other (Extended Data Fig. 5b and 5d), except for three pairs of individuals.”*

Pg. 29: “WHG” - I don’t think this abbreviation is defined elsewhere.

67) *We changed it back to “Loschbour” as this is the proxy used to model the Western Hunter-Gatherer (WHG) ancestry and do not mention the abbreviation anymore.*

Pg. 42: “Here, the burials of fathers and their subadult male offspring are located significantly closer to each other than any other pairs of individuals (Supplementary Note 12, Extended Data Fig. 6c)” - is there a p-value for this?

68) *There is actually one p-value per pair that we compared (father_daughter - father_son ($p=0.0306$), father_son - uncle_pat_nephew ($p=1.67e-6$), and father_son - uncle_pat_niece ($p=4.77e-7$)), this is detailed in Supplementary Note 12. We added the specific reference to the location of the p-values in this part of the text.*

Pg. 42: typo - “long a time”

69) Thanks, this is fixed.

Pg. 45: "legitimized through the 2nd degree relationship to GLN270B." - is this through GLN311? Could spell this out a bit more.

70) Yes it is through GLN311. We developed this argument, now reading "This was possibly indirectly legitimised through the 2nd degree relationship of GLN311 with GLN270B, which links both lineages early in the tree."

Pg. 47: "and k a correction coefficient worth 10% of t." - not sure I understand this.

71) We modified the sentence now reading "and k, which is a correction coefficient set at 10% of t."

Referee #4 (Remarks to the Author):

General comments:

The authors generated genome-wide DNA data for 94 individuals, 8 new 14C dates, and Sr ratio data for 57 individuals and reconstruct pedigrees to study the social organization of a Neolithic community (please see my comment below about the manuscript title which I think must be edited).

The manuscript is technically extremely sound, and I have no analytical concerns (at least regarding the generation and analysis of DNA). It's fantastic that the entire graveyard was studied, which is one particular strength of this work. The figures associated with this manuscript are also excellent and serve as a great tool for the reader.

I recommend some moderate revisions are made to this manuscript prior to publication.

The primary critiques that I have surround the style of writing and the novelty of this work. First, the body of this manuscript reads mostly as a series of observations instead of a cohesive "story" that is well-supported by the data. I applaud the authors for making so many astute observations, but would encourage them to ensure that the data are being used to clearly support central interpretations -- it feels at present like there are long stretches where they are presenting observations one after the other, which can be exhausting and confusing to the reader.

72) While we agree in parts, and have revised the manuscript with the aim to make it more appealing to the general reader, we are hesitant in venturing into story-telling, as tempting as this might seem. Rather, we feel that we should adhere to the observational side of the spectrum and to only resort to speculations, direct explanations and hypotheses if these are reasonably well supported. While there are a few studies on biological relatedness in prehistoric societies, nearly all of them constitute a different funerary context (burials around hamlets, settlement burials, mass graves, megalithic structures, cairns etc.), which makes it difficult to compare sites. In general, we feel that it is too early to make more general claims about prehistoric societies, but also the statistical support within each site is still relatively low, so this will require a larger number of comparable sites, of comparable size, to arrive at descriptions of broader trends that can shed light on prehistoric kinship practices in, for example, the Neolithic or Bronze Age.

However, having said this, we still feel that there is great merit, insight and novelty in our study, as it showcases, by the sheer number of individual observations, how rich the record becomes once all of the contextual data from each of the participating disciplines has been integrated. As such, we see our study rather as one that is pioneering approaches to studying kinship practices, rather than one that is attempting to close any chapters on the topic.

There is also room for improvement in tying in observations with their implications - e.g., "In addition, only six of the 20 adult females buried at the site are descendants of the main pedigree lines A and B. Notably, four of these six females (GLN212, 213, 277 and 289B) had no known offspring, although having reached reproductive age." now it would be very helpful to hypothesize about what this could mean and tie it in to the conclusions you are drawing. Has this pattern been seen before? If so, where? If not, why not? Why would this be special/important/unique/worth writing about? I tried to point out places throughout the manuscript where I noticed this in particular, although I should also note that there are places where this is done well. I especially thought the conclusion to this manuscript was well-written.

73) We answered this comment following the specific questions below, and modified the text accordingly.

While this paper is technically sound and interesting, I'm not fully convinced of its novelty. Other than being the largest pedigree constructed to date with ancient DNA and focusing on a cemetery instead of a structure like a megalith, what makes this paper exceptionally unique? Maybe this needs to be pointed out more clearly. I think that we are at a point in the field of ancient DNA research where being the largest study (especially in an area like Europe) cannot itself be considered particularly novel.

74) While we are usually not sold on superlatives either, we would like to point out that it is indeed the size of the site and the resulting pedigrees that allows for the richness in observations and the depth of discussion presented in our paper, as well as for stronger statistical support, which - and here we agree with reviewer 1 - can stand as an exemplar for future studies.

A small comment is that there are a moderate number of grammatical errors throughout the manuscript especially in terms of comma usage that should be fixed prior to publication. Some run-on sentence with multiple clauses should also be split for ease of interpretation and I indicated several of them in my detailed comments.

75) Thanks, with the help of the native speakers in our team, we have carefully checked the grammar and comma usage in the main text.

Finally, not having data for GLN270A weakens some of the conclusions that could be drawn about a potentially very important individual (and beyond this individual as well) - is it really impossible to generate sufficient data for this individual? I imagine if it was possible the authors would have done so -- but I would encourage an attempt to improve the data for this individual if one has not yet been made.

76) We are painfully aware of this point. We had made substantial efforts to gain any data for this individual through repeated sampling, DNA extractions and library preparations. Unfortunately, to no avail thus far. However, we are keeping at it and hope that continuous optimisation and increasing sensitivities of the wet lab methods will make this possible one day.

General recommended edit:

Title - This analysis doesn't really tell us about the organization of multiple communities, but instead only a single one. I would strongly suggest a change to "Extensive pedigrees reveal social organization of Neolithic community at Gurgy 'les Noisats'"

77) We agree and have changed the title a bit differently to "Extensive pedigrees reveal the social organisation of a Neolithic community."

Specific recommended edits:

Line 43 - Arguably not only burial grounds, but also sites of habitation, work sites, etc, where evidence of exchange and interaction may be even more evident than in burials. I would change this to "..these systems can only be studied indirectly through biological and material culture remains" (or something like this)

78) We agree and have made small changes to the abstract, but are extremely limited by the word count.

Line 45- Define "burial community" so that there is no confusion as to what you mean (this may mean different things to different readers and there is a chance of confusion)

79) For reasons of brevity, we would prefer to keep this term as we cannot think of other ways to read it. Of note, none of the other three reviewers found it troublesome or misleading.

Line 48- tone this down a bit: "...and can aid in the interpretation of kinship practices in prehistoric societies."

80) We discuss biological relatedness here, not kinship in general. Archaeogenetics is a useful means to reconstruct biological relatedness in the past, which can provide a framework that in turn can inform us about social kin relations within a group.

Archaeogenetics is one part of a toolbox that allows us to make inferences from evidence that is associated with societies of the past. As such, we are specific in saying 'it can', and we don't think we are overstating the role of archaeogenetics in any way.

Line 49- this wording is strange. I recommend "genome-wide DNA data as well as strontium isotope and contextual data"

81) Thanks, we modified the sentence slightly.

Line 54- for more clarity, change 'of the linked and unlinked individuals to the pedigrees'  'of individuals both linked and unlinked to the pedigrees'

82) Thanks, we modified the sentence.

Line 57- how are stable conditions and high fertility consistent with social ties? these things can certainly exist independently. why should we believe they are connected here?

83) See also answer to 4). We made this clearer by specifying stable 'health' conditions. While we agree that human health and fertility can be uncoupled per se, we should also arguably agree that stable social ties can benefit human health which can also be beneficial for fertility (the ability to produce offspring) and low mortality (see for instance Bocquet-Appel J-P. 2002. Paleanthropological Traces of a Neolithic Demographic Transition. Current Anthropology 43(4). p 637-650, now cited in the text).

Line 58 - change to 'first and last generation of the pedigree'

84) This sentence has been modified based on the remarks of reviewer 1 (see point 5 above), which now reads: "Age-structure differences and strontium isotope results by generation indicate that the site was used for just a few decades, providing new lines of evidence for shifting sedentary farming practices during the European Neolithic."

Lines 64-70 - this explanation is excellent!

Thank you!

Lines 80-84 - review this sentence, it seems incomplete

85) We modified the sentence, now reading: "The burials feature different body positions and orientations, and architectural variation from various cultural influences, but very few grave goods, limiting our ability to identify either a direct association to the Cerny culture, the organisation of the site, or to the selection of specific individuals".

Line 84 - "duration of use of the site" would be better

86) Thanks, we modified it accordingly.

Line 86- much more precise would be "we took samples from the remains of every individual in the graveyard"

87) This sentence has been modified according to reviewer 1's comment (point 8) and now reads "To investigate the intra-site structure and the characteristics of Gurgy, we sampled the remains of 110 out of 128 individuals with suitable skeletal preservation (Materials and Methods) and retrieved genome-wide aDNA data for 94 individuals (Supplementary table 3), ..."

Lines 86-89 - this sentence has too many clauses, break up into two (preferably starting a new sentence after "22 of which were published previously")

88) We separated the sentence in two parts, as suggested. They now read "To investigate the intra-site structure and the characteristics of Gurgy, we sampled the remains of 110 out of 128 individuals with suitable skeletal preservation (Materials and Methods) and retrieved genome-wide aDNA data for 94 individuals (Supplementary table 3), 22 of which were published previously¹⁴. We also generated immune genes data for 82 individuals, mitogenomes for 100 individuals and Y chromosome data for 57 individuals".

Line 103- add "connects" after Pedigree B

89) Thanks, we added it.

Lines 105-106 - better wording would be "one adult male has two second-degree relatives in Pedigree A"

90) Thanks. We modified the sentence as suggested.

Line 107 - it would be helpful to tell at this point how many individuals out of the original 94 have no detectable relatives at all.

91) We rephrased this sentence entirely, now reading: "We identified three additional pairs of first-degree relatives, and 11 remaining individuals who are not closely related to either of the pedigrees (Fig. 1a)".

Line 109 - "for ease of use" is unnecessary

92) We removed this portion of the sentence.

Lines 109-111 - this sentence needs to be re-written because it does not work well grammatically.

93) We have rephrased this paragraph, which now reads: "Throughout this text we use the terms 'mother/father', 'son/daughter', 'siblings', etc., as well as the 'male'/'female' binary in a biological sense. We acknowledge that these are western kinship terms, but they are not meant to imply kinship terminologies or identities here. We cannot know if they were understood in this way by the Gurgy community."

Lines 122-126 - add a citation for this "recently developed method" and a pointer to the supplementary note where it is described

94) We referred to Material and Methods where the webpage of the new method is cited, as well as to Supplementary Note 3 where the method is explained.

Line 127- change further  additional

95) We removed "further" but did not replace it by "additional" because we begin the sentence with "Additionally".

Line 135 - "beyond a 3rd degree" sounds strange

96) We replaced it with "more distantly than the 3rd degree".

Line 155- remove "itself"

97) We removed it.

Line 156- not having data from GLN270A is a big bummer. Is there really no hope of generating data? If there is any way that it is possible to do so, this should be done. The conclusions made in the rest of the paragraph are substantially weakened because of the lack of data from this individual.

98) We agree... See response to point 76. However, we think that this does not weaken the actual sensation, which is the finding of the lineage founder as secondary inhumation. The majority of archaeologists and anthropologists (including ourselves) would have expected to find this person in a more elaborate/prominent grave. Again, we agree that it is regrettable that we cannot say more about the relationship and role of individual GLN270A.

Line 190 - "all adult mothers" -- how many, exactly?

99) This concerns a minimum of 7 mothers buried at the site, but also all those who are not buried there, which would make a total of 37. However, it is not trivial to estimate the number of missing mothers, as some might have never been part of the founding generations of the Gurgy community (for instance the mother of GLN253 and 275, or the mother of GLN287 and 310B). This applies to the first few generations. At the same time, we cannot estimate this number for the last generations, as it is possible that some sub-lineages left no members buried at Gurgy behind. We therefore decided to modify the sentence which now reads: "Except for two individuals (GLN325, see above, and GLN288), all adult mothers, present (N=7) or absent, have no parents/ancestors buried at the site, which suggests an exogenous origin of these females".

Lines 190-191 - does the Sr ratio data corroborate this?

100) Yes it does, and we address this aspect of the data further in lines 238-244 of the text.

Lines 191-192 - is this a significant deviation from expectation? This number actually is higher than I might expect.

101) This is also difficult to assess as we do not know the a priori expectation, unless we assume absolutely strict and impermeable rules of female exogamy, in which all daughters had to leave the community and in turn all female partners in each generation had to come from outside communities. As we explain in the main text and in Supplementary Note 14.2, the number of offspring per couple is often high, with a strong imbalance in favour of males. We count 45 adult males necessary to explain the pedigrees in generations 2 to 5 in Pedigree A and 2 to 4 in Pedigree B. According to a sex ratio at birth (1.05 male to 1 female), which is expected to be balanced out at reproductive age, we would indeed expect many more lineage daughters, about 44, which leads us to consider these 6 females as a low number.

Lines 193-194 - what does this imply?

102) We moved this sentence to the end of the paragraph, where we provide a better rationale. It now reads: "For four of the six adult lineage daughters (GLN212, 213, 277 and 289B) who remained at Gurgy, no offspring could be identified at the site even

though they had reached reproductive age. Female exogamy may not have been practiced strictly, or alternatively, these lineage daughters could be reproductive partners of unlinked adult males (with no offspring linking them to the pedigrees): a scenario which further complicates the assumption of strict patrilocality and female exogamy. Alternative reasons for their stay in the community remain elusive."

Lines 204-208 - this is really nice!

Thanks!

Lines 219-220 - can you elaborate on this potential bias? has this been seen before? how might this be explained?

103) See point 13 and our response above. We can only speculate about potential explanations. The real reasons will remain unknown.

Line 230 - change to "a sex-based dietary division"

104) We modified it as proposed.

Line 232 - give more details on this purported sex-related different treatment? what is it and why? what other evidence supports this possibility?

105) We rephrased and developed this sentence according to the reviewer 1's comment. See point 14 and our response above.

Line 236 - is this seen elsewhere?

106) Yes, this had already been observed as a tendency by Le Roy et al. 2018, as cited in the text (citation #22). We added a new sentence to make it clearer, which reads: "This pattern has been previously observed in other Neolithic sites in the northern half of France".

Lines 238-239 - is this diversity significantly higher than expected? Why do you believe this is notably high?

107) This is a good point, thank you.

We note that relative mitochondrial diversity can only remain descriptive in nature. The extremely high number of related individuals in the group complicates the calculation of the mitochondrial diversity, therefore a comparison with published population data, which assumes a random draw from unrelated individuals, would only be possible from exogenous females and additional lineages from unrelated individuals. Thus, if we take the number of different mtDNA lineages among unrelated exogenous females and additional unrelated individuals at face value, we count 35 unique mitochondrial haplotypes among 57 such individuals, resulting in a proportion of 0.614 unique haplotypes. Applying the same principle, the data from Hazleton North (Fowler et al. 2021) result in a proportion of 0.618. In comparison, the Early Neolithic LBK cemetery Derenburg (Childebayeva et al. 2022) which has been used for a longer period of time and for which only very few biological relationships have been described returns a proportion 0.75. A random, chronologically dispersed, cross-regional sample of the French Neolithic meta-population (Rivollat et al. 2020) results in a proportion of 0.84. We are not necessarily expecting to reach this high a proportion within the perimeter of the mating network of Gurgy, but note that the mtDNA diversity is not drastically reduced and can indeed track the addition of newly observed mtDNA haplotypes in each generation in each lineage. Taken together, at the moment Hazleton North presents the only suitable parallel to Gurgy. For this reason, we decided not to make a firm qualifying statement and removed the adjective "high" in the corresponding sentence. For future reference, we added our considerations and this comparison also to Supplementary Note 6.

Line 251- "confirms the will to avoid inbreeding" -- this is strange wording, can it be changed to something like "confirms the avoidance of mating among close relatives"?

108) We rephrased the sentence as proposed.

Lines 306-314 - these are really interesting findings!

Line 644- why isn't IntCal20 being used? It should be.

109) *Thanks. We fully agree and have applied IntCal20.14c to all dates, and have updated Extended table 1, Extended data Fig. 10 and the modelling section in Supplementary Note 13, accordingly.*

Reviewer Reports on the First Revision:

Referees' comments:

Referee #1 (Remarks to the Author):

As before, my comments are limited to the archaeological and anthropological components of the project, including how these incorporate the results of the scientific analyses, as these are my area of expertise. I am very satisfied that the authors have addressed all of the queries raised in my initial review in their revised manuscript and its supporting materials, and/or in their response to reviewers document. The article is now even stronger than before, and I can only reiterate my previous comments as to its importance, originality, significance and robustness. The revised article is of very high value for archaeologists studying Neolithic Europe or kinship in prehistoric communities. It presents an exceptional study yielding very important results, and in my assessment it is ready for publication in this venue.

I have just a few minor comments on the revised article and supplementary materials, but none of these require any significant alteration to the submission.

Article:

L75-6: 'funerary contexts like megaliths⁶⁻⁸ or mass graves⁹, which typically cover high-status groups or individuals'. I don't think it is established that the mass grave cited in reference 9 contained high-status individuals. Whether megalithic tombs included high status groups or individuals may have varied. This does not undermine the point that it is crucial we compare results from a wide range of mortuary contexts.

L60: 'and supportive social network' add 'a' to read 'and a supportive local network'

L114: 'Throughout this text we use the terms 'mother/father', 'son/daughter', 'siblings', etc., as well as the 'male'/'female' binary in a biological sense. We acknowledge that these are western kinship terms, but they are not meant to imply kinship terminologies or identities': Perhaps insert 'Neolithic; between 'imply' and 'kinship', and reword slightly as male/female are being used here as terms of sex (rather than kinship).

L205: 'to both main pedigrees...' should this be: 'to either main pedigree'?

L212-3: 'Perhaps as a consequence, the sex-ratio of adult offspring buried on site is unbalanced at 4.5:1'. It is not clear why this should be a consequence of reciprocal exchange.

L293: 'polygynous reproductive unions': should that be 'reproductive unions with multiple partners'?

L294: 'It also suggests that serial monogamy including levirate and sororate unions (in which a woman re-partners with her deceased husband's brother or a man re-partners with his wife's sister) were rare, in contrast to recent findings from a later Neolithic long cairn in England⁸.'

Could this be reworded slightly as, while there are indeed a number of cases of what could be serial monogamy, there are no cases of levirate or sororate unions at Hazleton North? (The closest genetic relation between two males reproducing with the same woman is between NC1m and U2m who are third degree relatives, such as cousins, or NE2m and U11m who are a half-uncle and his half-nephew).

Supplementary materials:

I think reference to supplementary table 21 in the response to reviewers p4 & p11 rather refers to a new table with file code 436497_1_data_set_4092881_rr46n9. In any case, that new xl file contains the necessary archaeological information, although there are two columns, V and W, which have their headings in red font and no field entries – these need to be completed or removed (having read the authors' response number 22, it sounds like these will be removed).

The revised supplementary document presents a very clear summary of the archaeological evidence and the interpretation of the genetic and isotopic data in relation to that. The new map in section S1 is extremely useful, as is the combination of the new burial data xl and the figure 4.

Supp p3: 'both double but mainly single' perhaps change to 'Most burials were single but there were some cases of double interment'

P4/perhaps main article L181, L329: The description of the burials as lying on their right side, combined with the general procession of burials from east to west across the use-span of the site made me wonder if this meant most burials were positioned 'looking towards' their ancestors. If so, that may be worth mentioning as another indication of the importance of descent for the community.

Reviewed by Chris Fowler

Referee #2 (Remarks to the Author):

I would like to express my appreciation for the authors' efforts in responding to all the reviewers' comments and the work done on the original manuscript. All the points I raised in my previous round of review have been satisfactorily addressed. I have no further comments on the manuscript and eagerly anticipate its publication. I believe it will be a significant milestone in the holistic study of funerary sites and an excellent model for future research.

Referee #3 (Remarks to the Author):

I thank the authors for their extensive work addressing the reviewer comments and think that the changes have improved the manuscript. The added figure in Supplementary Note 2 is a helpful addition. A few minor lurking things the authors should consider are:

1. I appreciate the change in wording regarding female versus male diversity, but I'm still a bit concerned that the claim could be overstated. The new text says, "This female diversity within the Gurgy group also explains the overall phenotypic diversity of the group (Supplementary Note 9, Supplementary tables 15 and 16)." Can the authors back up this claim more explicitly? E.g., what is the phenotypic diversity of the male founders compared to the female migrants? The information is in the tables, but it would be cumbersome for a reader to examine this.

2. The evaluation of BREAD given in the response is excellent and I think it would be nice to include in the supplement. Perhaps the authors wish to retain it for a separate publication? In that case, briefly noting a few points may be helpful, but the editor and authors can see what seems best here.

3. It is great to see the statistical test for the strontium analysis now referenced in the Figure 1 caption. The details of this test would be great to include in the supplement, especially as there are several terms included and only some are significant. Similarly, a brief description of the permutation test done on the carbon and nitrogen analysis data (now referenced in the caption of Extended data Fig. 4d) seems worthwhile.

4. This may be nitpicky, but perhaps a brief note to describe in the supplement how the sites were thinned to be 200k bp apart would be helpful. (Seems like this could be stated more succinctly than in the response to reviewers, though I appreciate the detail the authors gave!)

Referee #4 (Remarks to the Author):

My suggestions during the initial round of review for the manuscript now titled "Extensive pedigrees reveal the social organisation of a Neolithic community" have been well-addressed, and I am supportive of the publication of this paper and have no further comments.

One final recommendation is to standardize use of numbers or written-out degree relationships throughout the manuscript (e.g., choose either 1st-degree, 2nd-degree or first-degree second-degree).

Three small typos are:

Line 101 – remove comma after 2nd degree

Line 166 – capitalize pedigree

Line 265 – add “-” after “3rd” so that it reads “3rd- or 4th-degree”

Overall, this is a very exciting study with important results, and I look forward to seeing it published.

Author Rebuttals to First Revision:

Referees' comments:

Referee #1 (Remarks to the Author):

As before, my comments are limited to the archaeological and anthropological components of the project, including how these incorporate the results of the scientific analyses, as these are my area of expertise. I am very satisfied that the authors have addressed all of the queries raised in my initial review in their revised manuscript and its supporting materials, and/or in their response to reviewers document. The article is now even stronger than before, and I can only reiterate my previous comments as to its importance, originality, significance and robustness. The revised article is of very high value for archaeologists studying Neolithic Europe or kinship in prehistoric communities. It presents an exceptional study yielding very important results, and in my assessment it is ready for publication in this venue.

I have just a few minor comments on the revised article and supplementary materials, but none of these require any significant alteration to the submission.

Article:

L75-6: 'funerary contexts like megaliths⁶⁻⁸ or mass graves⁹, which typically cover high-status groups or individuals'. I don't think it is established that the mass grave cited in reference 9 contained high-status individuals. Whether megalithic tombs included high status groups or individuals may have varied. This does not undermine the point that it is crucial we compare results from a wide range of mortuary contexts.

We agree and have noticed an unintentional shift of the relevant clause. This is corrected and the sentence now reads: "Studies on biological relatedness in the European Neolithic are still rare and so far have only focussed on groups from specific funerary contexts like megaliths⁶⁻⁸, which can cover high-status groups or individuals, or mass graves⁹, but have not included non-specific graveyards that may be more representative of the general population."

We also toned down the emphasis on megalithic tombs only included high status groups/individuals.

L60: 'and supportive social network' add 'a' to read 'and a supportive local network'

Thanks. Done.

L114: 'Throughout this text we use the terms 'mother/father', 'son/daughter', 'siblings', etc., as well as the 'male'/'female' binary in a biological sense. We acknowledge that these are western kinship terms, but they are not meant to imply kinship terminologies or identities': Perhaps insert 'Neolithic; between 'imply' and 'kinship', and reword slightly as male/female are being used here as terms of sex (rather than kinship).

We do not think this overall statement applies only to the Neolithic period, but to all of them, that's why we prefer to keep it broad and leave it as it is. We rephrased the first sentence of the paragraph to clarify the terms use for kinship or for sex, which now reads "Throughout this text we use the kinship terms 'mother/father', 'son/daughter', 'siblings', etc., as well as the binary sex terms 'male'/'female' in a genetic sense.'

L205: 'to both main pedigrees...' should this be: 'to either main pedigree'?

Yes, indeed. This is corrected now.

L212-3: 'Perhaps as a consequence, the sex-ratio of adult offspring buried on site is unbalanced at 4.5:1'. It is not clear why this should be a consequence of reciprocal exchange.

As we are speaking only of the adult offspring, a reciprocal exchange would result in the absence of all adult lineage daughters, but all adult lineage sons would be present, which is what we observe in Gurgy. Therefore, we would like to keep the current sentence.

L293: 'polygynous reproductive unions': should that be 'reproductive unions with multiple partners'?

Indeed, the word "polygynous" is incorrect here, as we mean it for both sexes. We modified it with 'polygamous', and talk about serial monogamy further in the paragraph, as stated below.

L294: 'It also suggests that serial monogamy including levirate and sororate unions (in which a woman re-partners with her deceased husband's brother or a man re-partners with his wife's sister) were rare, in contrast to recent findings from a later Neolithic long cairn in England⁸.'

Could this be reworded slightly as, while there are indeed a number of cases of what could be serial monogamy, there are no cases of levirate or sororate unions at Hazleton North? (The closest genetic relation between two males reproducing with the same woman is between NC1m and U2m who are third degree relatives, such as cousins, or NE2m and U11m who are a half-uncle and his half-nephew).

We agree and have noticed an unintentional shift of the initial statement. We rephrased and the paragraph now reads 'Further insights about the social organisation of the group can be gleaned from the striking lack of half-siblings in the entire sample, in contrast to recent findings from a later Neolithic long cairn in England⁸ (Supplementary Notes 2 and 3). This indicates that polygamous reproductive unions were uncommon or perhaps socially proscribed, or that the burial of offspring from such unions was carried out elsewhere. Likewise, it also suggests that serial monogamy, including levirate and sororate unions in which a woman re-partners with her deceased husband's brother or a man re-partners with his wife's sister, was rare.'

Supplementary materials:

I think reference to supplementary table 21 in the response to reviewers p4 & p11 rather refers to a new table with file code 436497_1_data_set_4092881_rr46n9. In any case, that new xl file contains the necessary archaeological information, although there are two columns, V and W, which have their headings in red font and no field entries – these need to be completed or removed (having read the authors' response number 22, it sounds like these will be removed).

Yes, the name of Table 21 has been changed during the submission process.

Many thanks for detecting this oversight. Indeed, we removed these two columns because, as explained in the previous Response to the reviewers, the position of the grave goods within the grave is difficult to encode in a table format and it was not possible to assess the pathologies on the bones in a reliable and systematic way given the rather poor macroscopic preservation.

The revised supplementary document presents a very clear summary of the archaeological evidence and the interpretation of the genetic and isotopic data in relation to that. The new map in section S1 is extremely useful, as is the combination of the new burial data xl and the figure 4.

Supp p3: 'both double but mainly single' perhaps change to 'Most burials were single but there were some cases of double interment'

We considered the proposed sentence and decided to rephrase as 'Most uncovered pits were single graves but there were some cases of double burials.'

P4/perhaps main article L181, L329: The description of the burials as lying on their right side, combined with the general procession of burials from east to west across the use-span of the site made me wonder if this meant most burials were positioned 'looking towards' their ancestors. If so, that may be worth mentioning as another indication of the importance of descent for the community.

Thanks for this very interesting suggestion. We checked the orientation of the bodies but most of them have their head orientated toward South, while the main ancestors of Pedigrees A and B are located at the North-East of the necropolis.

Reviewed by Chris Fowler

Referee #2 (Remarks to the Author):

I would like to express my appreciation for the authors' efforts in responding to all the reviewers' comments and the work done on the original manuscript. All the points I raised in my previous round of review have been satisfactorily addressed. I have no further comments on the manuscript and eagerly anticipate its publication. I believe it will be a significant milestone in the holistic study of funerary sites and an excellent model for future research.

Many thanks for this positive feedback and the helpful comments in the previous round.

Referee #3 (Remarks to the Author):

I thank the authors for their extensive work addressing the reviewer comments and think that the changes have improved the manuscript. The added figure in Supplementary Note 2 is a helpful addition. A few minor lurking things the authors should consider are:

1. I appreciate the change in wording regarding female versus male diversity, but I'm still a bit concerned that the claim could be overstated. The new text says, "This female diversity within the Gurgy group also explains the overall phenotypic diversity of the group (Supplementary Note 9, Supplementary tables 15 and 16)." Can the authors back up this claim more explicitly? E.g., what is the phenotypic diversity of the male founders compared to the female migrants? The information is in the tables, but it would be cumbersome for a reader to examine this.

We have not compared systematically the phenotypic diversity of lineage males and exogenous females, in particular because phenotypic diversity cannot be separated into and estimated per distinct classes after the first generation(s), given the high level of biological relatedness at the site. Our statement was meant to be observational rather than a strict correlation. Therefore, we removed the word ‘diversity’ and toned down the statement, which now reads ‘Moreover, this female diversity within the Gurgy group might also explain the overall phenotypic variation observed at the site (Supplementary Note 9, Supplementary tables 15 and 16).’

2. The evaluation of BREAD given in the response is excellent and I think it would be nice to include in the supplement. Perhaps the authors wish to retain it for a separate publication? In that case, briefly noting a few points may be helpful, but the editor and authors can see what seems best here.

Thanks. Indeed, a parallel publication has just been submitted on bioRxiv (<https://www.biorxiv.org/content/10.1101/2023.04.17.537144v1>). We provide the new link in the manuscript.

3. It is great to see the statistical test for the strontium analysis now referenced in the Figure 1 caption. The details of this test would be great to include in the supplement, especially as there are several terms included and only some are significant. Similarly, a brief description of the permutation test done on the carbon and nitrogen analysis data (now referenced in the caption of Extended data Fig. 4d) seems worthwhile.

Thanks. We added the explanation of the statistical test about strontium data in the supplementary p 35, and the new paragraph now reads “To statistically test for the significance of sex, age, and generation in predicting the distance between burials, we ran a multivariate linear regression, beginning with a full interaction model including these three variables. We used ANOVA (p-value cut-off of 0.05), and stepwise AIC (Akaike information criterion), to compare the nested models to find the simplest well-fit model. We found that the model which fits best is a model with generation and sex, as well as an interaction term between sex and generation, as predictor variables. The age group variable however could be completely removed from the model. The diagnostic plots for the models were all inspected to ensure that the assumptions of linear regression were satisfied. The coefficient of the interaction term "male:generation" was negative, meaning that per generation, the strontium ratio for females was increasing significantly faster than it was for males (p=0.01474).”

We added the detailed explanation of the permutation test done on subadults for carbon and nitrogen data p 45 of the supplementary, which now reads ‘However, carbon, nitrogen, sulphur isotope data highlight a significant dietary, sex-based difference in adults that could reflect a gender-biased access and/or differential treatment³⁶. To explore this sex-based difference in subadults (Extended data Fig. 4d), we ran a cluster comparison using the Calinski-Harabasz Index and calculated the ratio of the sum of between-cluster dispersion and of inter-cluster dispersion for all clusters. The empirical p-value, calculated using a permutation test with all $17!/9! = 24,310$ permutations of the possible cluster labels, is significant (p=0.01069), and remains significant when including the “undetermined” individuals, meaning the same difference can be observed in the diets of males and females during childhood (Supplementary table 24, Extended data Fig. 4d).’

4. This may be nitpicky, but perhaps a brief note to describe in the supplement how the sites were thinned to be 200k bp apart would be helpful. (Seems like this could be stated more succinctly than in the response

to reviewers, though I appreciate the detail the authors gave!)

We added a simplified explanation of this process p7 of the supplementary, now reading ‘Here, for individuals i and j , we thinned the data such that all sites were at least 200K bases apart to best satisfy the assumption of independence. We did so by taking all of the overlapping sites, and their relative positions. We started with chromosome 1, where we took the first overlapping site, retained it, then took the next site that was a minimum of 200K sites along the chromosome, and retained it, etc. We repeat this until we had no overlapping sites left. We then repeat this “per chromosome” process for chromosomes 2 through to 22. In this way we found a likely optimally large set of thinned, overlapping SNPs.’

Referee #4 (Remarks to the Author):

My suggestions during the initial round of review for the manuscript now titled "Extensive pedigrees reveal the social organisation of a Neolithic community" have been well-addressed, and I am supportive of the publication of this paper and have no further comments.

One final recommendation is to standardize use of numbers or written-out degree relationships throughout the manuscript (e.g., choose either 1st-degree, 2nd-degree or first-degree second-degree).

Thanks, we standardized the degrees of relatedness, opting for the use of numbers.

Three small typos are:

Line 101 – remove comma after 2nd degree

Line 166 – capitalize pedigree

Line 265 – add “-“ after “3rd” so that it reads “3rd- or 4th-degree”

Thanks. All three instances have been corrected.

Overall, this is a very exciting study with important results, and I look forward to seeing it published. Many thanks for the positive feedback and also in the previous round, which have helped to improve the manuscript substantially.